# A weakly structured stem for human origins in Africa

Aaron P. Ragsdale[1], Timothy D. Weaver[2], Elizabeth G. Atkinson[3], Eileen G. Hoal[4,5,6], Marlo Möller[4,5,6], Brenna M. Henn[2,7 ✉] & Simon Gravel[8 ✉]

Despite broad agreement that *Homo sapiens* originated in Africa, considerable uncertainty surrounds specific models of divergence and migration across the continent[1]. Progress is hampered by a shortage of fossil and genomic data, as well as variability in previous estimates of divergence times[1]. Here we seek to discriminate among such models by considering linkage disequilibrium and diversity-based statistics, optimized for rapid, complex demographic inference[2]. We infer detailed demographic models for populations across Africa, including eastern and western representatives, and newly sequenced whole genomes from 44 Nama (Khoe-San) individuals from southern Africa. We infer a reticulated African population history in which present-day population structure dates back to Marine Isotope Stage 5. The earliest population divergence among contemporary populations occurred 120,000 to 135,000 years ago and was preceded by links between two or more weakly differentiated ancestral *Homo* populations connected by gene flow over hundreds of thousands of years. Such weakly structured stem models explain patterns of polymorphism that had previously been attributed to contributions from archaic hominins in Africa[2–7]. In contrast to models with archaic introgression, we predict that fossil remains from coexisting ancestral populations should be genetically and morphologically similar, and that only an inferred 1–4% of genetic differentiation among contemporary human populations can be attributed to genetic drift between stem populations. We show that model misspecification explains the variation in previous estimates of divergence times, and argue that studying a range of models is key to making robust inferences about deep history.

Decades of study of human genome variation have suggested a predominantly tree-like model of recent population divergence from a single ancestral population in Africa. It has been difficult to reconcile this finding with the fossil and archaeological records of human occupation across the vast African continent. For example, fossils such as those from the sites of Jebel Irhoud in Morocco[8], Herto in Ethiopia[9] and Klasies River in South Africa[10] demonstrate that derived *Homo sapiens* anatomical features were found across the continent 300–100 thousand years ago (ka). Archaeological sites from the Middle Stone Age, of which some have been associated with *H. sapiens*, are also widely distributed across Africa. It is unclear whether these fossils and archaeological sites represent populations that contributed to contemporary *H. sapiens* as population precedents or were local 'dead ends'. Attempts to reconcile genetic and palaeoanthropological data include proposals[11–13] for a pan-African origin of *H. sapiens* in which populations in many regions of the continent contributed to the formation of *H. sapiens* beginning at least 300 ka.

Genetic models have been hampered in their contribution to this discussion because they primarily assume (or, at least, have been tested under) a tree-like model of isolation with migration. Alternative theoretical scenarios have been proposed, such as stepping-stone models[14] or population coalescence and fragmentation[13], but these approaches are more challenging to interpret and fit to data. However, new population-genetic tools now allow for inference on the basis of tens to hundreds of genomes from multiple populations and for greater model complexity[2,15,16]. Inspired by evidence for Neanderthal admixture with humans in Eurasia, several studies have shown that introducing an archaic hominin 'ghost' population contributing to African populations in the period surrounding the out-of-Africa migration event substantially improves the description of genetic data relative to single-origin models, mostly in western Africa[2–7], but also in southern[4,6] and central African[4–6,17] populations. This has driven speculation about the geographical range of the ghost population, possible links to specific fossils and the possibility of finding ancient DNA evidence[17]. However, these studies share two weaknesses. First, they contrast only a single-origin model with an archaic hominin admixture model, leaving out other plausible models[1] (Fig. 1). Second, they

[1]Department of Integrative Biology, University of Wisconsin–Madison, Madison, WI, USA. [2]Department of Anthropology, University of California, Davis, CA, USA. [3]Department of Molecular and Human Genetics, Baylor College of Medicine, Houston, TX, USA. [4]DSI-NRF Centre of Excellence for Biomedical Tuberculosis Research, Stellenbosch University, Cape Town, South Africa. [5]South African Medical Research Council Centre for Tuberculosis Research, Stellenbosch University, Cape Town, South Africa. [6]Division of Molecular Biology and Human Genetics, Faculty of Medicine and Health Sciences, Stellenbosch University, Cape Town, South Africa. [7]Genome Center, University of California, Davis, CA, USA. [8]Department of Human Genetics, McGill University, Montreal, Quebec, Canada. ✉e-mail: bmhenn@ucdavis.edu; simon.gravel@mcgill.ca

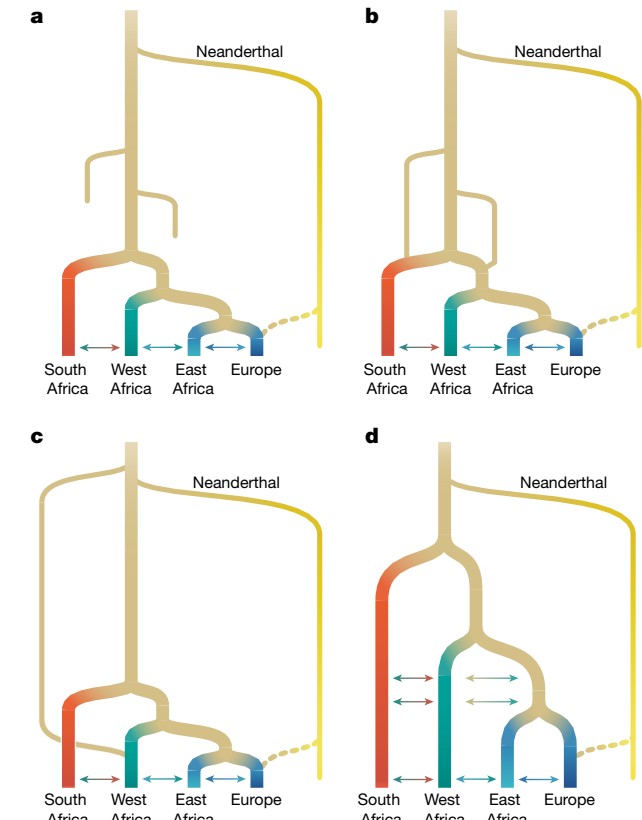

**Fig. 1 | Proposed conceptual models of early human history in Africa.**
**a**, Recent expansion. **b**, Recent expansion with regional persistence. **c**, Archaic admixture. **d**, African multiregional. The models have been designed to translate models from the palaeoanthropological literature into genetically testable demographic models (ref. 1 and Supplementary Information section 3). These parameters were then fitted to genetic data.

focus on a small subset of African diversity, either because of small sample sizes (2–5 genomes) or because they rely on data from the 1000 Genomes Project[18], which was limited to populations of recent West African or Bantu-speaking ancestry (Fig. 2). Ancient DNA from Eurasia has helped to clarify early human history outside Africa, but there is no comparably ancient DNA to elucidate early history in Africa[19].

We therefore aim to discriminate between a broader set of demographic models by studying the genomes of contemporary populations. We take as our starting point four models (single-population expansion, single-population expansion with regional persistence, archaic hominin admixture and multi-regional evolution; Fig. 1) using 290 genomes of individuals from southern, eastern and western Africa, as well as Eurasia. By including geographically and genetically diverse populations across Africa, we infer demographic models that explain more features of genetic diversity in more populations than previously reported. These analyses confirm the inadequacy of tree-like models and provide an opportunity to directly evaluate a wide range of alternative models.

We inferred detailed demographic histories using 4x–8x whole-genome sequencing data for four diverse African populations, comprising the Nama (Khoe-San from South Africa, newly presented here; see Supplementary Information section 1.2 for ethical and practical aspects of participant recruitment), the Mende (from Sierra Leone; from phase 3 of the 1000 Genomes Project[18]), the Gumuz (recent descendants of a hunter-gatherer group from Ethiopia[20,21]) and eastern African agriculturalists (Amhara and Oromo from Ethiopia[20]). The Amhara and Oromo populations, despite speaking distinct Afro-Asiatic languages, are highly genetically similar[21,22] so we combined the two groups for a

larger sample size (Fig. 2). We also included British individuals from the 1000 Genomes Project in our demographic models as a representative source of back-to-Africa gene flow and recent colonial admixture in South Africa. Finally, we used a high-coverage ancient Neanderthal genome from Vindija Cave in Croatia[23] to account for gene flow from Neanderthals into people from outside Africa, and gauge the relative time depth of divergence, assuming that Neanderthals diverged 550 ka from a common stem. We computed one- and two-locus statistics for which the expectation within and across populations can be computed efficiently and that are well suited for both low- and high-coverage genomes[2,24]. Using a maximum-likelihood inference framework, we then fitted to these statistics a family of parameterized demographic models that involve population splits, size changes, continuous and variable migration rates and punctuated admixture events, to learn about the nature of the population structure over the past million years.

## A Late Pleistocene common ancestry

We started with a model of geographical expansion from a single ancestral, unstructured source followed by migration between populations, without allowing for a contribution from an African archaic hominin lineage (Fig. 1a) or population structure before the expansion (Fig. 1d). As expected[2], this first model was a poor fit to the data qualitatively (Supplementary Fig. 10) and quantitatively (log-likelihood (LL) ≈ −189,300; Supplementary Table 3). We next explored a suite of parameterized models in which population structure predates the differentiation of contemporary groups (Supplementary Information section 3). Depending on the parameters, these encompassed models allowing for ancestral reticulation, such as fragmentation-and-coalescence or meta-population models (Fig. 1b), archaic hominin admixture (Fig. 1c) and African multi-regionalism (Fig. 1d). The recent expansion and the African multi-regional models (Fig. 1a,d) have the same topology, so interpretation of the model depends on the specified or inferred divergence times.

Regardless of the model choice for early epochs, maximum-likelihood inference of human demographic history for the past 150 kyr was remarkably robust. In a reticulated model, we use 'divergence' between populations to mean the time of their most recent shared ancestry. The earliest divergence among contemporary human populations differentiates the southern African Nama population from the other African groups at 110–135 ka, with low to moderate levels of subsequent gene flow (Table 1). In none of the high-likelihood models that we explored was the divergence between Nama and other populations earlier than around 140 ka. We conclude that geographical patterns of contemporary *H. sapiens* population structure probably arose during MIS 5. Although we do find evidence for earlier population structure in Africa, contemporary populations cannot be easily mapped onto the more ancient 'stem' groups because only a small proportion of drift between contemporary populations can be attributed to drift between stems (Fig. 4, Supplementary Information section 5.2 and Supplementary Figs. 16–19).

Given this consistency in inferred recent history and the numerical challenge of optimizing a large number of parameters, we fixed several parameters related to recent population history to focus on more-ancient events (Supplementary Information section 3.1). These parameters were ones supported by multiple genetic and archaeological studies[25]. Fixed parameters included the time of divergence between western and eastern African populations, set to 60 ka, just before the split of Eurasians and East Africans at 50 ka. We also fixed the amount of admixture from Neanderthals to the European population directly after the out-of-Africa migration to 1.5% at 45 ka.

We quantify the migration rates of populations after their divergence at around 120 ka. Before the agropastoralist expansion 5 ka, migration between the ancestors of the Nama and other groups is an order of magnitude weaker than that observed between western and eastern Africans (Table 1). All models infer relatively high gene flow between eastern

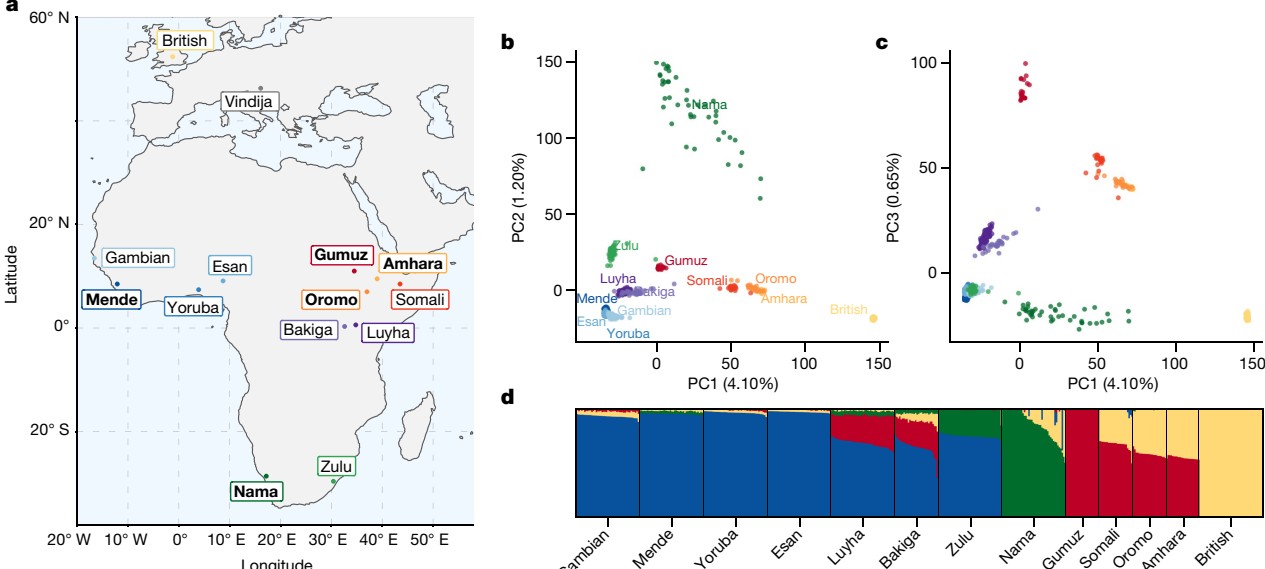

**Fig. 2 | Genetic diversity across Africa. a**, Selected populations from the 1000 Genomes Project and the African Diversity Reference Panel[18,20] illustrate diversity from western, eastern and southern Africa. We chose representative ethnic groups from each region (bold labels) to build parameterized models, including the newly genetically sequenced Nama populations from South Africa, Mende from Sierra Leone, Gumuz, Oromo and Amhara from Ethiopia, British individuals and a Neanderthal from Vindija Cave, Croatia. **b,c**, Principal component analysis highlights the range of genetic divergence anchored by western African, Nama, Gumuz and British individuals between principal components (PC) 1 and 2 (**b**), and 1 and 3 (**c**). Percentages show variance explained by each principal component. Colours represent the groups shown in bold in **a**. **d**, ADMIXTURE analysis using $K = 4$ principal components reveals signatures of recent gene flow in Africa that reflect colonial-period migration into the Nama, back-to-Africa gene flow among some Ethiopians, and Khoe-San admixture in the Zulu population.

and western Africa ($m \approx 2 \times 10^{-4}$, the constant proportion of migrant lineages per generation since their divergence 60 ka). We further find that back-to-Africa gene flow at the beginning of the Holocene epoch primarily affected the ancestors of the Ethiopian agricultural populations[26], comprising almost 65% of their genetic ancestry. We observe considerable gene flow from the Amhara and Oromo into the Nama, a signal that is probably a proxy for the movement of eastern African caprid (goat) and cattle pastoralists[27,28], here estimated to constitute a 25% ancestry contribution 2 ka. Although this gene flow is not apparent from the ADMIXTURE plot (Fig. 2), the ancestry is probably grouped into the Khoe-San component, which has drifted appreciably from its ancestral eastern African source. Colonial-period admixture from Europeans into the Nama was estimated at 15%, similar to proportions suggested by ADMIXTURE (Fig. 2).

## A weakly structured stem within Africa

To account for the population structure before 135 ka, three of our four models allowed for two or more stem populations, which could diverge either before or after the split from the Neanderthals. We considered models both with and without migration between these stem populations, and in both cases we tested two different types of gene flow during the expansion phase, as illustrated in Supplementary Fig. 6: in the first, one of the stem population expands (splits into contemporary populations), followed by continuous symmetric migration with the other stem population(s); in the second, one or more of the stem populations expands, with instantaneous 'pulse' (merger) events from the other stem population, so that recent populations are formed by mergers of multiple ancestral populations. Depending on the parameter values, this scenario encompasses archaic hominin introgression and fragmentation-and-coalescence models (such as Fig. 1b,c). For many parameters, confidence intervals based on bootstrapping are relatively narrow (Supplementary Tables 3–7), reflecting an informative statistical approach. However, model assumptions have

a greater effect on parameter estimates (and thus real uncertainty). To convey the uncertainty in the models, we highlight features of the two inferred models with high likelihoods. These are referred to as the multiple-merger and the continuous-migration models. Both allow for migration between stem branches, but differ primarily in the timing of the early divergence of stem populations and their relative effective population size ($N_e$) (Fig. 3). The two models also differ in the mode of divergence, with the multiple-merger model featuring a population reticulation (that is, loops in the population graph; Fig. 1b) during the Middle Pleistocene epoch (780 ka to 130 ka).

Allowing for continuous migration between the stem populations substantially improves the fits relative to zero migration between stems (LL $\approx$ −101,600 compared with −107,700 in the merger model, Supplementary Tables 6 and 7; and LL $\approx$ −115,300 versus −126,500 in the continuous migration model, Supplementary Tables 4 and 5). With continuous migration between stems, population structure extends back to more than 1 million years ago (Table 1). Migration between the stems in these models is moderate, with a fraction of migrant lineages ($m$) in each generation estimated as $m = 6.3 \times 10^{-5}$–$1.3 \times 10^{-4}$. For comparison, this is similar to the inferred migration rates between connected contemporary populations over the past 50 ka (Table 1). This ongoing (or at least, periodic) gene flow qualitatively distinguishes these models from previously proposed archaic hominin admixture models (Fig. 1c), as the early branches remain closely related and each branch contributes large amounts to all contemporary populations (Fig. 4). Because of this relatedness, only 1–4% of genetic differentiation among contemporary populations can be traced back to this early population structure (Supplementary Information section 5.2).

Under the continuous-migration model, one of the two stems (stem 1) diverges into lineages leading to contemporary populations in western, southern and eastern Africa, and the other (stem 2) contributes variable ancestry to those populations. This migration from stem 2 is highest with the Mende ($m = 1.6 \times 10^{-4}$) compared with the Nama and populations from eastern Africa ($m = 5.9 \times 10^{-5}$ and $3.1 \times 10^{-5}$, respectively),

**Table 1 | Migration and divergence parameters from best-fit models**

| Likelihood | Label | Population pair | Divergence time (ka) | Migration rate per generation | Migration duration (kyr) |
|---|---|---|---|---|---|
| **Continuous-migration model** | | | | | |
| LL=−115,300 | a | Stem1, stem2 | 1,223 | $6.26 \times 10^{-5}$ | 1,089 |
| | b | Stem2, Nama | NA | $5.85 \times 10^{-5}$ | 129 |
| | c, d | Stem2, other Africans[a] | NA | $3.10 \times 10^{-5}$, **$1.62 \times 10^{-4}$** | 129, 55 |
| | e, f | Nama, other Africans[a] | 135 | $4.10 \times 10^{-5}$, $9.20 \times 10^{-6}$ | 134, 60 |
| | g | Mende, East Africans | 60 | **$2.13 \times 10^{-4}$** | 60 |
| | h | East Africans, British | 50 | $4.16 \times 10^{-5}$ | 50 |
| | i | Gumuz, Amhara/Oromo | 12 | **$3.37 \times 10^{-4}$** | 12 |
| **Merger model** | | | | | |
| LL=−101,600 | a | Stem1, stem2 | 1,692 | **$1.26 \times 10^{-4}$** | 1,213 |
| | – | Stem1S, stem1E | 479 | 0 (fixed) | – |
| | b | Stem2 to Nama | 119 | **0.70** | Pulse |
| | c | Stem2 to stem1E | 98 | **0.52** | Pulse |
| | d | Stem2 to Mende | 25 | **0.19** | Pulse |
| | e, f | Nama, other Africans[a] | 119 | $4.5 \times 10^{-5}$, $9.8 \times 10^{-6}$ | 120, 60 |
| | g | Mende, East Africans[a] | 60 | **$1.97 \times 10^{-4}$** | 60 |
| | h | East Africans, British | 50 | $3.82 \times 10^{-5}$ | 50 |
| | i | Gumuz, Amhara/Oromo | 12 | **$3.59 \times 10^{-4}$** | 12 |

Labelled migration rates correspond to the symmetric continuous-migration bands shown in Fig. 3. Both the continuous-migration and the merger models inferred a relatively deep split of human stem branches, although these branches were connected by ongoing migration that maintained their genetic similarity. Bold text indicates migration rates above $10^{-4}$. In both models, the branch ancestral to the Nama shares a common ancestral population with the other African groups around 120–135 ka. After this divergence, the population ancestral to other African groups branched into West and East African groups at 60 ka.

[a]Migration rates and durations are shown between branches ancestral to Nama and East Africans and their ancestors, and Nama and Mende, respectively. Divergence times correspond to the most recent common ancestral population and do not account for continuous migration or earlier reticulations. Further information for the continuous model is provided in Supplementary Table 5 and for the merger model in Supplementary Table 7. NA, not applicable.

with migration allowed to occur until 5 ka. A sampled lineage from the Nama, Mende and Gumuz have probabilities of being in stem 2 at the time of stem 1 expansion (135 ka) of approximately 0.145, 0.20 and 0.130, respectively, although these probabilities change over time, precluding the notion of a fixed admixture proportion.

By contrast, in the multiple-merger model, stem populations merge with varying proportions to form the different contemporary groups. We observe a sharp bottleneck in stem 1 down to $N_e = 100$ after the split of the Neanderthal branch. This represents the lower bound allowed in our optimization (an $N_e$ of 100), although the size of this bottleneck is poorly constrained (95% confidence interval 100–851). After a long period of exchange with stem 2, stem 1 then fractures into stem 1E and stem 1S at 478 ka. The timing of this divergence was also poorly constrained (95% confidence interval 276–478 ka). These populations evolve independently until 119 ka (101–125 ka) when stem 1S and stem 2 combine to form the ancestors of the Nama, with proportions of 30% and 70%, respectively. Similarly, stem 1E and stem 2 combine in equal proportions (50% each) to form the ancestors of the western and eastern Africans (and thus also all individuals who later disperse during the out-of-Africa event). Finally, the Mende receive a large additional pulse of gene flow from stem 2, replacing 19% (18–21%) of their population 25 ka (22–26 ka). The later stem 2 contribution to the western African Mende resulted in better model fits ($\Delta LL \approx 60,000$). This may indicate that an ancestral stem 2 population occupied western or central Africa, broadly speaking. The differing proportions in the Nama and eastern Africans may also indicate a geographical separation of stem 1S in southern Africa and stem 1E in eastern Africa.

To assess the robustness of the inferred models to analysis and reference population choices, Supplementary Information sections 6 and 7 include reanalyses with changes in the European and West African populations, as well as the recombination maps, filtering strategies and parameter optimization strategies. Although we find some differences in the inferred parameters (see Supplementary Information sections 7.1.1 and 7.2), the best-fit models across all reanalyses are quantitatively consistent.

## Reconciling lines of genetic evidence

Previous studies have found support for archaic hominin admixture in Africa using two-locus statistics[2,17], conditional site frequency spectra (cSFS)[7] and the reconstruction of gene genealogies[16]. However, none of these studies considered a weakly structured stem. We validated our inferred models with additional independent approaches. We find that the observed cSFS (conditional on the derived allele being carried in the Neanderthal sample) is well described by the merger model (Fig. 5a–c and Supplementary Figs. 20–23), even though this statistic was not used in the fit. Our best-fit models outperform archaic hominin admixture models fitted directly to the cSFS (for example, compare with figure 1 in ref. 7). Specifically, it is the addition of migration between stems that results in a qualitative improvement of the agreement (compare Supplementary Figs. 22 and 23).

We used the software Relate[16] to infer the distribution in the coalescence rates over time in both real data and data simulated from our inferred models. Many previous studies have found a reduction of coalescence rates between 1 million years ago and 100 ka in humans and thus inferred an increase in $N_e$ during the same period[29]. This increase in inferred $N_e$ could be attributable to either an increase in population size or to ancestral population structure during the Middle Pleistocene[30]. All the models, including the single-origin model, recapitulate an inferred ancestral increase in $N_e$ between 100 ka and 1 million years ago (Supplementary Fig. 26 and Supplementary Information section 7.3.2). The single-origin model achieves this by an increase in $N_e$ during that period, whereas the best-fit models recapitulate this pattern without corresponding changes in population size.

Relative cross-coalescence rates (RCCRs) have recently been used to estimate divergence between pairs of populations, as measured by the

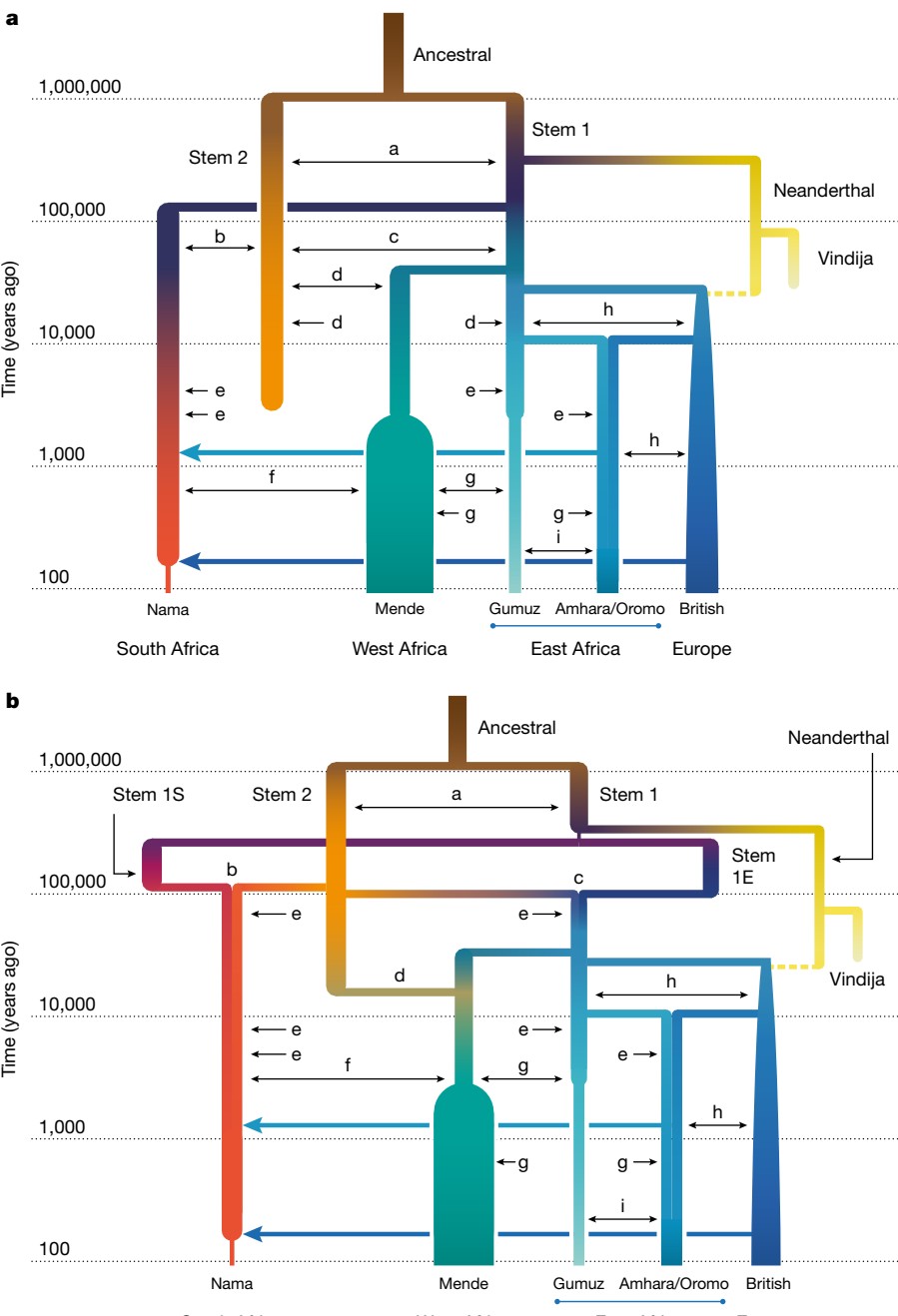

**Fig. 3 | A weakly structured stem best describes two-locus statistics.**
**a**,**b**, In the two best-fitting parameterizations of early population structure, continuous migration (**a**) and multiple mergers (**b**), models that include ongoing migration between stem populations outperform those in which stem populations are isolated. Most of the recent populations are also connected by continuous, reciprocal migration that is indicated by double-headed arrows (labels matched to migration rates and divergence times in Table 1). These migrations last for the duration of the coexistence of contemporaneous populations with constant migration rates over those intervals. The merger-with-stem-migration model (**b**, with LL = −101,600) outperformed the continuous-migration model (**a**, with LL = −115,300). Colours are used to distinguish overlapping branches. The letters a–i represent continuous migration between pairs of populations, as described in Table 1.

rate of coalescence between two groups divided by the mean within population coalescence. Simulations of RCCR accuracy, however, focus on a clean split between populations, whereby groups diverge without subsequent gene flow. Published estimates[25] of the earliest human divergences with RCCR, which range from 150 ka to 100 ka, may be substantially biased when compared with more-complex models with gene flow as inferred here. We find that midpoint estimates of RCCR are poor estimates for population divergence, often underestimating divergence time by 50% or more (for example, Mende versus Gumuz

is about 15 ka compared with a true divergence of 60 ka), and recent migration can lead to the misordering of divergence events (Fig. 5e). We suggest that RCCR analyses that do not fit multiple parameters, including gene flow, should be interpreted with caution.

Other studies[1,25] have fitted tree-like demographic models to African populations using distributions of allele frequencies or related statistics, finding inconsistent divergence times, some of which are older than those we find here. In Supplementary Information section 7.4, we show that this discrepancy can be explained by model misspecification:

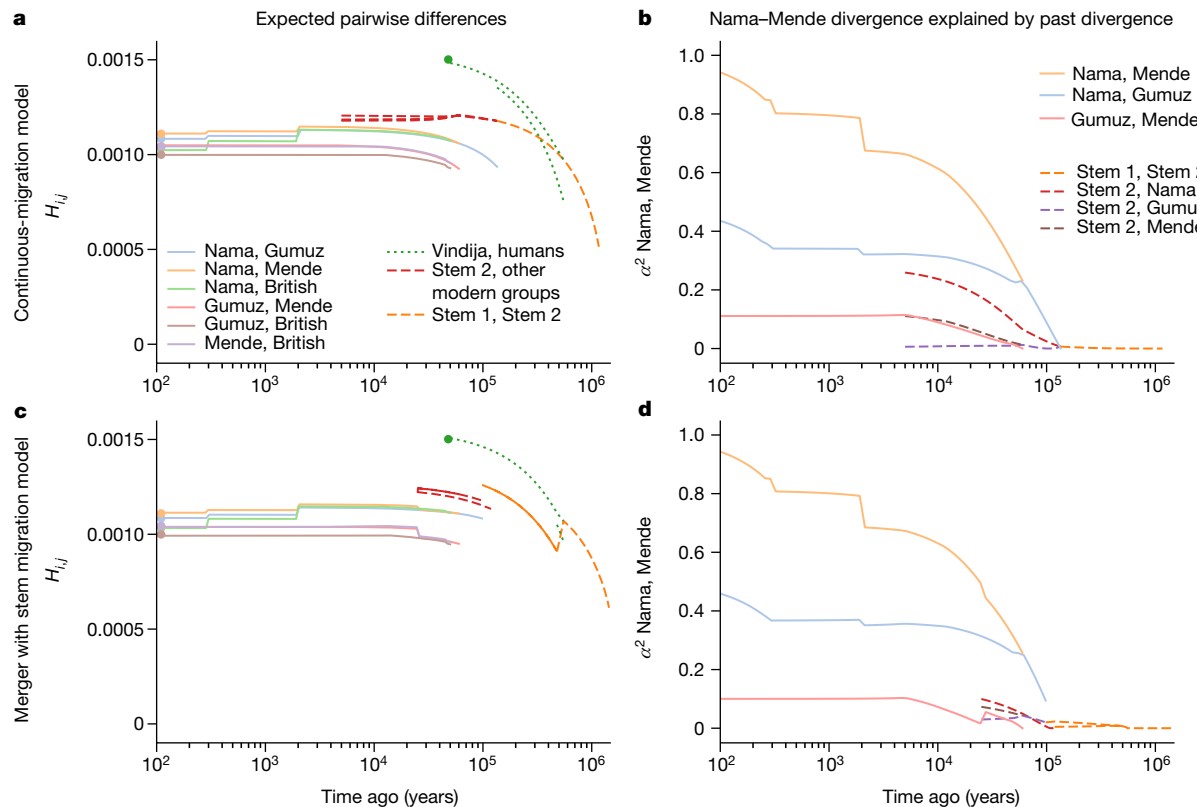

**Fig. 4 | Structure among stems is weak and present-day structure is generally recent. a–d**, From the best-fit models of our two parameterizations (**a**,**b**, continuous migration; **c**,**d**, merger with stem migration), we predicted differentiation and shared drift between populations at past time points. **a**,**c**, We computed expected pairwise differences $H_{i,j}$ between individuals sampled from populations $i$ and $j$ existing at time $t$. **b**,**d**, To understand how drift between stems explains contemporary structure, we computed the

proportion $\alpha^2$ of drift between pairs of sampled contemporary populations (here the Nama and Mende) that aligns with drift between past populations (see Supplementary Information section 5.2 for details and additional comparisons in Supplementary Figs. 16–19). Both models infer deep population structure with modest contributions to contemporary genetic differentiation. Most present-day differentiation dates back to the past 100 kyr.

if divergence is estimated by using an isolation with migration model with constant population sizes, but the correct model has ancient population growth or population structure, the divergence time in the inferred model is much earlier than in the correct model. Intuitively, growth or structure in the ancestral population will each increase coalescence times relative to a randomly mating population of constant size, so a model that assumes constant population sizes would require an older divergence time to fit the observed distribution of coalescence times and related statistics[31,32].

## Discussion

Any attempt to build detailed models of human history is subject to model misspecification. This is true of previous studies, which often assumed that data inconsistent with a single-origin model should be explained by archaic hominin admixture. It is also true of this study. Although it is difficult to fully explore the space of plausible models of early human population structure, we sought to capture uncertainty in the model by exploring multiple parameterizations of early history. The best-fit models presented here include reticulation and migration between early human populations, rather than archaic hominin admixture from long-isolated branches (Fig. 1c). Elements of both recent expansion and African multiregionalism (Fig. 1a,d) feature in our best-fit models, as indicated in the recent time of contemporary population divergence and the gene flow between disparate stems, respectively.

We cannot rule out the possibility that more-complex models involving additional stems, more-complex population structure, or hybrid models including both weak structure and archaic hominin admixture,

may better explain the data. Because parameters related to the split time, migration rates and relative sizes of the early stems were variable across models, reflecting a degree of confounding among these parameters, we refrained from introducing additional branches associated with more parameters during that period. Rather than interpreting the two stems as representing well-defined and stable populations over hundreds of thousands of years, we interpret the weakly structured stem as consistent with a population fragmentation-and-coalescence model[13]. Other African populations, such as those from Central Africa, other Khoe-San groups or pre-Holocene ancient DNA samples, could further test our proposed models.

### Formation of population structure in Africa

Our inferred models paint a consistent picture of the Middle to Late Pleistocene as a critical period of change, assuming that estimates from the recombination clock accurately relate to geological chronologies (Supplementary Information section 8). During the late Middle Pleistocene, the multiple-merger model indicates three major stem lineages in Africa, tentatively assigned to southern (stem 1S), eastern (stem 1E) and western/central Africa (stem 2). Geographical association was informed by the present population location with the greatest ancestry contribution from each stem. For example, stem 1S contributes 70% to the ancestral formation of the Khoe-San. The extent of the isolation 400 ka between stem 1S, stem 1E and stem 2 suggests that these stems were not proximate to each other. Although the length of isolation among the stems is variable across fits, models with a period of divergence, isolation and then a merger event (that is, a reticulation) out-performed models with bifurcating divergence and continuous gene flow.

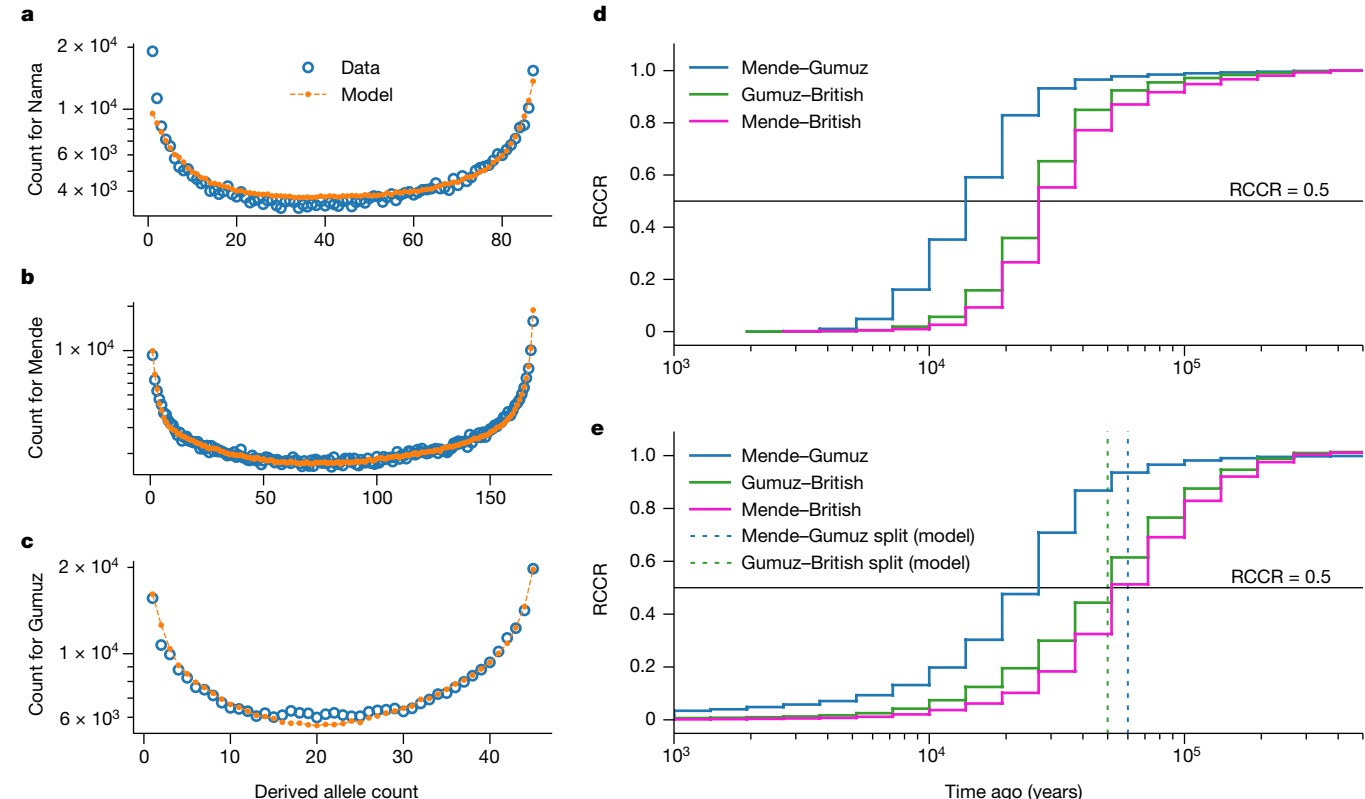

**Fig. 5 | Model validation using independent statistics. a–c**, Using our best-fit models, we simulated expected cSFS and compared the simulated spectra to those observed from the data. Our inferred models provide a good fit to the data, even though this summary was not used in our inference. Across the three populations (**a**, Nama; **b**, Mende; **c**, Gumuz), ancestral-state misidentification was consistently inferred to be 1.5–1.7% for intergenic loci (Supplementary Information section 6.2.2). **d,e**, We used Relate[16] to reconstruct genome-wide genealogies, which we used to estimate coalescence-rate trajectories and cross-coalescence rates between pairs of populations. Although coalescence-rate distributions are informative about past evolutionary processes, interpretation can be hindered by migration and population structure, and translating RCCR curves into population divergence times is especially prone to misinterpretation. **d**, Real data; **e**, our model. In our model, the Mende–Gumuz split occurs before the Gumuz–British split. However, the model also predicts a recent elevated Mende–Gumuz RCCR. This pattern, also observed in the data, does not indicate that the Mende and Gumuz split more recently than the Gumuz and British populations.

A population reticulation involves multiple stems that contribute genetically to the formation of a group. One way in which this can happen is through the geographical expansion of one or both stems. For example, if, during MIS 5, either stem 1S (Fig. 3b) from southern Africa moved northwards and thus encountered stem 2, or stem 2 moved from central–western Africa southwards into stem 1S, then we could observe disproportionate ancestry contributions from different stems in contemporary groups. We observed two merger events. The first, between stem 1S and stem 2, resulted in the formation of an ancestral Khoe-San population around 120 ka. The second event, between stem 1E and stem 2 about 100 ka, resulted in the formation of the ancestors of eastern and western Africans, including the ancestors of people outside Africa. Reticulated models do not have a unique and well-defined basal human population divergence. We suggest conceptualizing the events at 120 ka as the time of most recent shared ancestry among sampled populations. However, interpreting population divergence times in population genetics is always difficult, owing to the co-estimation of divergence time and subsequent migration; methods assuming clean and reticulated splits can infer different split dates (Supplementary Figs. 28 and 36). Therefore, in the literature, wide variation exists in estimates of divergence time[1,25].

Shifts in wet and dry conditions across the African continent between 140 ka and 100 ka may have promoted these merger events between divergent stems. Precipitation does not neatly track interglacial cycles in Africa, and heterogeneity across regions may mean that the beginning of an arid period in eastern Africa is conversely the start of a wet period in southern Africa[33]. The rapid rise in sea levels during the MIS 5e interglacial might have triggered migration inland away from the coasts, as has been suggested, for example, for the palaeo-Agulhas plain[34]. After these merger events, the stems subsequently fractured into subpopulations which persisted over the past 120 ka. These subpopulations can be linked to contemporary groups despite subsequent gene flow across the continent. For example, a genetic lineage sampled in the Gumuz has a probability of 0.7 of being inherited from the ancestral eastern subpopulation 55 ka, compared with a probability of 0.06 of being inherited from the southern subpopulation (see Table S8 for additional comparisons).

We also find that stem 2 continued to contribute to western Africans during the Last Glacial Maximum (26 ka to 20 ka), indicating that this gene flow probably occurred in western and/or central Africa (Table 1). Such an interpretation is reinforced by differential migration rates between regions; that is, the gene flow from stem 2 to western Africans is estimated to be five times that of the rate to eastern Africans during this period. We performed a variety of validation tests to explore the sensitivity of our assumptions, including relaxing fixed parameters (Supplementary Information section 6). Most of the validation tests resulted in parameters similar to the models discussed above. However, one exception was the inferred out-of-Africa and eastern–western African divergences, which were 10–15 ka younger than our fixed parameters. These younger dates are at odds with the accepted timing of the

out-of-Africa expansion that contributed to later human populations at approximately 50 ka, based on archaeological, climatic and fossil information[35–38]. Because the inference approach is unbiased in simulations, we interpret the free estimate for eastern African versus European divergence as reflecting our inclusion of only a single out-of-Africa population in the model, the lack of a nearby source for back-to-Africa gene flow, and other regionally complex parameters, rather than a systematic bias that may affect all parameters in the model. Older pan-African features of our inferred models are minimally affected by the choice of these fixed parameters (Supplementary Information section 7.2).

### Contrasting ancestral structure models

Evidence for archaic hominin admixture in Eurasia has bolstered the plausibility of archaic hominin admixture having also occurred in Africa. Previous work that sought to explain patterns of polymorphism inconsistent with a single-origin model therefore focused on archaic hominin admixture as an alternative model, by referring to additional (ghost) branches required to fit the data as archaic[2–7] and assuming (or inferring) deep divergences. These perspectives have oriented interpretations of both genomic (for example, selection[39]) and fossil (such as the evolution of early *H. sapiens*[40]) data. Here we have shown that a weakly structured stem model better captures the apparently inconsistent patterns of polymorphisms.

Preferring models of a weakly structured stem to archaic-admixture models has a range of implications. First, with a weakly structured stem, there is no need to posit that an archaic hominin population in Africa remained reproductively isolated from the ancestral human lineage for hundreds of thousands of years before the initiation of gene flow. Instead, there would simply have been continuous or recurrent contact between two or more groups present in Africa.

Second, there is evidence for both deleterious and adaptive archaic-hominin-derived alleles in contemporary genomes in the form of a depletion of Neanderthal ancestry in regulatory regions[41], or an increased frequency of archaic-hominin-related haplotypes such as at *EPAS1* among Tibetan people[42]. Under previous African archaic-hominin admixture models, the estimated 8–10% introgression rate is much higher than Neanderthal gene flow and would have plausibly been fertile ground for considerable selection for or against archaic-hominin-derived haplotypes[39]. By contrast, adaptation under a weakly structured stem would have occurred continuously over much longer periods. Polymorphism patterns that are inconsistent with the single-stem model predictions have been used to infer putative archaic admixed segments[3,7,17,39], negative selection against such segments[39] and pervasive positive selection[43]. However, such approaches are subject to large numbers of false positives in the presence of population structure with migration[41], and their interpretation should be re-examined in the light of a weakly structured stem model within Africa.

Third, multiple studies have shown a correspondence between phenotypic differentiation, usually assessed by measurements of the cranium, and genetic differentiation among human populations and between humans and Neanderthals[44–46] (see also Supplementary Information section 5.4). This correspondence potentially allows predictions of our model to be related to the fossil record. Some *H. sapiens* fossils, such as those from Iho Eleru in Nigeria (13 ka)[47], Ishango in the Democratic Republic of Congo (20–25 ka)[48] and Nazlet Khater in Egypt (35–40 ka)[49], have morphological features that may reflect recent gene flow from archaic hominins[47,48], and have been used in support of previously inferred archaic admixture scenarios[7,12,25]. The weakly structured stem model is not incompatible with archaic admixture having occurred in the ancestry of these fossils, but would imply, by contrast, that such individuals are unlikely to have contributed much ancestry to contemporary humans. The fossil record of Africa is sparse during the earlier time period of the stems (≥200 ka). Of the fossils that date to this period, some are fairly similar overall in morphology to contemporary humans (for example, Omo 1 from Omo Kibish in Ethiopia[50,51]), whereas others are similar in some morphological features to contemporary humans (for example, Irhoud 1 from Jebel Irhoud in Morocco[8,52]); others are different enough in morphology to have been assigned to species other than *H. sapiens* (for example, DH1 from Dinaledi in South Africa[53,54]). If, as our model predicts, the genetic differences between the stems were similar to those among contemporary human populations, the most morphologically divergent fossils are unlikely to represent branches that contributed appreciably to contemporary human ancestries.

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

## Methods

### Data and sequencing

We generated a sequencing dataset by combining existing and newly recruited populations who are now part of the African Diversity Reference Panel (ADRP)[20,22], as well as the 1000 Genomes Project (1KGP) populations[18]. These included the Amhara, Bakiga, Gumuz, Nama (newly generated), Oromo, Somali and Zulu populations from the ADRP and ESN, GWD, LWK, MSL, YRI, CEU, GBR, CHB and PJL from 1KGP (these groups are defined elsewhere[18]). After filtering for relatedness and retaining Nama individuals with more than 70% estimated Khoe-San ancestry, we focused on data from 289 individuals, including 44 Nama. These were merged with the high-coverage Neanderthal genome from Vindija Cave[23]. We kept variants from regions that fell within the 1KGP strict-callability mask, overlapped with at least 100 continuously called base pairs in the Neanderthal genome and were annotated as intergenic. ADMIXTURE and principal component analyses were done on a subset of variants filtered to remove variants in high linkage disequilibrium ($r^2$ threshold of 0.1). Additional details on the data and sequencing are available in Supplementary Information section 1.

### Linkage disequilibrium and diversity statistics

We used multi-population linkage disequilibrium and pairwise diversity statistics to fit parameterized demographic models to the data[2]. Unbiased linkage disequilibrium statistics were computed from all variants in retained intergenic regions[24], for pairs of variants separated by recombination distances $r = 5 \times 10^{-6} - 5 \times 10^{-3}$ Morgans. These were assigned to 16 recombination distance bins, and average statistics were computed within each bin (Supplementary Information section 2.2). Expected statistics under each model were computed in Moments, which also performed likelihood-based parameter optimization. The cSFS were computed conditioned on the Vindija Neanderthal carrying the derived allele relative to the ancestral allele determined by a six-primate alignment[18].

### Model specification and fitting

Model parameters include population sizes and size changes, split times, continuous migration rates and admixture times and proportions. The simplest model we tested was a bifurcating tree-like structure, allowing for subsequent migrations and recent known admixture events. To include ancestral population structure, we tested models that included multiple stem groups, each of which were allowed their own population size and could be connected by continuous migration. We tested multiple scenarios of early population structure, including long-lasting continuous migration between stem populations and scenarios of periods of isolation with subsequent merger events (Supplementary Information section 3). To avoid overfitting, we incrementally added complexity to our model optimization and we fixed a number of parameters that are constrained by historical records or are consistently estimated across multiple models and previous studies (Supplementary Information section 3.1). Likelihoods were computed using a composite multivariate Gaussian likelihood approach, and confidence intervals were estimated by refitting each model to 200 block-bootstrap replicate datasets (Supplementary Information section 3.2). We iteratively used gradient descent and L-BFGS-B optimization routines to fit each parameterized model (Supplementary Information section 3.3).

### Gene genealogy reconstruction

We used Relate[16] to reconstruct genome-wide gene genealogies from the focal populations in the merged ADRP and 1KGP datasets (Supplementary Information section 4). From reconstructed genealogies, we computed coalescence rates within and between populations, which provide an estimate for effective population sizes over time and the relative cross-coalescence rates between pairs of populations. To compare reconstructed genealogies from data to model predictions, we used msprime[55,56] to simulate genomic data for equal numbers of samples for each population in our inferred models. We then applied Relate to these simulated datasets using the same mutation and recombination rates and generation time.

### Reporting summary

Further information on research design is available in the Nature Portfolio Reporting Summary linked to this article.

### Data availability

Nama sequencing data are available from the European Genome-Phenome Archive (EGA), accession number EGAD00001006198. Data access is permitted for non-commercial, population origins or ancestry research upon application to the South African Data Access Committee with appropriate institutional review board approval. The African Diversity Reference Panel can be found at accession EGAS00001000960.

### Code availability

Code for the software used in this paper is found at the following locations: moments-LD (https://bitbucket.org/simongravel/moments), Demes (https://github.com/popsim-consortium/demes-python), Relate (https://myersgroup.github.io/relate/), msprime (https://github.com/tskit-dev/msprime) and tskit (https://github.com/tskit-dev/tskit).

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

**Acknowledgements** We thank participants for the DNA contributions that enabled this study; in particular, we wish to highlight the generous participation of the Richtersveld Nama community in South Africa and help from local research assistants W. De Klerk and H. Kaimann. Additional assistance and community engagement was conducted by J. Myrick, C. Gignoux, C. Uren and C. Werely. We thank the African Genome Diversity Project for data generation, including T. Carensten, D. Gurdasani and M. Sandhu; L. Anderson-Trocmé and G. Femerling for assistance in creating the map in Fig. 2 and Supplementary Fig. 2, respectively; and N. M. Morales-Garcia for data visualization discussion and designing Figs. 1 and 3. This research was supported by CIHR project grant 437576, Natural Sciences and Engineering Research Council of Canada (NSERC) grant RGPIN-2017-04816, the Canada Research Chair program to S.G. and the Canada Foundation for Innovation, and an NIH grant R35GM133531 to B.M.H.; and E.G.A. was supported by NIH K01 MH121659 and K12 GM102778. The content is solely the responsibility of the authors and does not necessarily represent the official views of the National Institutes of Health. M.M. and E.H. acknowledge the support of the DSI-NRF Centre of Excellence for Biomedical Tuberculosis Research, the South African Medical Research Council Centre for Tuberculosis Research, and the Division of Molecular Biology and Human Genetics at Stellenbosch University, Cape Town, South Africa.

**Author contributions** A.P.R., B.M.H. and S.G. designed the study. B.M.H., E.H. and M.M. designed recruitment protocols and recruited participants. B.M.H. and E.G.A. performed data quality control. A.P.R., B.M.H. and S.G. designed the statistical analyses. A.P.R. conducted the statistical analyses. A.P.R., T.D.W., B.M.H. and S.G. interpreted the results and wrote the first draft of the article. All authors read and edited the paper.

**Competing interests** The authors declare no competing interests.

**Additional information**
**Correspondence and requests for materials** should be addressed to Brenna M. Henn or Simon Gravel.

# Reporting Summary

## Statistics

For all statistical analyses, confirm that the following items are present in the figure legend, table legend, main text, or Methods section.

| n/a | Confirmed | |
|---|---|---|
| ☒ | ☐ | The exact sample size (*n*) for each experimental group/condition, given as a discrete number and unit of measurement |
| ☒ | ☐ | A statement on whether measurements were taken from distinct samples or whether the same sample was measured repeatedly |
| ☐ | ☒ | The statistical test(s) used AND whether they are one- or two-sided<br>*Only common tests should be described solely by name; describe more complex techniques in the Methods section.* |
| ☒ | ☐ | A description of all covariates tested |
| ☐ | ☒ | A description of any assumptions or corrections, such as tests of normality and adjustment for multiple comparisons |
| ☐ | ☒ | A full description of the statistical parameters including central tendency (e.g. means) or other basic estimates (e.g. regression coefficient) AND variation (e.g. standard deviation) or associated estimates of uncertainty (e.g. confidence intervals) |
| ☒ | ☐ | For null hypothesis testing, the test statistic (e.g. *F*, *t*, *r*) with confidence intervals, effect sizes, degrees of freedom and *P* value noted<br>*Give P values as exact values whenever suitable.* |
| ☒ | ☐ | For Bayesian analysis, information on the choice of priors and Markov chain Monte Carlo settings |
| ☒ | ☐ | For hierarchical and complex designs, identification of the appropriate level for tests and full reporting of outcomes |
| ☒ | ☐ | Estimates of effect sizes (e.g. Cohen's *d*, Pearson's *r*), indicating how they were calculated |

*Our web collection on statistics for biologists contains articles on many of the points above.*

## Software and code

Policy information about availability of computer code

| Data collection | No software was used to collect data for this paper. Software used to generate variant calls are described elsewhere in van Eeden et al (Genome Biology, 2022) and Gurdasani et al (Nature, 2015). |
|---|---|
| Data analysis | Code for the analysis software used in this paper is found at the following locations:<br>moments-LD (https://bitbucket.org/simongravel/moments),<br>Demes (https://github.com/popsim-consortium/demes-python),<br>Relate (https://myersgroup.github.io/relate/),<br>msprime (https://github.com/tskit-dev/msprime),<br>tskit (https://github.com/tskit-dev/tskit),<br>and custom scripts implementing analyses using each of these software are available at https://github.com/apragsdale/african-structure-paper. |

For manuscripts utilizing custom algorithms or software that are central to the research but not yet described in published literature, software must be made available to editors and reviewers. We strongly encourage code deposition in a community repository (e.g. GitHub). See the Nature Portfolio guidelines for submitting code & software for further information.

# Data

Policy information about availability of data

All manuscripts must include a data availability statement. This statement should provide the following information, where applicable:
- Accession codes, unique identifiers, or web links for publicly available datasets
- A description of any restrictions on data availability
- For clinical datasets or third party data, please ensure that the statement adheres to our policy

Nama sequencing data are available from the European Genome-Phenome Archive (EGA), accession number: EGAD00001006198. The African Diversity Reference Panel can be found at accession: EGAS00001000960.

# Human research participants

Policy information about studies involving human research participants and Sex and Gender in Research.

| | |
|---|---|
| Reporting on sex and gender | Adults over the age of 18 were invited to participate in the study. No preference was stated for male or female participants, and no individual was excluded on the basis of their sex. Results apply equally to both sexes. |
| Population characteristics | DNA samples were collected from three Nama communities in the Richtersveld region of South Africa, which borders southern Namibia. Members of the Nama community in the Richtersveld were initially approached regarding a genetic ancestry study in 2011. The Nama are a Khoekhoe speaking population who traditionally subsisted by sheep, goat and cattle pastoralism. Today, members of the community continue to be involved in pastoralism, as well as local mining, and tourism wage labor. All individuals self-identifying as Nama, Nama-Dama, or Coloured were encouraged to participate. No individual was excluded on the basis of ethnicity during recruitment. |
| Recruitment | Collection primarily occurred at home, by first approaching a family member, gauging interest in the study, entering the home or sitting outside by their invitation, oral and written consent occurring with family members present and then finally – completing the DNA sampling. By sampling individuals at home, members of the family are able to voice concerns, whether or not they decide to participate, and thus the final decisions are made in a slow and ethical manner. This process sometimes involved introducing the study and then returning at a later day in order to give participants sufficient time to consider. Families were approached based on a research assistant's determination that the family would be interested in participating, or via community meetings. |
| Ethics oversight | Written consent was recorded per our IRB protocol with human subjects approval from Stanford University (Protocol #13829), Stellenbosch University (N11/07/210) and later maintained via SUNY Stony Brook (Protocol #727494). |

Note that full information on the approval of the study protocol must also be provided in the manuscript.

# Field-specific reporting

Please select the one below that is the best fit for your research. If you are not sure, read the appropriate sections before making your selection.

☐ Life sciences   ☐ Behavioural & social sciences   ☒ Ecological, evolutionary & environmental sciences

For a reference copy of the document with all sections, see nature.com/documents/nr-reporting-summary-flat.pdf

# Ecological, evolutionary & environmental sciences study design

All studies must disclose on these points even when the disclosure is negative.

| | |
|---|---|
| Study description | We use low coverage whole genomes from several human populations to estimate demographic parameters associated with long-term evolutionary history. Such parameters include effective population size, divergence time, gene flow and population growth. The focus of the study is understanding the extent to which population structure exists in African populations during the origin of Homo sapiens. |
| Research sample | Low coverage (4-8x) Illumina short read data were generated for the Nama, Gumuz, Amhara and Oromo populations as part of the African Diversity Reference Panel (Sanger / Wellcome Trust). Data were jointly called with 1000 Genomes Project, and the GBR, YRI, MSL, TSI samples were parsed for various comparative analyses. |
| Sampling strategy | A sample size of ~100 was estimated for initial DNA collection in the Nama with the aim of capturing heterogeneity in ancestry across multiple villages and accounting for potential family members. For moments.LD analysis, we required a population to have a minimum of 25 individuals in order to capture variants at 2% or greater. |
| Data collection | Demographic and DNA collection was initiated in 2012 with a joint team of South African and American researchers which included both geneticists and anthropologists (BMH, MM, along with Christopher Gignoux, Caitlin Uren, Justin Myrick and Cedric Werely). |

Research assistants were fluent in Afrikaans, Nama and English (Hendrik Kaimann and Willem DeKlerk). Demographic data were recorded by pen and paper, then later transcribed into an Excel sheet (BMH, JWM).

| | |
|---|---|
| Timing and spatial scale | N/A |
| Data exclusions | 22 samples were excluded due to potential cross-contamination as detected by FREEMIX. |
| Reproducibility | N/A |
| Randomization | N/A |
| Blinding | N/A |

Did the study involve field work? ☒ Yes ☐ No

## Field work, collection and transport

| | |
|---|---|
| Field conditions | N/A |
| Location | Richtersveld, South Africa |
| Access & import/export | Data were exported with permission from the Department of Health, Republic of South Africa. Permit number: J1/2/4/2 NO 1/13 |
| Disturbance | N/A |

# Reporting for specific materials, systems and methods

We require information from authors about some types of materials, experimental systems and methods used in many studies. Here, indicate whether each material, system or method listed is relevant to your study. If you are not sure if a list item applies to your research, read the appropriate section before selecting a response.

## Materials & experimental systems

| n/a | Involved in the study |
|---|---|
| ☒ | ☐ Antibodies |
| ☒ | ☐ Eukaryotic cell lines |
| ☒ | ☐ Palaeontology and archaeology |
| ☒ | ☐ Animals and other organisms |
| ☒ | ☐ Clinical data |
| ☒ | ☐ Dual use research of concern |

## Methods

| n/a | Involved in the study |
|---|---|
| ☒ | ☐ ChIP-seq |
| ☒ | ☐ Flow cytometry |
| ☒ | ☐ MRI-based neuroimaging |

