## [Peer Review File · Nature]

Manuscript Title: A weakly structured stem for human origins in Africa

Reviewer Comments & Author Rebuttals

Reviewer Reports on the Initial Version:

Referee expertise:

Referee #1: evolutionary genetics

Referee #2: population genetics

Referee #3: human evolution

Referees' comments:

Referee #1 (Remarks to the Author):

The manuscript by Ragsdale et al investigates models of human origins in Africa. They make use of low coverage sequence data from 4 populations in Africa, representing western Africa, southern Africa and two populations from east Africa. They also include non-African comparative data from the 1000 genomes project and the Neanderthal and Denisovan genomes. In the study, they set-up several demographic models and use a maximum likelihood framework to see if the models fit the empirical data using two-locus summary statistics (which the authors published previously). These models represented various population splits, size changes, continuous and variable migration rates, and admixture events. They then use the fits of empirical data to these models to hypothesize about human population history over the past 1 M years. Their analyses show that contemporary human populations diverged from each other between 100k and 200k years ago. Before this however they found evidence of long periods of weak structure between two or more ancestral populations and that this “weak stem” was a more likely explanation of the deep population structure in Africa, rather than models of “archaic introgression”.

The article is certainly interesting; adding resolution to the question of human origins in Africa; evaluating specific models of population divergence and migration. However, it doesn't seem overly novel; they essentially consider a slightly broader class of models that are more inclusive of various forms of ancestral structure. They use this to reject models of ghost admixture and simpler IM / clean split models. It is methodologically interesting, applying the novel two-locus summary statistics with low-middle coverage genomes. It is possible that such statistics have extra information about deep human history. Overall the manuscript is well written and the model testing is a good step in the right direction of testing models of human origins in Africa. However, I still felt there is some room for improvement with regards to some additional explanation, clarification and discussion. I added detailed comments below, which the authors may consider to improve clarity and information in the manuscript.

Comments:

1. The use of low coverage sequence data should be motivated further and discussed more with regards to possible biases and limitations. For example, do the genomes have differences in coverage and how does this affect comparisons. For example, the authors mentioned: “Some statistics, such as $E[DiDj]$, require at least two diploid samples from a population to compute. Since we used a single Neanderthal sample, such statistics for the Neanderthal population were not used in the fit. By contrast, there are statistics that only require a single sample in a given population to estimate. These include cross-population nucleotide diversity measures, as well as some LD statistics involving more than one population.”

-- Since trustable diploid calls from low coverage data is difficult to obtain – how does this affect the results?

2. Also, the use of these specific low coverage genomes should be motivated in the light of high coverage genomes from the same areas/populations investigated and additional important areas that are available (from the HGDP project for example). For example, Rainforest Hunter gatherers (well represented by high coverage population level sequences in HGDP published by Bergstrom et al in 2020) - numerous studies showed RHGs to be an important early diverging lineage – it is strange that the authors did not include this important population dataset when datasets are available. The HGDP set also have a few high coverage Khoisan genomes which together with other published full genome datasets (Schlebusch et al 2020) could have provided a less admixed high coverage Khoisan dataset, compared to low coverage admixed Nama used here

3. The authors mentioned: “We also included the British (GBR) from the 1000 Genomes Project in our demographic models as a representative source of back-to-Africa gene flow and recent colonial admixture in South Africa. –The Nama have been shown in previous studies to also have East African and West African admixture - was this considered?”

4. The authors tested two recombination maps for influences of different recombination rates between populations on estimates. Their LD statistics that they strongly rely on for inference are presumably very dependent on the recombination maps used. I was unsure whether HapMapII and OMNI YRI are really reliable given the breadth of African diversity and depths of time under study. The authors should discuss this more. Also – is the HapMapII map that they use, the YRI hapmapII map or the combined HapMapII recombination map? If both maps are based on YRI is is even more worrisome as other population recombination rates are not represented.

5. Regarding the following sections:

“Finally, we used a high-coverage ancient Neanderthal genome from Vindija Cave, Croatia²³ to account for gene flow from Neanderthals into non-Africans and gauge the relative time depth of divergence, assuming Neanderthals diverged 550ka from a common stem.”

and

“The conditional site frequency spectrum (or cSFS) is the distribution of allele frequencies restricted to loci that satisfy a given condition. Specifically, we consider the distribution of allele frequencies in presentday populations conditioned on the Vindija Neanderthal carrying the derived allele relative to the inferred ancestral allele.”

--Gene flow from neandethal into Africans have been shown in recent times (Chen et al 2020). The

authors should discuss how this will affect their results. Also, with regards to the two quoted sections above

6. Regarding this section

“The earliest divergence among contemporary human populations differentiates the southern African Nama from other African groups between 110–135ka, with low to moderate levels of subsequent gene flow (Table 1). In none of the high-likelihood models which we explored did the divergence between Nama and other populations exceed ~140ka.”

Are the authors sure the significant recent geneflow that the Nama received does not influence estimates? There have been several previous estimates (not based on cross-coalescence methods that the authors indeed discussed) that have estimated the divergence of the Khoisan to over 200KYA when accounting for different mutation rates (Gronau et al 2011, Veeramah et al 2012 and Schlebusch et al 2020). The authors fail to comment on these studies and estimates. Even the Nama group split specifically (Schlebusch et al 2020) was estimated before to much deeper times. The author fails to comment on this – even when it was on the same population group

7. Connecting to the previous comment.

The authors only comment on previous cross coalescent population divergence estimates:

“Published estimates of the earliest human divergences with rCCR, which range from 150ka-100ka²⁹, may be significantly biased when compared to more complex models with gene flow as inferred here. We find that midpoint estimates of rCCR are poor estimates for population divergence, often underestimating divergence time by 50% or greater (e.g., Mende vs. Gumuz ~15ka compared to a true divergence of 60ka), and recent migration can lead to the misordering of divergence events (Figure 5E). We suggest that rCCR analyses which do not fit multiple parameters including gene flow should be interpreted with caution.”

What about other studies and other estimations of population splits that have been done before?

These split times, corrected for mutation rates were nicely summarized by Bergstrom et al 2021 Nature – in their Figure 2c. All the non rCCR estimates were much deeper estimates and the authors should comment on this in light of their own shallower split-times.

8. Still connecting to the previous two comments

I'm very surprised that the authors consistently get such low divergence time estimates. Their simulations do assume recent mutation and recombination rates, yet perhaps they are not sufficiently taking into account recent gene flow among contemporary populations? For instance, their claim that the earliest population divergence among contemporary populations occurs 120-135ka and that simple "model misspecification" explains variation in previous divergence time estimates.

This doesn't seem right, because the kinds of bias we would expect under such simple models is opposite to what they suggest. For example, a clean split model that assumes no post-split gene flow will give more recent divergences than reality. So, they can't use that as an explanation for why those simple models give much older split times.

The authors might then argue that ancestral structure would bias such simple models in this way -- However the cross coalescence analyses that the authors do have a drop off after 300k that indicate a bottleneck. Inversely in their weak stem model should result in an increase in N_e here but there is a drop in CC. The relate IICR curves show a large reduction in N_e >300ka. This pattern in the IICR

curves is something the authors seem to avoid addressing, instead very generally saying that between 100-1000ka ago there was a large increase in N_e . The weak stem model they suggest should give rise to high N_e 's while they suggest population structure exists (100-1000ka), but the curves contradict this.

So, I am left curious as to how and why they consistently get such low divergence time estimates.

9. The fact that the authors fix many of the parameters (e.g. Out-of-Africa at 50ka – which seems quite recent!) to gain resolution on deeper events, could potentially be problematic. The authors should discuss their fixed parameters more and how much it is influencing the other estimates

10. I was not sure which model that the authors tested correspond to Figure 1 D-multiregional. It just seemed that they tested the single origin models and different versions of the weak stem model. They authors should more clearly link their models that they tested with the models shown in Figure 1.

11. The authors tend to do self-referencing and, in some cases, prefer referencing their own findings rather than original papers that showed specific findings. They also do not reference papers that previously demonstrated many of their findings (Back-to-Africa gene flow, West African population expansion associated with agriculture, East Africans into Nama ~25%, 15% colonial admixture into Nama etc).

12. Regarding this sentence:

“We observe significant gene flow from the Amhara and Oromo into the Nama, a signal which is likely a proxy for the movement of eastern African caprid and cattle pastoralists 25,26, here estimated to constitute a 25% ancestry contribution 2,000 ya.”

The reader might wonder why this is not seen as such in the ADMIXTURE analyses included in Fig 2 D – where the admixture just looks like European admixture – the authors should comment on it and explain it in the text.

13. Regarding this section:

“Following these merger events, the stems subsequently fracture into subpopulations which then appear to persist over the past ~ 120ka. These...”

We know that these populations were most probably connected and not fractured into sub populations to up until the Bantu-expansion- there is archeological evidence of hunter-gatherers living all across the continent with no gaps in connectedness. Then with the Bantu-expansion the HG groups were fractured and replaced and only a few groups remain. Where would these “in-between” groups fit in the model? The population “fracturing” might just be a sampling bias on a previous gradient of connectedness. Sampling specific points in a “isolation-by-distance” space would look like a diverging model with “fracturing into subpopulations” . The authors should discuss this more in terms of how the bantu-expansion influenced the fracturing of groups and how the landscape of geneflow might be explained before this period

14. Regarding this sentence:

“By contrast, an archaic hominin population in Africa would need to have stayed in relative

reproductive isolation from the ancestral human lineage over hundreds of thousands of years despite closer geographic proximity and reproductive compatibility.”

The authors used this statement to argue that their weak stem model are more likely than archaic admixture models and that it was unlikely for these very different populations to co-exist in the same geographic space.

This however apparently happened with very solid evidence of the Homo naledi, Kabwe/Broken Hill (Grun et al 2020, nature) fossil remains. The authors commented on H. naledi, which most likely was to archaic or differentiated to have mixed with our lineage. But what about Kabwe that was possibly closer to our lineage and coexisted with us to relatively recent times.

15. At one point in the main text the authors say they cannot assign any of the inferred stem groups to be the direct ancestor of any contemporary populations (the weak stem model has gene flow between stem populations for a long time, meaning only 1-4% genetic differentiation in modern populations arose as drift in this period of population structure). But then they later (tentatively) suggest Stem 1S = southern Africa, Stem1E = Eastern Africa, and Stem 2 = Western or Central Africa. I think the authors should be very careful not to overinterpret results based on the model with highest likelihood. They themselves offer the caveat that this study does not reject more complex scenarios with more stem populations or hybrid models including ancestral structure and archaic admixture, thus they should be careful to assign geographic locations to hypothetical inferred stem populations.

16. The figures in the Supplement are quite difficult to interpret. What the various panels mean, and specifically what "statistics" are represented on the Y-axis. Even after reading the paper I am unsure of the specific diversity-based statistics that were used in ML estimation of parameters of models.

17. It is not clear whether authors had permission and co-working of relative community councils regarding the Khoisan data

Referee #2 (Remarks to the Author):

The manuscript by Ragsdale and colleagues reports an investigation into the origin of population structure in the early human lineage based on a comparison of the performance of four competing demographic models in explaining the genomic diversity of 290 whole-genome sequences.

The best-fitting model identified in this study suggests an early divergence of two stem populations with continuous gene flow between them. It further suggests that differential merging of these stem lineages (probably in different parts of the continent) could have resulted in the geographically stratified population structure that is observed among the contemporary African populations. The model further estimates the structure observed in the contemporary African populations to have emerged in the last 150 Kya. The results challenge the popular notions that archaic admixture (similar to Neanderthal and Denisovan in non-Africans) has contributed substantially to the genomes of current-day Africans.

The manuscript deals with one of the difficult and debated questions in human evolution and provides a novel perspective of early human history. Given, the novelty and immense importance of these insights, it is critical to verify whether the dataset used is optimal for testing these complex demographic models, whether the models give similar predictions for a range of datasets and parameters and whether the analyses are extensive and rigorous.

I have the following major recommendations :

1. As the confluence of ancestries in a population has the potential to influence the fitting of demographic models, the choice of populations (and ancestries represented in them) assumes critical importance in a study like this.

The ADMIXTURE plots show three of the populations tested (Nama, Oromo and Somali) to have substantial gene flow from the Out of Africa group (GBR). Moreover, some gene flow within the three East African groups and also from Bantu-speaker to some of the East and South African populations is plausible. In addition, gene flow from East Africa to Nama in the last two millennia is well known and has been reiterated by this study. Given this multi-level interconnectivity and gene flow between populations, some of which are too subtle to be visualized by ADMIXTURE, it is important to assess the impact of these gene flow events on the fitting of the demographic models. I would recommend the following additional analyses/iterations to improve the robustness with respect to admixture -

i. Replace the Nama with "less admixed" Khoesan genomes from Schlebusch et al. 2020 and test whether the model fitting is impacted by the Eurasian admixture in Nama. In case this is not feasible at least the Khoesan ancestry cut-off for inclusion should be increased from 70% to perhaps 80 or 90%. and then the results are compared.

ii. An iteration where a Chinese/Japanese population is used instead of the British (as there is no major gene flow from these groups into Africa) to represent out of Africa population could increase the confidence in the results.

iii. I am guessing that to reduce the impact of the Bantu expansions on the model fitting, the authors have included MSL instead the more common West African proxy - YRI. However, an iteration where YRI or ESN replaces MSL could also be used to assess the extent to which the inclusion of Niger-Kordofanian ancestry might influence the fitting of these models.

iv. Probably due to scarcity of population-scale WGS data the Central African forager-related ancestry was not included. However, this is a major limitation as one of the key components of African diversity is completely missing in this study. Could the authors include some additional analyses that included Central African foragers (maybe from Simons Genome or HGDP dataset) to check whether and how they fit into the model. Although due to the smaller sample size the results won't be as reliable it would still be good for the readers to know.

v. Similar to the approach used for the single Neanderthal genomes some of the African ancient

genomes from the recent Lipson et al. study could also be tested with these models.

vi. Also, the sample sizes vary widely among the 6 groups, ranging from 30 to 100 individuals. Could the authors comment on whether this could have influenced the results to some extent and if possible provide some supplementary data to support their comment?

2. The authors have included several dates from literature (such as those for divergence between western and eastern African populations, Out of Africa migration) as fixed parameters in the model testing. However, some of these dates have a rather wide range while others might get challenged/changed in near future. Therefore, it would be interesting to rerun the models with alternative dates (earlier /later) for these events and provide an estimate of the extent of variations in final results with changes in the fixed parameters. For instance, if Out of Africa is dated at 70 Kya instead of 50 Kya, what would the rest of the divergence estimates look like?

Although authors provide some examples such as the one for population size expansion in the MSL that corresponds to archaeological records of agricultural innovations, they are all from a relatively recent time scale. Therefore, I would recommend also a set of analyses where a few of the key dates such as the split between East and West, OoA, etc are iteratively withheld, and the models are forced to estimate them. This could provide an assessment of how well estimates generated by the best fitting models concur with the estimates that were used as fixed parameters.

I fully understand that implementing the alternative analyses suggested in #1 and #2 above might require substantial time and effort. However, as the results presented here have the potential to lead to a paradigm shift in the field, it is very important to ensure that they are based on rigorous testing and evaluation.

3. The inclusion of WGS from over 80 Nama participants is definitely a highlight of this study.

Due to controversies in the recent past, a San Council has been set up in South Africa for assessing the potential impact and approving research (including genetic studies) on the Khoe and San people (including Nama) from the country. It is important to know if the authors have approached the council and applied for its approval for the study? I would strongly recommend them to get this research approved by the council.

In case this is not feasible, the authors should provide a clear rationale for not being able to do so. Also, a detailed description of the ethical considerations and community engagement steps involved prior to the study needs to be added to the supplementary information. Given that the study is based on a marginalized and previously exploited group, it is critical to ensure that ethical considerations and community engagement has been adequately nuanced and sensitive.

Further details such as whether the data was collected for this particular study? whether informed consent allowed for secondary use? if yes, what kind of studies are permissible? needs to be included in the Supplementary Information.

Minor recommendations :

1. The ADMIXTURE plot is presented at K=4. Was this the optimal K based on CV scores?. Also, why the Khoesan ancestry partitioning was done at K=6 (Section 1.1 S Info) and not K=4? I would recommend that the authors should add a full ADMIXTURE plot starting from K=2 to the best value of K in a supplementary figure?
2. Supp Info page 2 last line. Please check the spelling " Neanderhal"?
2. The same dataset has been referred to as ADRP (African Diversity Reference Panels) and AGRP (legend S Figure 18, penultimate line on Page 11). The authors should use one name consistently. Also, the original study by Gurdasani et al. 2015 was named African Genome Variation Project (AGVP) which is different from both the names used here.
3. Some more details such as - the number of SNPs in the two datasets, the size of the final dataset after merging, and also after genic SNP removal will be helpful.
4. The authors have excluded all the genic SNPs in their analysis. It would be good to know if the inclusion of these SNPs would have changed the results substantially.
5. The order in which the figures are referred to in the text needs fixing. Figure 2C is cited before 2A and B. Similarly Figure 4 seems to be referred to before Figure 3.
6. What are the horizontal green lines in Figure 5 show? Needs to be described in the legend.
7. As most of the methodological details of the study have been provided in a huge supplementary information file, it is difficult to relate the analyses mentioned in the text to a particular section of the SI file. It would be more convenient if the authors refer to a specific section of the SI instead of just saying SI in the text.

Referee #3 (Remarks to the Author):

Dear authors,

I found your paper really interesting, and a significant contribution to the current debate on the pattern and process of modern human evolution in Africa. You set out to formally test mathematically a series of complex models of evolution, something that is clearly needed. Like all models, those proposed are simplifications and accommodations of reality, but the results should contribute to moving the discussions forward. I have a few issues that I think you should explain further and/or consider, and a few suggestions at the end.

Marta

The paper obtains (in my view) five key results:

1. Single origin w/o introgr but w/ migration: poor fit
2. Living African populations diversified 110-135ka (Nama first split), w/ low to moderate gene flow afterwards
3. Some structure BEFORE differentiation of current populations
4. Two models with highest likelihoods: A: "continuous-migration" (LL = -115, 500); B: "multiple-merger" (LL = -102, 600)
 - With continuous migration between stems, population structure extends back to 1.1–1.4Ma
 - In both models, the branch ancestral to the Nama shares a common ancestral population with the other African groups ~120–135ka.
 - only 1% to 4% of genetic differentiation among contemporary populations can be traced back to this early population structure
5. Best fit model: "Multiple-merger"
 - Stems 1 & 2: continuous migration until ~550ka (Neanderthal split from Stem 1)
 - sharp bottleneck in Stem 1 (down to $N_e = 117$) after the split of the Neanderthal branch.
 - Stem 1 fractures into "Stem 1E" and "Stem 1S" 479ka, followed by independent evolution.
 - ~120ka: merging of Stem 1S (29%) and Stem 2 (71%) to form the ancestors of the Nama
 - ~100ka: merging of Stem 1E (50%) and Stem 2 (50%) to form the ancestors of the W & E Africans
 - Further gene flow from Stem 2 (18%) to W Africans 25ka

I think these results warrant the authors' conclusions that (a) single origin models without some form of multiple stem contribution prior to modern human differentiation are not likely; (b) that all living humans share a common African ancestral population 110-135Ka during MIS 5 from which they subsequently diversified; and (c) that despite the fact that discriminating amongst multiple configurations of what happened prior to that ancestral MIS 5 population is not simple, likelihoods can be determined hierarchically, and the best-fit model (amongst those tested) is one in which the population ancestral to all modern humans results from the merger of two lineages (contributing differently to different current regional groups) which originally experienced gene flow between them, but which also had a longish period of drift. The data used in the paper represent mostly a new genomic dataset that aims at better representation of diversity amongst living African peoples, providing new and better insights into population history. The data are presented clearly, although some of the figures & graphs could do with fuller captions. The paper uses a wide range of mathematical and statistical tests. These are partly explained in the main text, and partly in the supplementaries, although for the non-VERY-specialist, some of these explanations are not easy to follow. I believe the analyses appropriately test the models set-out by the authors, but as I am not an expert, I cannot comment further.

From the above, it should be clear that I think this is a good paper that makes an important contribution, but I do have some issues with some of the interpretations and contextualisation of the results and conclusions:

1. regarding the fixed parameters, it is not clear how these were set (why, for example, the out-of-Africa at 50ka, or the Neanderthal introgression into Eurasians at 45ka?), or what impact it would have on the models if these were different. I think the manuscript would benefit from a bit more clarity on this.

2. Also, you say “These constraints allowed us to integrate information from previous genetic and archaeological research to infer robust migration rates. For example, all models infer relatively high gene flow between eastern and western Africa ($m \approx 2 \times 10^{-4}$, the proportion of migrant lineages per generation)” (pgs 3/4). But why relatively high gene flow between eastern and western Africa? This is not supported by your Admixture plot results – these show that (unsurprisingly) eastern African farmers, such as the Luyha and Bakiga, are an unequal combination (merger in your terms!) of western African Bantu and local eastern African groups. But this happened in East Africa, and there’s minimal evidence of E African gene impact in Western African Bantu. In fact, there is the same process with the Zulu – a ‘merger’ of Western African Bantu and local Khoisan genes – and yet this is not described and/or computed as relatively high gene flow between southern and western Africa. Can you please address this?

3. In pgs 4/5, when describing the two most likely models, you say “Both allow for migration between stem branches, but differ primarily in the timing of the early divergence of stem populations and their relative N_e (Figure 3).”

a. The parameters of models A and B discussed in the main text presumably correspond to those on Tables S4 and S6 respectively? Can this please be made clear? Assuming that is the case, can the standard error of the timing of Stems 1/2 divergence (1,163ka in A, and 1,442ka in B), and any other divergence date mentioned in the main text, be included please? These are very substantial – A: $1,163,072 \pm 390,803$ and B: $1,442,022 \pm 426,449$;

b. This is my ignorance, but why are the standard errors of the N_e estimate of models S4 and S6 so much larger than those of the other 3 models?

4. Table 1, Model B, lists the migration between Stems 1 & 2 as occurring for a period of 963Ka – this corresponds to the period between Tstems (1,442ka) to the fracture of stem 1 into 1E and 1S (479ka) – does this imply that stem1 and stem 2 were still in continuous genetic contact at the time of the Neanderthal split (550ka) and subsequent drastic bottleneck of remaining African population? I find it hard to conceive demographically and geographically.

5. I have an issue with the geography of stem populations:

a. in pg. 6, you say “Mende receive a large additional pulse of gene flow from Stem 2, replacing 18% of their population 25ka.... This may indicate that an ancestral Stem 2 population occupied western or Central Africa, broadly speaking. The differing proportions in the Nama and eastern Africans may also indicate geographic separation of Stem 1S in southern Africa and Stem 1E in eastern Africa”,

i. are these geographic associations really warranted? Can you exclude a model in which all this is happening East Africa, for example Ethiopia and Tanzania, and pan-Africanness of sapiens is the result of expansion in MIS5?

b. again in the discussion: “During the Middle Stone Age, the multiple merger model indicates three major stem lineages in Africa, tentatively assigned to southern (Stem 1S), eastern (Stem 1E) and western/central Africa (Stem 2).”

i. Besides the fact that, according to the models these stem populations pre-date the MSA (see my comment 6 below), again, attributing geography on the basis of where populations with different genetic proportions are today is not justified

c. Finally, the section about the ecological riddle “They also help explain an ecological riddle posed by the archaic hominin admixture model. Neanderthal populations were separated from early Homo

sapiens by thousands of kilometers and continental geographic barriers. By contrast, an archaic hominin population in Africa would need to have stayed in relative reproductive isolation from the ancestral human lineage over hundreds of thousands of years despite closer geographic proximity and reproductive compatibility”.

i. I have a real problem with this. Let’s consider for a moment the thousands of km separating Africans from Neanderthals, picking randomly on Nairobi as a point of reference:

1. Nairobi – Cape Town: 5,147.5 km
2. Nairobi – Tel Aviv: 5,434.3 km
3. Nairobi – London: 6,817.7 km
4. Nairobi – Dakar: 8,204.0 km
5. Etc

Given that the distance between East and West Africa is substantially larger than between East Africa and northern Europe, never mind the Levant, and that the ecological barriers between regions of Africa are sufficient to generate biogeographical structure in other mammals, this paragraph does not make sense. On the contrary, if the stem populations were indeed distributed all over Africa, how can hominin diversity 300-200ka be explained? While some geographic structure probably explains the long-standing genetic identity between the stem populations, I believe that defining that geography without ancient genomes can be very misleading and not consistent with other lines of evidence.

6. My last substantive issue. I found it difficult to assess how meaningful the differences in likelihood are.

- a. S2 – single origin model: LL = -189,434
- b. S3 – continuous migr, w/o stem migr LL = -126,644
- c. S4 – continuous migration LL = -115,500
- d. S5 – merger w/o stem migration LL = -107,652
- e. S6 – merger w/ stem migration LL = -102,633

How significant are these differences? In terms of population history, the difference between S5 and S6 is huge, for example. At the end, the models have, by necessity, much uncertainty in the many parameters fixed and estimated, and this is in stark contrast to the certainty/absolute value of the likelihood measure that determines the “best history” amongst clearly many possible histories. I know this is in the nature of testing models mathematically, but perhaps you may consider making it clear that these are hypotheses and not past realities.

Suggested improvements

Besides the above more substantive issues, I have some suggestions that the authors may want to consider:

1. The second paragraph of the introduction is great, and clearly sets out the problems with previous models/studies and the point of the paper. But the 1st paragraph is really not inspiring....
 - a. Why are MSA sites significant?
 - b. The youngest MSA is ~11ka, not 40ka
 - c. A few (very few!) MSA sites date to 315-130ka, so not equally distributed in time-space
 - d. Fossils with sapiens traits are not SIMILARLY distributed across the continent – Jebel Irhoud is twice as old as Herto and there are many fossils with no sapiens-derived traits 300ka

e. Why are the Kibish fossils not included in the list?

f. Unless there is an ancient genome to prove otherwise, all fossils are likely to be dead-ends... Clearly, this par is trying to give some palaeo-context to the study, but it definitely needs a bit of work

2. In pg. 3, you say “We conclude that geographic patterns of contemporary Homo sapiens population structure date back to the late Middle Stone Age in Africa, likely arising during MIS 5.” MIS 5 is not the ‘late’ MSA, but the main period of MSA expansion half-way through the history of the industry

3. In the same paragraph, you say “Although we find evidence for earlier population structure in Africa (see below), contemporary populations cannot be easily mapped onto the more ancient ‘stem’ groups as only a small proportion of drift between contemporary populations can be attributed to drift between stems (Figures 4 and S10–S13)” – I think this could do with a bit more explaining; and S10 & S11 suggest that there’s virtually no drift shared with stems?

4. Table 1 of main text, parameter ‘c’ of merger-model: I think the divergence time be ~95ka instead of ~98ka

5. You end that paragraph in pg 4/5 saying “The two models also differ in the mode of divergence during the Middle Stone Age”. Yet, I don’t see how the models’ estimates can be related to historical events during the last 300ka – Model A has continuous migration between 2 pops (one of which is ancestral to sapiens and Neanderthals) for ~900 ka, decreasing after MIS 5; Model B has continuous migration between 2 pops until ~ 550ka, then this stops, and pop 1 eventually contributes differentially to the Nama and the ancestors of other Africans after MIS 5, and pop 2 gives rise to a second human ancestral lineage and to Neanderthal, the human pop 2 nearly disappears, then splits into 2 ~479ka, 2a + 2b pops evolve independently until they meet/merge variously with descendants of pop 1 during MIS 5. In fact, both models have populations coming together to form the ancestry of living Africans during MIS 5, so do not differ in the mode of divergence during the MSA specifically. Why not refer to the Middle Pleistocene (780-126 ka) ? The MSA is not a period, but an industry with variable spatial and temporal distribution.

6. Finally, I found the temporal scale of Figure 4 very misleading – it hides the actual scale of the population processes being described.

Author Rebuttals to Initial Comments:

We thank all three reviewers for their supportive and helpful comments. We ran a number of new analyses to address the different points raised by the reviewers, and expanded the main text and supplement to clarify our hypotheses and methods. We also provided a new section and supplementary material explaining the community engagement and consent processes. Detailed changes are described below. We think that these modifications strengthen our conclusions and help clarify the manuscript and its relationship with previous analyses.

Referees' comments:

Referee #1 (Remarks to the Author):

The manuscript by Ragsdale et al investigates models of human origins in Africa. They make use of low coverage sequence data from 4 populations in Africa, representing western Africa, southern Africa and two populations from east Africa. They also include non-African comparative data from the 1000 genomes project and the Neanderthal and Denisovan genomes. In the study, they set-up several demographic models and use a maximum likelihood framework to see if the models fit the empirical data using two-locus summary statistics (which the authors published previously). These models represented various population splits, size changes, continuous and variable migration rates, and admixture events. They then use the fits of empirical data to these models to hypothesize about human population history over the past 1 M years. Their analyses show that contemporary human populations diverged from each other between 100k and 200k years ago. Before this however they found evidence of long periods of weak structure between two or more ancestral populations and that this “weak stem” was a more likely explanation of the deep population structure in Africa, rather than models of “archaic introgression”.

The article is certainly interesting; adding resolution to the question of human origins in Africa; evaluating specific models of population divergence and migration. However, it doesn't seem overly novel; they essentially consider a slightly broader class of models that are more inclusive of various forms of ancestral structure. They use this to reject models of ghost admixture and simpler IM / clean split models. It is methodologically interesting, applying the novel two-locus summary statistics with low-middle coverage genomes. It is possible that such statistics have extra information about deep human history. Overall the manuscript is well written and the model testing is a good step in the right direction of testing models of human origins in Africa. However, I still felt there is some room for improvement with regards to some additional explanation, clarification and discussion. I added detailed comments below, which the authors may consider to improve clarity and information in the manuscript.

We thank the reviewer for the supportive comments.

Comments:

1. The use of low coverage sequence data should be motivated further and discussed more with regards to possible biases and limitations. For example, do the genomes have differences in coverage and how does this affect comparisons. For example, the authors mentioned:

“Some statistics, such as $E[D_{ij}]$, require at least two diploid samples from a population to compute. Since we used a single Neanderthal sample, such statistics for the Neanderthal population were not used in the fit. By contrast, there are statistics that only require a single sample in a given population to estimate. These include cross-population nucleotide diversity measures, as well as some LD statistics involving more than one population.”

-- Since trustable diploid calls from low coverage data is difficult to obtain – how does this affect the results?

In the paper introducing this inference approach (Ragsdale & Gravel, 2019, Plos Genetics), we performed validation of the approach using jointly-called low-coverage data. Fig A6 C-F in that reference shows that LD statistics computed for the same set of individuals using high and low coverage are almost identical, which is in contrast to the distribution of allele frequencies.

Figure A6: Effect of mutation types and low coverage on Hill-Robertson statistics. We used 40 individuals that overlapped between the 1000 Genomes data and the 90 Han Chinese data to compute (A,C) σ_d^2 , (B,D) $E[Dz]/E[\pi_2]$ and (E) the folded AFS across intergenic sites. (A-B) In our analyses in the main text, we used all mutations (transitions and transversions). Here, we compare LD curves for statistics estimated from transitions (solid, blue) or transversions (dashed, orange) only. Differences in statistics between the two mutation types are negligible. (C-D) The 90 Han Chinese data was high coverage, while 1000 Genomes data (which we used in our analysis) was low coverage. The LD curves are largely unaffected by the level of low coverage in the 1000 Genomes data. (E-F) For comparison, the allele frequency spectrum is sensitive to coverage, as the singleton bin of the AFS is significantly underestimated in the 1000 Genomes data (F).

Following reviewer comments, we further validated this by contrasting the 1000 Genomes high-coverage and low-coverage datasets in two-population models. For each dataset, we selected pairs of populations and computed the full set of two-locus statistics. We then performed maximum-likelihood parameter inference for two-population models. As in Ragsdale & Gravel (2019), we found a very modest difference between statistics computed using high-coverage data and statistics computed using low-coverage data. The resulting inferences were also very consistent such that we feel confident our results are not biased by coverage.

Figure 2: Demographic models inferred using two-locus statistics using data for the same populations, generated using high-coverage data (left) and jointly-called low-coverage data (right).

2. Also, the use of these specific low coverage genomes should be motivated in the light of high coverage genomes from the same areas/populations investigated and additional important areas that are available (from the HGDP project for example). For example, Rainforest Hunter gatherers (well represented by high coverage population level sequences in HGDP published by Bergstrom et al in 2020) - numerous studies showed RHGs to be an important early diverging lineage – it is strange that the authors did not include this important population dataset when datasets are available. The HGDP set also have a few high coverage Khoisan genomes which together with other published full genome datasets (Schlebusch et al 2020) could have provided a less admixed high coverage Khoisan dataset, compared to low coverage admixed Nama used here.

Even though we agree that these different datasets are relevant to the questions discussed here, our choices were dictated by a few considerations.

First, our choice of populations was designed to improve our understanding of early human demography in Africa. The Nama derive ancestry from a deeply diverged population branch, and thus their demographic history is of unique interest. Given the archaeological record as well as previous genetics work, we expected that a minimal model for the Nama must include pastoralist East African groups that contributed to the Nama, as well as European colonial admixture. The inclusion of 3 Ethiopian populations here represent some uncertainty in the possible source population for the pastoralist contribution to the Nama; the Amhara/Oromo also have substantive ‘back-to-Africa’ ancestry. The Gumuz serves as a representative of southwest Ethiopian ancestry which pre-dates any Holocene back-to-Africa migration. While we completely agree that it is of great interest to add additional populations to this model (and the reviewers have made many good suggestions), there are important technical considerations that led us to believe that this was best left for future work. We have clarified our motivation for this particular choice of populations in the manuscript, supp section 1.3.

Among the technical limitations, we wanted to have >20 of individuals per population, to ensure accurate and robust estimation of LD statistics. This ruled out the HGDP and Schlebusch et al. 2020. The Neanderthal was the one exception we made, since that sample was important to narrow down the relationship between Neanderthal and the early branches. However, the small sample size in Neanderthal forced us to rely only on a small subset of our statistics and fix some parameters (Supplementary section 2.3). In other words, small sample sizes lead to much more poorly constrained models.

Second, we wanted to be able to perform joint calling and quality control of all our samples, as our experience has shown that mixing datasets after calling greatly increases the risk of batch effects. In reply to reviewer comments, we directly tested whether combining data from a high-coverage population and a low-coverage population could result in biases. Above, we have shown that either the high- or the low-coverage 1000G datasets gave consistent estimates. However, mixing low- and high-coverage datasets (where variant calling was performed separately) leads to inferences that were discordant:

Figure 3: Demographic models inferred using two-locus statistics using data for the same populations, generated using high-coverage data in one population and low-coverage data in the other.

As might be expected, batch effects in post-hoc merged datasets increase the apparent divergence time between populations (details in Supplementary section 7.5). While these discrepancies might possibly be alleviated by additional post-hoc quality control, we believe that the most robust approach is to jointly call and QC data from all populations.

Unfortunately, performing joint calling with all the populations suggested by the reviewers is not possible. In some cases, the raw data (fastq) is simply not accessible to us. For example, the Schlebusch 2020 paper analyzed merged VCFs rather than re-calling the data jointly, presumably for the same reason. Even if it were feasible, the data acquisition, computational and analytical effort involved in jointly re-calling all these data is overwhelming.

We do believe that the questions raised by the reviewers are of great interest, and we are indeed preparing a (jointly-called!) high-coverage dataset with additional genomes from African populations, such as Central African foragers, that will be ideal for addressing these questions. However, the generation and interpretation of these data represents years of work by an international team and therefore is better left for a future publication.

Regarding admixture in Khoe-San: our inferred models do account for recent admixture, so that it is not crucial to obtain less-admixed samples. However, to further ensure robustness, we repeated the analysis twice removing individuals with low ADMIXTURE-inferred Nama ancestry. We imposed a >90% and >99% of modal component for the Nama, and found reassuringly that the inferred models were entirely consistent, with the only difference being the inferred recent admixture proportion into the Nama (see section 6.1.2 and 7.1.3).

3. *The authors mentioned: “We also included the British (GBR) from the 1000 Genomes Project in our demographic models as a representative source of back-to-Africa gene flow and recent colonial admixture in South Africa. –The Nama have been shown in previous studies to also have East African and West African admixture - was this considered?”*

The reviewer is correct, and this was indeed considered in the analysis. Populations in the model include representatives of East African (Gumuz, Amhara/Oromo) and West African ancestry, and the best-fitting model includes migrations from these two sources into Southern Africans. Please see Figure 3 with the teal and blue arrows, as well as $f_{GBR \rightarrow Nama}$ and $f_{EP \rightarrow Nama}$ pulses in the supplementary tables S2-S6.

4. *The authors tested two recombination maps for influences of different recombination rates between populations on estimates. Their LD statistics that they strongly rely on for inference are presumably very dependent on the recombination maps used. I was unsure whether HapMapII and OMNI YRI are really reliable given the breadth of African diversity and depths of time under study. The authors should discuss this more. Also – is the HapMapII map that they use, the YRI hapmapII map or the combined HapMapII recombination map? If both maps are based on YRI is is even more worrisome as other population recombination rates are not represented.*

We used the combined HapMap II map and the YRI map to show that the choice of maps had little effect on the statistics. In addition, we have now included a reanalysis with a Nama-specific recombination map. We find that the results are broadly unchanged, see Sections 6.1.4 and 7.1.2 in the Supplement.

5. *Regarding the following sections:*

“Finally, we used a high-coverage ancient Neanderthal genome from Vindija Cave, Croatia²³ to account for gene flow from Neanderthals into non-Africans and gauge the relative time depth of divergence, assuming Neanderthals diverged 550ka from a common stem.”

and

“The conditional site frequency spectrum (or cSFS) is the distribution of allele frequencies restricted to loci that satisfy a given condition. Specifically, we consider the distribution of allele frequencies in presentday populations conditioned on the Vindija Neanderthal carrying the derived allele relative to the inferred ancestral allele.”

--Gene flow from neandethal into Africans have been shown in recent times (Chen et al 2020). The authors should discuss how this will affect their results.

Chen et al, 2020 do indeed find Neanderthal ancestry in Africa. Specifically, they find approximately 17Mb of Neanderthal ancestry per individual, or up to about 0.3%. They attribute the bulk of this signal to back-to-Africa migrations.

Since our models do feature Back-to-Africa migrations, following the reviewer suggestion, we checked whether the model-predicted proportion of Neanderthal ancestry in African samples matches these observations. Indeed, we find that the two best-fitting models predict between 0.1 and 0.8% of Neanderthal ancestry across African populations in our analysis, consistent with Chen et al.

We agree with the reviewer that this is a relevant observation. We added a supplementary section describing this analysis, called “Back-to-Africa Gene flow and Neanderthal ancestry in Africa”, which reads:

“Back-to-Africa migrations likely introduced Neanderthal ancestry into Africa (Chen et al., 2020). This Neanderthal ancestry would cause reticulation in the ancestry of the African populations, and is therefore relevant to the modeling presented here. The models we inferred account for both Neanderthal admixture in Eurasia and Back-to-Africa migrations, such that they predict the amount of Neanderthal ancestry that we would expect to find in African individuals. We find that African populations without high proportions of recent Eurasian ancestry are predicted to carry 0.1 – 0.2% Neanderthal ancestry (Table S7), which is remarkably consistent with the results of Chen et al. (2020). Since the IBD statistics from Chen et al. (2020) are not used in the present analysis, this serves as an independent validation of the present models. More importantly, since the Neanderthal admixture in Africa is properly accounted for in the model, we do not expect the presence of such ancestry to bias our estimate of reticulation in Africa.”

6. Regarding this section: “The earliest divergence among contemporary human populations differentiates the southern African Nama from other African groups between 110–135ka, with low to moderate levels of subsequent gene flow (Table 1). In none of the high-likelihood models which we explored did the divergence between Nama and other populations exceed ~140ka.”

Are the authors sure the significant recent geneflow that the Nama received does not influence estimates?

As mentioned above, we re-ran our model fits after selecting only individuals whose ADMIXTURE-inferred proportion of Nama-like ancestry is above 90% and 99%. Aside from parameters pertaining to admixture proportion in the Nama, we found little difference in inference. See Section S7.1.3:

In the demographic inference results presented in the main text, we did not impose a minimum threshold on inferred Nama ancestry (as estimated using ADMIXTURE). Because our proposed models allow for post-divergence gene flow as well as recent admixture events, we can jointly learn deeper history and recent admixture dynamics. This is in contrast to methods that do not account for ongoing migration or recent admixture events, for which un-admixed genomes (if they even exist) are needed for unbiased estimates of earlier history. Although the methods used in this paper (moments-LD) are able to infer early and recent history jointly, we tested the robustness of the inferred early history to varying ancestry proportions in the Nama population. To do this, we used ADMIXTURE to cluster inferred ancestry components, noting that ADMIXTURE-inferred ancestries are coarse and can be misleading about true underlying admixture proportions. Nonetheless, we included Nama individuals that exceeded either 90% or 99% ancestry primarily shared among Nama individuals.

For both thresholds, early history was robust in all tested models. Divergence times and migration rates did not vary significantly from the original fits that included all unrelated Nama individuals. The primary differences between these fits and the original fits that did not impose an ancestry threshold were the inferred admixture proportions of East African agriculturalists (2ka) and Europeans (10 generations ago) in the Nama population. The East African admixture proportion was reduced from $\approx 25\%$ to 20 – 22%, while the European admixture proportion was reduced from $\approx 15\%$ to either 5 – 6% (with a 90% ancestry cutoff) or 3–5% (with a 99% ancestry cutoff). No other parameters were strongly affected by subsetting the Nama individuals by ADMIXTURE-inferred ancestry proportions. Because only the inferred proportions of recent admixture in our model fits changed, we conclude that our inferences of early history are robust to recent admixture, as long as that recent admixture is accounted for in the demographic models.

There have several been previous estimates (not based on cross-coalescence methods that the authors indeed discussed) that have estimated the divergence of the Khoisan to over 200KYA when accounting for different mutation rates (Gronau et al 2011, Veeramah et al 2012 and Schlebusch et al 2020). The authors fail to comment on these studies and estimates. Even the Nama group split specifically (Schlebusch et al 2020) was estimated before to much deeper times. The author fails to comment on this – even when it was on the same population group

We agree that this deserves more discussion.

The discrepancy between estimated split time in previous papers has been discussed extensively in two recent reviews (Henn et al. 2018 and Bergstrom et al. 2021), and are indeed very relevant. We did discuss these discrepancies in the supplement, but we were missing text in the main paper linking to this discussion. Please see Section 7.3 and in particular Figure S35. In brief, prior reviews have highlighted several sources of variation in divergence time estimates, a) mutation rate, b) inference method, c) data type, d) populations chosen for comparison. It is beyond the scope of this paper to reconcile all of the observed variability in divergence estimates, but we highlight a few points.

- 1) Following the reviewer comments, we further investigated whether the recent gene flow into the Nama might have influenced the discrepancy between present results and Schlebusch et al. 2017, which were based on an ancient genome with no ancestry derived from the recent gene flow. To better understand the behavior of previous approaches that used allele frequencies and IM models, we simulated genomes

sampled both before and after eastern African gene flow, according to our inferred models. We then performed inference based on allele frequency and an IM model (Fig S35). The difference in estimated split times using sampling before or after admixture (square vs. circle) was small under the 4 major simulated models. We conclude that the difference in eastern African ancestry between ancient and present individuals is not the primary reason for the discrepancy, which is more likely due to differences in model specification.

- 2) Figure S35 shows that, by contrast, model misspecification strongly affects divergence time estimates. Using data simulated using all four of our models (including our reticulated or 'merger' models), allele frequency spectra approaches that assume a single split with migration between constant-size populations infer variable and very deep split times.

We have added text in the main text, at the end of Section Reconciling multiple lines of genetic evidence:

Other studies have fit tree-like demographic models to African populations using distributions of allele frequencies or related statistics, finding inconsistent divergence times, some of which are older than those we find here 17,29 . In the Supp. Information (Section 7.5), we show that this discrepancy can be explained by model misspecification: if divergence is estimated using an isolation with migration (IM) model with constant population sizes, but the correct model has ancient population growth or population structure, the split time in the inferred IM model is much deeper than in the correct model. Intuitively, growth or structure in the ancestral population will each increase coalescence times relative to a randomly mating population of constant size, so a model that assumes constant population sizes would require an older divergence time to fit the observed distribution of coalescence times and related statistics 30,31.”

- 3) Our interpretation is that these deep split times in IM models with migration allows the IM model to capture the relatively low coalescence rate 100-300ka. This interpretation is further supported by a recent preprint:

<https://www.biorxiv.org/content/10.1101/2022.06.17.496540v1>

- 4) This highlights an additional complication in trying to compare inferred split times across models that specify migration or reticulation differently. For example there is a large difference in the inferred split times for models with and without migration on Fig. 31. IM models with migration have a single event where an ancestral population splits. Reticulated models have multiple such events. It is therefore difficult to meaningfully draw a one-to-one correspondence between the inferred split times of such distinct models. Our manuscript thus focuses on interpreting the best-fitting models and explaining the observed discrepancies.

We have further clarified the language to emphasize that we are describing, for each population, the most recent split time: *“In a reticulated model, we use “divergence” between populations to describe the time of their most recent shared ancestry.”*

7. Connecting to the previous comment.

The authors only comment on previous cross coalescent population divergence estimates:

“Published estimates of the earliest human divergences with rCCR, which range from 150ka-100ka²⁹, may be significantly biased when compared to more complex models with gene flow as inferred here. We find that midpoint estimates of rCCR are poor estimates for population divergence, often underestimating divergence time by 50% or greater (e.g., Mende vs. Gumuz ~15ka compared to a true divergence of 60ka), and recent migration can lead to the misordering of divergence events (Figure 5E). We suggest that rCCR analyses which do not fit multiple parameters including gene flow should be interpreted with caution.”

What about other studies and other estimations of population splits that have been done before? These split times, corrected for mutation rates were nicely summarized by Bergstrom et al 2021 Nature – in their Figure 2c. All the non rCCR estimates were much deeper estimates and the authors should comment on this in light of their own shallower split-times.

We believe that this comment is addressed in our reply to point 6.

8. Still connecting to the previous two comments

I'm very surprised that the authors consistently get such low divergence time estimates. Their simulations do assume recent mutation and recombination rates, yet perhaps they are not sufficiently taking into account recent gene flow among contemporary populations? For instance, their claim that the earliest population divergence among contemporary populations occurs 120-135ka and that simple "model misspecification" explains variation in previous divergence time estimates.

This doesn't seem right, because the kinds of bias we would expect under such simple models is opposite to what they suggest. For example, a clean split model that assumes no post-split gene flow will give more recent divergences than reality. So, they can't use that as an explanation for why those simple models give much older split times.

We also believe that this has been answered in the reply to point 6. Intuition about what misspecified models do with data generated from reticulated models is difficult. For this reason, we have performed simulations under our reticulated models, and then inferred split times under both clean split and IM models. These simulations show that IM models do indeed infer deep split times that may not correspond to the most recent divergence. This has now been clarified in the main text and supplement.

We note that our single-origin model fit to the data also features a relatively recent split time (110ka). We believe this is because our model allows for an ancestral population expansion before divergence among different groups. This “expansion” reflects the well-documented partial confounding of population size changes and population structure (<https://www.nature.com/articles/hdy2015104>, see also recent preprint: <https://www.biorxiv.org/content/10.1101/2022.06.17.496540v1>). Despite this partial confounding, this single-origin model describes the data poorly relative to the structured models.

The authors might then argue that ancestral structure would bias such simple models in this way -- However the cross coalescence analyses that the authors do have a drop off after 300k that indicate a bottleneck. Inversely in their weak stem model should result in an increase in N_e

here but there is a drop in CC. The relate IICR curves show a large reduction in $N_e > 300ka$. This pattern in the IICR curves is something the authors seem to avoid addressing, instead very generally saying that between 100-1000ka ago there was a large increase in N_e . The weak stem model they suggest should give rise to high N_e 's while they suggest population structure exists (100-1000ka), but the curves contradict this.

We agree that the IIRC curves simulated under our models show a dip in N_e that occurs deeper in the past compared to the Relate-inferred IIRC in the data (Figure S19 and S20). We did discuss this in Section 7.3.2 in the supplement, but were missing a link to this section in the main text. This has been corrected.

"All models, including the single-origin model, recapitulate an inferred ancestral increase in N_e between 100ka-1Ma extending deeper in the past than the Relate estimate (Figure S25 and SI Section 7.3.2)."

There are a few reasons why we did not discuss this more extensively in the main text. First, this discrepancy occurs as well in the single-origin model, which infers a population size increase around 500ka, leading to a drop off of IIRC extending back to 1000ka (S25A). So while we agree that there is a discrepancy with the Relate results, we do not feel that this particular discrepancy is informative in our effort to distinguish between the differently structured models.

Second, two major limitations of the Relate approach complicate the analysis. The Relate curves themselves showed imperfect agreement even across two runs using the exact same individuals: Figure S24 A and B compute the IIRC from the Relate trees for the same individuals, and differ only in whether other individuals were included when generating the tree to estimate the coalescence times (i.e, we computed a tree with more individuals, then discarded the branches and coalescence events that were not ancestral to the focal individuals). While these runs should give identical results if the Relate tree is correct, they varied appreciably. In addition, a recent paper showed systematic biases in estimation of ancient TMRCA in Relate, even under idealized simulations (Brandt et al, Evaluation of methods for estimating coalescence times using ancestral recombination graphs, Genetics 2022), see their Figure 4:

The period of maximum bias in these analyses corresponds to a number of generations around $2 N_e$ generations, which in humans corresponds (very roughly) to 20k generations or 600,000 years. In other words, TMRCA's based on Relate in that period are known to be biased in a setting where our approach is not.

Third, while this reduced IIRC occurs deeper in the past than the Relate-inferred IIRC (fig S23, S24), the timing of the drop is consistent with the early population size increase found using a single-origin models fit using a West African populations and the distribution of allele frequencies (Eg Gutenkunst 2009 or Gravel 2012), once these have been scaled to an updated mutation rate.

So it seemed beyond the scope of the main text of the manuscript to discuss quantitative discrepancies with Relate for that time period. Indeed, we are not familiar with any manuscript that has shown quantitative agreement between a detailed model prediction and a Relate curve.

So, I am left curious as to how and why they consistently get such low divergence time estimates.

We hope that the arguments above will have addressed the concerns. In short: Our models allow for either reticulation or early population expansion. We find that models without these features, used in much of the previous work, tend to infer deeper divergence times in the presence of reticulation. They also do not describe the data as well as models with ancestral growth or ancestral reticulation.

9. The fact that the authors fix many of the parameters (e.g. Out-of-Africa at 50ka – which seems quite recent!) to gain resolution on deeper events, could potentially be problematic. The authors should discuss their fixed parameters more and how much it is influencing the other estimates

To address this concern, shared by reviewer 2, we have added a number of additional analyses. We first refit the demographic models with deeper times for those fixed parameters, namely 60ka for the Out-of-Africa event, and 100ka for the divergence of East and West Africans. This resulted in inferences with slightly older divergence times for the Nama and the timing of the stem population events, with those increases scaling roughly proportionally to the relative differences of the choices of fixed parameters. However, this choice of older fixed parameters also resulted in a poorer fit to the data.

We also refit our demographic models while allowing those parameters to be jointly optimized. This resulted in more recent dates for the OOA and E/W African splits, at roughly 35-40ka and 40-60ka, respectively. These dates are quite recent, probably due to model misspecification regarding the mode of population divergence during the OOA event. Section 3.1 expands on our choices for fixing parameters, and we address this particular choice of fixing the dates of the OOA and E/W African split, writing:

“In our preliminary models, the unconstrained inferred split times between East African populations and Europeans was inferred to have occurred by around 40ka, which is more recent than we may have expected given the well-documented presence of modern humans outside of Africa at that time (Hublin et al., 2020; Hajdinjak et al., 2021, e.g.), as well as previous genetics-based dating of splits between African and Eurasian populations (Jouganous et al.,

2017; Kamm et al., 2020, e.g.). The unconstrained inferred split between East and West African populations occurred about 10ka earlier. These recent split dates seem to be at odds with archaeological evidence for earlier Out-of-Africa (OOA) expansion, and with the estimated date for Neanderthal admixture into OOA populations. The parameter uncertainty due to finite genomes does not account for this difference: in the high-likelihood models, the 95% confidence intervals for the East African-European split was roughly 38 ± 5 ka, and for the East/West African split was 45 ± 10 ka, with the uncertainties reflecting 2 standard deviations according to bootstrap of genomic regions. Thus we expect these discrepancies to result from model misspecification in the details of the divergence between the ancestors of OOA and east African populations. Our models suppose a single split prior to the OOA event followed by constant symmetric migrations between eastern and OOA populations, which is certainly an oversimplification.

Since a detailed model for the Out of Africa and Back-to-Africa was not our main focus, we chose to keep this simplified model, but fix the split between east African-European split at 50ka. This allowed us to include Neanderthal admixture into the OOA population at the previously-estimated 45ka. Fixing this parameter pushed back the estimated value for the split between the ancestors of East and West African populations at around 60ka, a value that we took as fixed in most subsequent analyses. We discuss the sensitivity of our model to these assumptions in Section 7.2.”

10. I was not sure which model that the authors tested correspond to Figure 1 D-multiregional. It just seemed that they tested the single origin models and different versions of the weak stem model. They authors should more clearly link their models that they tested with the models shown in Figure 1.

When fitting demographic models to genetic data, we specify a tree topology and parameters that are either fixed or to be fit by the model. We then let the moments software optimize the parameters. Two models that are very distinct from an anthropological perspective (for example, because the split times occur at very different epochs) may have the same model specification.

We ensured that the four qualitative models outlined in Figure 1 were included in the space of parameters that were searched. However, the fitting procedure selected parameter values that correspond to the weak stem model.

We have clarified this in the main text:

“We began with a model of geographic expansion from a single ancestral, unstructured source followed by migration between populations, without allowing for contribution from an African archaic hominin lineage or population structure prior to the expansion (Figure 1A or Figure 1D). As expected, this first model preferred a recent expansion model but was also a poor fit to the data qualitatively (Figure S5) and quantitatively (log-likelihood (LL) $\approx -189,400$, Table S2). We next explored a suite of parameterized models in which population structure predates the differentiation of contemporary groups (Supp. Information section 3). Depending on the parameters, these encompassed models allowing for ancestral

reticulation (such as fragmentation-and-coalescence or meta-population models, Figure 1B), archaic hominin admixture (Figure 1C), and African multi-regionalism (Figure 1D)."

11. *The authors tend to do self-referencing and, in some cases, prefer referencing their own findings rather than original papers that showed specific findings. They also do not reference papers that previously demonstrated many of their findings (Back-to-Africa gene flow, West African population expansion associated with agriculture, East Africans into Nama ~25%, 15% colonial admixture into Nama etc).*

While we are not immune to a bias towards our own work, a quick count shows that 7 out of the 43 original references include one of the co-authors. This does not seem excessive to us. Three of these [9,24,37] point to specific methods we have developed and used in this work. Two point to the datasets we have used in this work [19, 21]. One is a review paper that outlined the problem that we attempted to solve in this work [17]. Unless we missed something in our count, that leaves one possibly discretionary reference [25].

We have added an additional reference to the Back to Africa gene flow proportions in Ethiopians (Molinaro 2019, *Scientific Reports*).

We do use the Henn, Steele, and Weaver review quite a bit since there is a limit on the number of references in Nature, such that we cannot cite all the relevant primary papers (including our own!).

12. *Regarding this sentence:*

"We observe significant gene flow from the Amhara and Oromo into the Nama, a signal which is likely a proxy for the movement of eastern African caprid and cattle pastoralists 25,26, here estimated to constitute a 25% ancestry contribution 2,000 ya."

The reader might wonder why this is not seen as such in the ADMIXTURE analyses included in Fig 2 D – where the admixture just looks like European admixture – the authors should comment on it and explain it in the text.

Thank you for the comment. Indeed, Figure 2D shows that ADMIXTURE detects virtually no East African ancestry in that Nama (i.e. no red in ADMIXTURE bar plots). Our interpretation is that the East African ancestry is homogenous across individuals within the Nama (having occurred relatively deep in the past 2,000ya) and thus when allele-frequency specific clusters are identified by the algorithm they do not parse out this ancestry separately but rather as indistinguishable from the Khoe-San component. It is worth noting that the subsequent bottleneck in the Nama, observed here and in their long runs of homozygosity, will cause a deviation from the ancestral East African allele freq via drift – even if the Amhara/Oromo are appropriate source populations. In this context, the likelihood of the ADMIXTURE model will not increase by adding separate East-African-like and ancestral Nama-like components. Nonetheless, we are able to reconstruct a pulse of migration from East Africa with an appreciable contribution via *moments*. (See also reply to point 2 of reviewer 3).

“We observe significant gene flow from the Amhara and Oromo into the Nama, a signal which is likely a proxy for the movement of eastern African caprid and cattle pastoralists^{25,26}, here estimated to constitute a 25% ancestry contribution 2,000 ya. While this gene flow is not apparent from the ADMIXTURE plot (Figure 2D), the ancestry is likely grouped into the Khoe-San component which has drifted appreciably from its ancestry Eastern African source.”

13. Regarding this section:

“Following these merger events, the stems subsequently fracture into subpopulations which then appear to persist over the past ~ 120ka. These...”

We know that these populations were most probably connected and not fractured into subpopulations to up until the Bantu-expansion- there is archeological evidence of hunter-gatherers living all across the continent with no gaps in connectedness. Then with the Bantu-expansion the HG groups were fractured and replaced and only a few groups remain. Where would these “in-between” groups fit in the model? The population “fracturing” might just be a sampling bias on a previous gradient of connectedness. Sampling specific points in a “isolation-by-distance” space would look like a diverging model with “fracturing into subpopulations”. The authors should discuss this more in terms of how the bantu-expansion influenced the fracturing of groups and how the landscape of geneflow might be explained before this period

We agree that the “fracturing” (or splitting) in a model does not need to correspond to an abrupt split that would, for example, have been discernable by people living at the time. There are many scenarios, such as continuous isolation-by-distance models followed by gradual isolation, that could lead our model to infer a discrete split. While the reviewer is correct to point out that some previous papers have highlighted a hypothesis of a hunter-gatherer network across the continent prior to the Bantu-expansion, this model generally remains poorly parameterized. In Schienfeldt (2019, PNAS), there is no migration allowed among Eastern African hunter-gatherers but only gene flow sourced from agricultural/pastoralist groups during the late Holocene. In Patin (2009, PLoS Genetics), gene flow between the Eastern and Western Central African foragers was on the order of $m=5e-4$, so fairly low. The extent to which hunter-gatherers from eastern Africa exchanged migrants with those from e.g. southern Africa remains an open question. The archaeological data are also incredibly patchy across time and space.

Here, we do find migration occurs between E/W/S regions prior to 5,000 ya but these migration estimates remain relatively low. For example, we note the migration between the ancestors of the Nama and other hunter-gatherer groups during that period (parameter ‘e’ in both models) is an order of magnitude weaker than that observed between other groups ($m=4.4e-5$, compared to e.g. $2e-4$ between Mende and East African groups). Our results are not at odds with a scenario of many HG demes across Africa prior to the Bantu expansion, but suggest that migration between disparate regions was limited. While we cannot confidently distinguish between an isolation with migration and a continuous model with discrete sampling, our model seems at odds with a model of a well-mixed extended population. We conclude that populations between regions were likely highly structured before the Bantu expansion, with low levels of gene flow among them. A fuller account of the migration network will need to wait for ancestries extracted from admixed Bantu groups, and better spatially-aware approaches.

14. Regarding this sentence:

“By contrast, an archaic hominin population in Africa would need to have stayed in relative reproductive isolation from the ancestral human lineage over hundreds of thousands of years despite closer geographic proximity and reproductive compatibility.”

The authors used this statement to argue that their weak stem model are more likely than archaic admixture models and that it was unlikely for these very different populations to co-exist in the same geographic space.

*This however apparently happened with very solid evidence of the *Homo naledi*, Kabwe/Broken Hill (Grun et al 2020, nature) fossil remains. The authors commented on *H. naledi*, which most likely was to archaic or differentiated to have mixed with our lineage. But what about Kabwe that was possibly closer to our lineage and coexisted with us to relatively recent times.*

In the paragraph from which this sentence comes, we were not trying to argue that archaic hominin populations could not coexist with ancestral human populations. The purpose of this paragraph was to make connections between our results and ecology. In the revised version of this paragraph, instead of framing this discussion as an “ecological riddle”, we now simply highlight that there are potential ecological implications of preferring weakly-structured-stem models over archaic admixture models.

More generally, we agree with the reviewer that there is evidence for coexistence of archaic hominin populations and ancestral human populations with the dating of the *Homo naledi* and Kabwe/Broken Hill fossils, and even perhaps at Omo Kibish with the contrasting morphologies of Omo 1 and Omo 2. However, archaic hominin admixture models require not just coexistence but coexistence with 1) reproductive isolation between populations over hundreds of thousands of years and 2) an abrupt loosening of this isolation. In contrast, weakly-structured-stem models simply require some population structure.

The last paragraph of our manuscript in which we mention *Homo naledi*, which the reviewer alludes to, was not meant to be a comprehensive discussion of the fossil record. Instead, the goal was to use examples from the fossil record to help explain how our models could be related to the fossil record. We picked the *Homo naledi* fossils as an example of fossils that are morphologically so divergent from contemporary humans that they are unlikely to represent a population that was part of the weakly structured stem. Without detailed morphological analyses, which seemed out of scope, it is difficult to say if fossils from Jebel Irhoud or Kabwe/Broken Hill could have come from populations that were part of the weakly structured stem.

15. At one point in the main text the authors say they cannot assign any of the inferred stem groups to be the direct ancestor of any contemporary populations (the weak stem model has gene flow between stem populations for a long time, meaning only 1-4% genetic differentiation in modern populations arose as drift in this period of population structure). But then they later (tentatively) suggest Stem 1S = southern Africa, Stem1E = Eastern Africa, and Stem 2 = Western or Central Africa.

I think the authors should be very careful not to overinterpret results based on the model with highest likelihood. They themselves offer the caveat that this study does not reject more complex scenarios with more stem populations or hybrid models including ancestral structure and archaic admixture, thus they should be careful to assign geographic locations to hypothetical inferred stem populations.

We agree and added language to clarify that we are speculating:

“Speculatively, this may indicate that an ancestral Stem 2 population occupied western or Central Africa, broadly speaking. The differing proportions in the Nama and eastern Africans may also indicate geographic separation of Stem 1S in southern Africa and Stem 1E in eastern Africa.”

16. The figures in the Supplement are quite difficult to interpret. What the various panels mean, and specifically what "statistics" are represented on the Y-axis. Even after reading the paper I am unsure of the specific diversity-based statistics that were used in ML estimation of parameters of models.

Part of the power of the approach is that it uses a family of statistics that are informative of different aspects of genetic diversity. One downside is that this is a lot to take in! Supplementary section 2.1 provides an overview of the statistics, and we refer the reader to the original methods paper for a more in-depth discussion.

A second downside is that interpreting the effect of individual model parameters on the overall fit can be difficult. While we agree that interpreting the plots in supplementary figures S6-S12 is challenging, we find it important to provide an illustration of the quality of the fit for the different statistics used.

17. It is not clear whether authors had permission and co-working of relative community councils regarding the Khoisan data

Please see our extended response to this query below.

Referee #2 (Remarks to the Author):

The manuscript by Ragsdale and colleagues reports an investigation into the origin of population structure in the early human lineage based on a comparison of the performance of four competing demographic models in explaining the genomic diversity of 290 whole-genome sequences.

The best-fitting model identified in this study suggests an early divergence of two stem populations with continuous gene flow between them. It further suggests that differential merging of these stem lineages (probably in different parts of the continent) could have resulted in the geographically stratified population structure that is observed among the contemporary African populations. The model further estimates the structure observed in the contemporary African populations to have emerged in the last 150 Kya. The results challenge the popular

notions that archaic admixture (similar to Neanderthal and Denisovan in non-Africans) has contributed substantially to the genomes of current-day Africans.

The manuscript deals with one of the difficult and debated questions in human evolution and provides a novel perspective of early human history. Given, the novelty and immense importance of these insights, it is critical to verify whether the dataset used is optimal for testing these complex demographic models, whether the models give similar predictions for a range of datasets and parameters and whether the analyses are extensive and rigorous.

We thank the reviewer for highlighting the importance of these insights!

I have the following major recommendations :

1. As the confluence of ancestries in a population has the potential to influence the fitting of demographic models, the choice of populations (and ancestries represented in them) assumes critical importance in a study like this.

The ADMIXTURE plots show three of the populations tested (Nama, Oromo and Somali) to have substantial gene flow from the Out of Africa group (GBR). Moreover, some gene flow within the three East African groups and also from Bantu-speaker to some of the East and South African populations is plausible. In addition, gene flow from East Africa to Nama in the last two millennia is well known and has been reiterated by this study. Given this multi-level interconnectivity and gene flow between populations, some of which are too subtle to be visualized by ADMIXTURE, it is important to assess the impact of these gene flow events on the fitting of the demographic models. I would recommend the following additional analyses/iterations to improve the robustness with respect to admixture -

i. Replace the Nama with "less admixed" Khoesan genomes from Schlebusch et al. 2020 and test whether the model fitting is impacted by the Eurasian admixture in Nama. In case this is not feasible at least the Khoesan ancestry cut-off for inclusion should be increased from 70% to perhaps 80 or 90%. and then the results are compared.

Thank you for the suggestion. We agree that some of these patterns occur too deeply to be well visualized by ADMIXTURE or PCA (please also see response to reviewer 1, point 12).

Including the separately-called Schlebusch et al. data would require us to re-call the genomic data jointly in order to mitigate batch effects (see also response to reviewer 1, point 2). Small sample size, coverage and computational time are all major concerns to perform this check. In addition, we do not presently have access to the original fastq data for the Ethiopian populations. The sequence data used in this analysis relied on prior African Diversity Reference Panel (ADRP) joint calling of genotypes, and all steps of quality control and data filtering were performed uniformly at the beginning of the project.

We therefore followed your suggestion of varying the cutoff for the European component in the Khoesan, selecting cutoffs of 90% and 99%. We found that the choice of a cutoff had almost no impact on the inferred parameters, with the exception of the inferred proportion of recent admixture into the Nama (See Section 7.1.3).

ii. An iteration where a Chinese/Japanese population is used instead of the British (as there is no major gene flow from these groups into Africa) to represent out of Africa population could increase the confidence in the results.

This is also an interesting idea, however we are somewhat confused as to what this replacement would demonstrate. We still need to include a European ancestry population to account for the colonial admixture into the Khoe-San 10 generations ago, and this would add substantial complexity to the model. Without a European population in the model, the model would likely still use the Asian ancestry population as a proxy for the out-of-Africa ancestry, so we are not sure how such results would be interpretable.

In an attempt to answer the question of robustness to the OOA population, we did replace the GBR by the CEU in the fit and found no significant parameter changes (Supplementary section 7.1.1). We also repeated this analysis using the Tuscan (TSI) instead of GBR, which have had documented gene flow with African populations. This led to one qualitative difference in one of the models, i.e., the inferred split time for the Nama was 50% deeper in the continuous migration model. Other models, including the highest likelihood model, were broadly unaffected. These results are discussed in Section 7.1.1, and referred to in the main text together with other robustness testing:

“To assess robustness of the inferred models to analysis and reference population choices, Sections 6 and 7 present re-analyses changing the European and West African populations, as well as the recombination maps, filtering strategies and parameter optimization strategies. While we find some differences in the inferred parameters (see in particular 7.1.1 and 7.2), the best fit model across all reanalyses are quantitatively consistent.”

iii. I am guessing that to reduce the impact of the Bantu expansions on the model fitting, the authors have included MSL instead the more common West African proxy - YRI. However, an iteration where YRI or ESN replaces MSL could also be used to assess the extent to which the inclusion of Niger-Kordofanian ancestry might influence the fitting of these models.

Thanks for suggesting this. We have done this analysis (Supplementary section 7.1.1) and indeed find that parameters are broadly unaffected by the substitution.

iv. Probably due to scarcity of population-scale WGS data the Central African forager-related ancestry was not included. However, this is a major limitation as one of the key components of African diversity is completely missing in this study. Could the authors include some additional analyses that included Central African foragers (maybe from Simons Genome or HGDP dataset) to check whether and how they fit into the model. Although due to the smaller sample size the results won't be as reliable it would still be good for the readers to know.

Please see response to Reviewer 1, point 2. We agree that this would be desirable, but the time needed to access the raw data (if available), jointly call, and analyze these datasets would likely be longer than the time it will take for new, more comprehensive datasets to be generated. We are currently generating such a dataset, but this is a multi-year international effort.

As suggested by the reviewer, we have added Supplementary section 1.3 explaining our rationale for our choices of population inclusions and exclusions. From that section:

“The number of publicly available whole-genome sequenced individuals from a diverse set of African groups has grown in recent years (e.g., Mallick et al., 2016; Bergstrom et al., 2020; Schlebusch et al., 2020). Our choice of populations was motivated by both technical and study design considerations. First, while the diversity and linkage disequilibrium statistics used in our inferences are robust to low coverage sequencing data (Ragsdale and Gravel, 2019), we wanted to have at least 20 individuals per population to increase accuracy in statistical measurements by reducing the variance due to small sample sizes (Ragsdale and Gravel, 2020). Second, merging of datasets post-variant calling can introduce biases in demographic inference due to batch effects (as demonstrated in Section 7.6). We therefore wanted to be able to perform joint calling in all of our samples. Together, these considerations make the jointly called Thousand Genomes and ADRP dataset best-suited for our demographic inference.

From a study design perspective, our choice of populations aimed to improve our understanding of early human demography in Africa. The Nama derive ancestry from a relatively deeply diverged population branch, and thus their demographic history is of unique interest. Given previous archaeological and genetic evidence, we expected that a minimal model for the Nama must include pastoralist East African groups that contributed to the Nama (e.g., Uren et al., 2016), as well as European colonial admixture. The inclusion of three Ethiopian populations here represent some uncertainty in the possible source population for the pastoralist contribution to the Nama, and the Amhara and Oromo also have substantive “back-to-Africa” ancestry. The Gumuz serve as a representative of southwest Ethiopian ancestry that pre-dates the Holocene back-to-Africa migration. Finally, because previous studies have suggested a role for “ghost” archaic admixture or deep structure in West African populations (e.g., Speidel et al., 2019; Durvasula and Sankararaman, 2020), we chose to also include representatives of West African ancestry.”

v. Similar to the approach used for the single Neanderthal genomes some of the African ancient genomes from the recent Lipson et al. study could also be tested with these models.

Linkage disequilibrium statistics are difficult to measure accurately in ancient genomes, so that we need to restrict ourselves to a small subset of our statistics for ancient populations. This gives us much less resolution than for modern samples. Given the exceptional importance of the Neanderthal genome in refining our understanding of human expansions out of Africa, we decided to include the genome in our models and analyses. Even then, we had to fix some parameters established in previous studies to account for the reduced resolution in the Neanderthal.

The Lipson et al. genomes represent much more recently diverged populations, and so the addition of these genomes, while possible, would be unlikely to be informative about the deeper history in our opinion.

vi. Also, the sample sizes vary widely among the 6 groups, ranging from 30 to 100 individuals. Could the authors comment on whether this could have influenced the results to some extent and if possible provide some supplementary data to support their comment?

We made sure to use populations for which we had sufficient sample sizes to accurately estimate LD statistics, according to the analysis of [Ragsdale and Gravel, Unbiased Estimation of Linkage Disequilibrium from Unphased Data, MBE, 2019]. In that article, Fig1A-B shows that the LD estimation method we used (shown as solid lines) produces unbiased results relative to the truth (dotted line), for sample sizes above 10.

The main effect of sample size is on the variance of the estimates. Fig S3 of the same reference shows that the variance decreases roughly as $1/n^2$

Figure S3: **Variance of estimators decays with sample size.** Variances decay as $\sim \frac{1}{n^2}$ with diploid sample size n . These were computed from one million replicates sampled with the given sample size from known haplotype frequencies.

Larger samples are thus preferable, but this difference in variance across populations is accounted for in our model by the bootstrap procedure, and we do not expect it to bias the inference results.

2. The authors have included several dates from literature (such as those for divergence between western and eastern African populations, Out of Africa migration) as fixed parameters in the model testing. However, some of these dates have a rather wide range while others might get challenged/changed in near future. Therefore, it would be interesting to rerun the models

with alternative dates (earlier /later) for these events and provide an estimate of the extent of variations in final results with changes in the fixed parameters. For instance, if Out of Africa is dated at 70 Kya instead of 50 Kya, what would the rest of the divergence estimates look like?

This concern was shared by Reviewer 1 (comment #9). We ran additional analyses that fixed those dates to older times, as well as allowed them to be jointly fit, and we have expanded on these modeling choices in the supplement. Please see our reply to Reviewer 1, comment 9 for details.

Although authors provide some examples such as the one for population size expansion in the MSL that corresponds to archaeological records of agricultural innovations, they are all from a relatively recent time scale. Therefore, I would recommend also a set of analyses where a few of the key dates such as the split between East and West, OoA, etc are iteratively withheld, and the models are forced to estimate them. This could provide an assessment of how well estimates generated by the best fitting models concur with the estimates that were used as fixed parameters.

We thank the reviewer for this suggestion. In addition to fixing those divergence times to older dates, we allowed them to be jointly fit with other model parameters. Inferred models placed those dates even more recently, which we attribute to model misspecification surrounding the OOA event. This had only a small effect on other inferred parameters. Because the timing of the Out-of-Africa event is not the primary focus of this work, we decided to keep that date fixed to our previous choice of 50ka. We provide more details regarding this choice and an expanded explanation in the Supplement Section 7.2 and in our reply to Reviewer 1, comment 9.

I fully understand that implementing the alternative analyses suggested in #1 and #2 above might require substantial time and effort. However, as the results presented here have the potential to lead to a paradigm shift in the field, it is very important to ensure that they are based on rigorous testing and evaluation.

We thank the reviewer for the helpful suggestions. We were able to address most of them, and find that the added analyses improve our confidence in the robustness of the results.

3. *The inclusion of WGS from over 80 Nama participants is definitely a highlight of this study.*

Due to controversies in the recent past, a San Council has been set up in South Africa for assessing the potential impact and approving research (including genetic studies) on the Khoe and San people (including Nama) from the country. It is important to know if the authors have approached the council and applied for its approval for the study? I would strongly recommend them to get this research approved by the council.

In case this is not feasible, the authors should provide a clear rationale for not being able to do so. Also, a detailed description of the ethical considerations and community engagement steps involved prior to the study needs to be added to the supplementary information. Given that the

study is based on a marginalized and previously exploited group, it is critical to ensure that ethical considerations and community engagement has been adequately nuanced and sensitive.

Further details such as whether the data was collected for this particular study? whether informed consent allowed for secondary use? if yes, what kind of studies are permissible? needs to be included in the Supplementary Information.

We agree that this issue is of extreme importance, and now provide much more additional information in the Supplement Section 1.2.

Concerning the Jurisdiction of the South African San council:

In this manuscript, we present genomic data from the South African Nama community in the Richtersveld Community Conservancy. While the Nama are genetically similar to other San populations, they are not culturally and linguistically related to other San groups in South Africa (e.g. #Khomani San or displaced !Kung in Kimberley). As such, they do not fall under the jurisdiction of the South African San Council. We (BMH) have confirmed this directly by meeting with the San Council. Indeed, there is a strong preference to separate the two in order to receive appropriate government support and recognition; we follow their suggestion of hyphenating the word “Khoe-San” to distinguish between the different cultures.

In 2011, members of the Nama community in the Richtersveld were initially approached regarding a genetic ancestry study. The members included church elders, representatives at the Richtersveld National Park including the appointee for Peoples and Conservation (from which we obtained permission to sample individuals who still live in the park), the Richtersveld Cultural World Heritage Site Coordinator, and the Kuboes tourist office. All of these individuals self-identified as Nama. There was strong support for research that examined the extent to which the Nama are related to other Khoe-San populations, the impact of colonial migration, the time depth of their presence in southern Africa and relationships to other African groups. One primary concern was to make sure the research results were reported back to the community. Based on the positive response we received, permission was granted from the Stellenbosch University HREC to continue the study. Demographic and DNA collection was initiated in 2012 with a joint team of South African and American researchers which included both geneticists and anthropologists. Research assistants were fluent in Afrikaans, Nama and English. Collection primarily occurred at home, by first approaching a family member, gauging interest in the study, entering the home or sitting outside by their invitation, oral and written consent occurring with family members present and then finally completing the DNA sampling. By sampling individuals at home, members of the family are able to voice concerns, whether or not they decide to participate, and thus the final decisions are made in a slow and ethical manner. This process sometimes involved introducing the study and then returning at a later day in order to give participants sufficient time to consider.

DNA were collected for the express purpose of understanding Nama population history, their relationship to other Africans and human evolution. No commercial activities are allowed. The IRB has approved the data for public deposition and sharing provided that the research is related to population history or human evolution.

Direct quotes from the consent process include the following:

“We are here because we want to learn more about the history of South Africa and of the Nama and N|u (Bushmen) people. For our research we ask people a few questions about their family, collect DNA from saliva /or blood and measure particular things about them: height and skin color. We study how genetic (DNA, *explained above and in supplemental sheets*) data may affect physical characteristics like eyes or skin color, and to learn more about your group’s history and the history of African people. We have already learned that many people in the Northern Cape have San ancestors. The KhoeSan may be the most genetically diverse group in the world. We would like to study more KhoeSan people to better understand their special genetic diversity. To get useful results we will need approximately 200 people to take part in this research.”

“Your DNA will only be used for genetic research that is directly related to population history and human evolution.”

“Your DNA data will be combined with other individuals and the dataset will be put in a public database for scientists to use. Your name will NOT be in the database.”

We have added additional detail to Section 1.2 of the supplement. A photo of a community poster is described, as well as copied below. “*Return of Results*: Results stemming from genetic analyses have been communicated in 2015, 2019 and 2021 via community presentations, a radio interview and in person to representatives of the Richtersveld National Park. We created a poster illustrating Khoe-San ancestry proportions, discriminating ancestry on maternal vs. paternal lineages, the relationship between the Nama and Kalahari Khoe-San populations, and the evolution of light skin pigmentation in southern Africa. The poster was printed in Afrikaans and presented to the tourism office in Kuboes (a central location) for display. A display panel on genetic ancestry is currently under construction for the Richtersveld National Park entrance, created in consultation with the authors here. Future community meetings are anticipated in 2023. We remain committed to upholding this decade long relationship with the Nama, and value the community’s input in the research process.”

Die Genetiese Geskiedenis van die Nama in die Noord-Kaap

Brenna M. Hann^{1,2,3}, Justin Myrick⁴, Caitlin Urent⁵, Christopher R. Gignoux⁶, Julie M. Granka⁷, Meng Li⁸, Alicia R. Martin⁹, Sirju Kim, Cedric Wernly¹⁰, Carlos D. Bustamante¹¹, Marcus Feldman¹², Eileen G. Hoar¹³, Mario Müller¹⁴

bmhann@ucdavis.edu
marfom@sun.ac.za

¹ Departement van Ekologie en Evolusie en die Graadse Program in Genetika, Stony Brook Universiteit, SUNY, USA.
² Departement van Antropologie, University of California, Davis, USA.
³ Departement van Genetika of Biologiese Wetenskappe, Stanford Universiteit, USA.
⁴ DST-NRF Sentrum van Uitnemendheid vir Biomediese Tuberkulose Navorsing, Suid-Afrikaanse Mediese Navorsingsraad Sentrum vir Tuberkulose Navorsing, Divisie van Molekulêre Biologie en Mensgenetika, Fakulteit Geneeskunde en Gesondheidswetenskappe, Universiteit Stellenbosch, Kaapstad, Suid-Afrika

Navorsers van die Universiteit Stellenbosch in Kaapstad, Stanford Universiteit en Stony Brook University (VSA) het speekselmonsters van ~200 Nama wat in die Richtersveld woon versamel om die geskiedenis van die Khoesane te verstaan. Uit die speeksel het ons die DNA ontleed. DNA bevat inligting oor mense se voorouers: waar hulle vandaan kom, aan wie hulle verwant is. Navorsing het in 2012 begin en oor die afgelope sewe jaar het ons heel wat geleer oor die geskiedenis van die Nama-gemeenskappe.

Familie: 'n Nama-individueel het gemiddeld 3 Khoesane grootouers en een Europese grootouer. Oor die afgelope 150 jaar het beide Europeërs en Damara met die Richtersveld Nama getrou.

Moeders: 9 out of 10 Nama wat aan hierdie studie deelgeneem het, het 'n moederlike Khoesane afkoms gehad.

Vaders: 3 uit 10 Nama-mans het egter 'n vaderlike Khoesane afkoms gehad. Ongeveer die helfte van die Nama-mans het 'n vaderlike voorouer uit Europa of die Nabye Ooste gehad. 'n Paar mans se vaderlike afkoms is van Damara of Bantu-sprekers. Ons weet nie hoekom so min Khoesane mans die afgelope paar honderd jaar gelede kinders gehad het nie.

Daar is 3 verskillende Khoesane-voorgeslagte in Suid-Afrika.

Gedeelte van Khoesane Afkoms in elke populasie

Bevolkings: #Khomani San en die Nama woon in verskillende streke van die Noord-Kaap, en die Nama en Nju tale verskil baie van mekaar. Die #Khomani San en die Nama deel egter baie soortgelyke voorvaders omdat hulle DNA baie soortgelyk is. Maar ons weet nie wanneer die twee bevolkings van mekaar verdeel het nie. Nama het Khoesane voorouers wat in 'n kring rondom die Kalahari-woestyn gewoon het. Die Nama is unieke noordelike jagter-versamelaars soos die Ju/'hoansi San. Die noordelike en suidelike Khoesane het meer as 25 000 jaar gelede van mekaar geskeel.

Velkleur: Die Nama het ligter vel as Wes-Afrikaanse of Oos-Afrikaanse mense, maar donkerder as Europeërs. Die Khoesane het baie unieke gene wat velpigmentasie beïnvloed. Voordat die Europeërs na Suid-Afrika gekom het, het die Nama-mense en hul voorouers al tuisende jare 'n ligte velkleur gehad. Die voorouers van die Nama het hul gene vir ligte velkleur 2000-3000 jaar gelede al gekry van die mense van Oos-Afrika, toe Oos-Afrikaners na Suid-Afrika migreer het en vir die eerste keer skape en bokke vir die voorouers van die Nama gebring het. Vir die afgelope 2000-3000 jaar was die gene vir ligte velkleur vir baie geslagte van ouers na hul kinders oorgedra totdat dit wydverspreid was in die Nama en hul voorouers in Suid-Afrika.

Minor recommendations :

1. The ADMIXTURE plot is presented at $K=4$. Was this the optimal K based on CV scores?
Also, why the Khoesane ancestry partitioning was done at $K=6$ (Section 1.1 S Info) and not $K=4$?
I would recommend that the authors should add a full ADMIXTURE plot starting from $K=2$ to the best value of K in a supplementary figure?

We have added the following description to Section 1.1:

“ADMIXTURE: Nama individuals with >70% estimated Khoe-San ancestry were retained for analysis here, after partitioning ancestry into $K = 6$ clusters with an ADMIXTURE (Alexander et al., 2009) analysis on the full ADRP dataset (van Eeden et al. 2022). The 27 ADRP populations represent a wide variety of West African, East African, North African and Bantu-speaking South Africans. Europeans (1000 Genomes: CEU) were included to account for colonial admixture. A Khoe-San specific ancestry emerged at $K = 3$, predominant in the Nama. This ancestry remained stable at higher K 's. At $K = 6$, an ancestry representing the southern Bantu-speakers (Sotho and Zulu) also emerged, at low frequency in some Nama. Higher K 's highlighted population-specific ancestries or those with no representation in the Nama, therefore we focused on $K = 6$ for selection of high Khoe-San ancestry individuals. For illustration, ADMIXTURE analysis in Figure 1 was restricted to 1,157 individuals representing a subset of the full ADRP and several 1000 Genomes populations used in analyses presented here (see Section 1.3).”

The full set of ADMIXTURE plots are available in van Eeden et al. 2022, Genome Biology (now cited here). We use the ADMIXTURE plot in Figure 1 to illustrate the recent structure in the populations, rather than to make a detailed statistical claim. In the Figure 1 subset of the data, $K=5$ had the best cross-validation. Because the additional component introduced at $K=5$ identified additional structure within West African ancestries, as is expected given the large sample size of these populations in our analysis. While statistically valid, this additional component made the interpretation of the plot more confusing with respect to main discussion points in this paper.

2. Supp Info page 2 last line. Please check the spelling "Neanderhal"?

This has been fixed.

2. The same dataset has been referred to as ADRP (African Diversity Reference Panels) and AGRP (legend S Figure 18, penultimate line on Page 11). The authors should use one name consistently. Also, the original study by Gurdasani et al. 2015 was named African Genome Variation Project (AGVP) which is different from both the names used here.

Thanks for highlighting this, as this is highly confusing for everyone. Technically, Sanger Institute generated 3 datasets. The first was a low coverage AGVP, published in Gurdasani et al. 2015. The second was meant to include additional low coverage African populations, but in the end only the 84 Nama presented here were added; this became known as the ADRP. Finally, a third dataset of high coverage individuals that were a subset of AGVP and the Nama were generated and termed AGR/P. We hope that this clarifies the different acronyms. In the manuscript, we have chosen to use ADRP throughout.

3. Some more details such as - the number of SNPs in the two datasets, the size of the final dataset after merging, and also after genic SNP removal will be helpful.

As outlined in point 2, we are focusing in the main inference on a single combined dataset including the Nama, Mende, Gumuz, Amhara, Oromo, and British. We have now provided the relevant number of SNPS in Supplementary section 1.4, which reads:

“All analyses presented in this work focus on biallelic single nucleotide polymorphisms within the 1000 Genomes Phase 3 strict mask ($L_{\text{included}} = 2,064,554,803\text{bp}$). To enable comparison with Neanderthal DNA, we excluded regions for which the Vindija Neanderthal sample had less than 100 contiguous base pairs ($L_{\text{included}} = 1, 519, 643, 507\text{bp}$). For the moments-LD analysis, we further focused on intergenic regions because these appear less affected by natural selection compared to both synonymous and nonsynonymous variation (Ragsdale et al., 2018) ($L_{\text{included}} = 637, 639, 065\text{bp}$).

The 290 individuals used in the moments-LD inference featured 18,461,915 polymorphic biallelic SNPs genome-wide in the callable regions with Neanderthal alignment, of which 8,115,115 were in intergenic and thus putatively neutral regions.”

4. The authors have excluded all the genic SNPs in their analysis. It would be good to know if the inclusion of these SNPs would have changed the results substantially.

In general, we would expect the inclusion of coding snps to lead to moderate biases in demographic inference, due to the increased role of selection in these regions.

For example, in single-locus statistics, basing demographic inference on coding variation leads to observable biases. Basing demographic inference on synonymous variation also leads to biases relative to intergenic variation, but much weaker (see, e.g., Ragsdale et al, 10.1016/j.gde.2018.10.001). However, when computing genome-wide averages, the contribution of genic sites makes relatively modest contributions to genome-wide statistics.

To check that this was also true for two-locus statistics and that there were no large differences between the different regions, we re-computed LD and diversity statistics from across the autosomal genome (excluding only low-mappability regions in the Thousand Genomes strict mask), and compared to those statistics computed from intergenic regions alone. Differences were barely discernible for most statistics. For some statistics, small systematic differences could be seen but were small enough to fall within one bootstrap standard error (Figure S8). Differences were small enough that we would not expect qualitative differences in inferred demographic models.

We therefore chose to perform model inference using intergenic regions alone to reduce bias due to selection in and around protein-coding regions

5. The order in which the figures are referred to in the text needs fixing. Figure 2C is cited before 2A and B. Similarly Figure 4 seems to be referred to before Figure 3.

Thanks, we have reviewed the ordering of figures and citations for consistency.

6. What are the horizontal green lines in Figure 5 show? Needs to be described in the legend.

The dotted green lines have now been labeled. Thanks for pointing this out!

7. As most of the methodological details of the study have been provided in a huge supplementary information file, it is difficult to relate the analyses mentioned in the text to a particular section of the SI file. It would be more convenient if the authors refer to a specific section of the SI instead of just saying SI in the text.

Agreed, and this is even more important now given the many new supplementary sections! This has been fixed.

Referee #3 (Remarks to the Author):

Dear authors,

I found your paper really interesting, and a significant contribution to the current debate on the pattern and process of modern human evolution in Africa. You set out to formally test mathematically a series of complex models of evolution, something that is clearly needed. Like all models, those proposed are simplifications and accommodations of reality, but the results should contribute to moving the discussions forward. I have a few issues that I think you should explain further and/or consider, and a few suggestions at the end.

Marta

The paper obtains (in my view) five key results:

- 1. Single origin w/o introgr but w/ migration: poor fit*
- 2. Living African populations diversified 110-135ka (Nama first split), w/ low to moderate gene flow afterwards*
- 3. Some structure BEFORE differentiation of current populations*
- 4. Two models with highest likelihoods: A: "continuous-migration" (LL = -115, 500); B: "multiple-merger" (LL = -102, 600)*
 - With continuous migration between stems, population structure extends back to 1.1–1.4Ma*
 - In both models, the branch ancestral to the Nama shares a common ancestral population with the other African groups ~120–135ka.*
 - only 1% to 4% of genetic differentiation among contemporary populations can be traced back to this early population structure*
- 5. Best fit model: "Multiple-merger"*
 - Stems 1 & 2: continuous migration until ~550ka (Neanderthal split from Stem 1)*
 - sharp bottleneck in Stem 1 (down to $N_e = 117$) after the split of the Neanderthal branch.*
 - Stem 1 fractures into "Stem 1E" and "Stem 1S" 479ka, followed by independent evolution.*
 - ~120ka: merging of Stem 1S (29%) and Stem 2 (71%) to form the ancestors of the Nama*

- ~100ka: merging of Stem 1E (50%) and Stem 2 (50%) to form the ancestors of the W & E Africans
- Further gene flow from Stem 2 (18%) to W Africans 25ka

I think these results warrant the authors' conclusions that (a) single origin models without some form of multiple stem contribution prior to modern human differentiation are not likely; (b) that all living humans share a common African ancestral population 110-135Ka during MIS 5 from which they subsequently diversified; and (c) that despite the fact that discriminating amongst multiple configurations of what happened prior to that ancestral MIS 5 population is not simple, likelihoods can be determined hierarchically, and the best-fit model (amongst those tested) is one in which the population ancestral to all modern humans results from the merger of two lineages (contributing differently to different current regional groups) which originally experienced gene flow between them, but which also had a longish period of drift. The data used in the paper represent mostly a new genomic dataset that aims at better representation of diversity amongst living African peoples, providing new and better insights into population history. The data are presented clearly, although some of the figures & graphs could do with fuller captions.

The paper uses a wide range of mathematical and statistical tests. These are partly explained in the main text, and partly in the supplementaries, although for the non-VERY-specialist, some of these explanations are not easy to follow. I believe the analyses appropriately test the models set-out by the authors, but as I am not an expert, I cannot comment further.

From the above, it should be clear that I think this is a good paper that makes an important contribution, but I do have some issues with some of the interpretations and contextualisation of the results and conclusions:

1. regarding the fixed parameters, it is not clear how these were set (why, for example, the out-of-Africa at 50ka, or the Neanderthal introgression into Eurasians at 45ka?), or what impact it would have on the models if these were different. I think the manuscript would benefit from a bit more clarity on this.

See response to point #9 from Reviewer #1. In short, we have expanded supplementary section 1.3, explaining our detailed choices for parameter fixed, and added new sections (sections 6 and 7) testing robustness to these assumptions.

2. Also, you say “These constraints allowed us to integrate information from previous genetic and archaeological research to infer robust migration rates. For example, all models infer relatively high gene flow between eastern and western Africa ($m \approx 2 \times 10^{-4}$, the proportion of migrant lineages per generation)” (pgs 3/4). But why relatively high gene flow between eastern and western Africa? This is not supported by your Admixture plot results – these show that (unsurprisingly) eastern African farmers, such as the Luyha and Bakiga, are an unequal combination (merger in your terms!) of western African Bantu and local eastern African groups. But this happened in East Africa, and there’s minimal evidence of E African gene impact in

Western African Bantu. In fact, there is the same process with the Zulu – a ‘merger’ of Western African Bantu and local Khoisan genes – and yet this is not described and/or computed as relatively high gene flow between southern and western Africa. Can you please address this?

We revised this section in the main text to be clearer. We think that there were three sources of possible confusion here. First, the example estimate of East/West African $m \approx 2 \times 10^{-4}$ is not fixed but inferred from the models. The constraints alluded to are the fixed parameters discussed above this sentence. Additionally, keep in mind that this migration rate reflects the long term migration from the divergence between Eastern and Western Africans for 60ka.

Second, regarding the apparent discrepancy between the high gene flow and the admixture results: ADMIXTURE is designed to accurately capture ancestral components in recently admixed populations, where individuals vary appreciably in their ancestral proportions. It does less well at capturing older events and long term migration, in which individuals are more likely to have similar amounts of the contributing ancestral components. Given the allele frequencies in Eastern Africans (Gumuz, Amhara) and those in contemporary Yorubans are fairly distinct, it is easier to distinguish between these two components, even though they remain connected via migration. This has been made more explicit:

“While this gene flow is not apparent from the ADMIXTURE plot (Figure 2), the ancestry is likely grouped into the Khoe-San component which has drifted appreciably from its ancestral Eastern African source. “

We use ADMIXTURE and PCA as a first pass to visualize recent structure among population samples, rather than a mode of inferring history.

Finally, while we included them in the ADMIXTURE graph in Figure 2 to obtain a broad representation of recent structure, there are no Zulu in our inferred moments models. The Nama, Amhara and Oromo sampled here have very little Bantu admixture. Thus, there is no description in our model of western (Bantu) gene flow into South Africa and Ethiopia during the last 2,000 years.

3. In pgs 4/5, when describing the two most likely models, you say “Both allow for migration between stem branches, but differ primarily in the timing of the early divergence of stem populations and their relative N_e (Figure 3).”

a. The parameters of models A and B discussed in the main text presumably correspond to those on Tables S4 and S6 respectively? Can this please be made clear? Assuming that is the case, can the standard error of the timing of Stems 1/2 divergence (1,163ka in A, and 1,442ka in B), and any other divergence date mentioned in the main text, be included please? These are very substantial – A: 1,163,072 \pm 390,803 and B: 1,442,022 \pm 426,449;

Thank you for this comment. We agree and have added the reference to the extended Supp Tables in Table 1. Additionally, we have added the SEs to the specific estimated dates mentioned in the main text as appropriate.

b. This is my ignorance, but why are the standard errors of the Ne estimate of models S4 and S6 so much larger than those of the other 3 models?

This is a great question. In S4 and S6 we allowed migration between the stem branches. Ne, divergence times, and migration rates are all parameters which covary (e.g., Deep divergence with high migration and shallow divergence with low migration can both produce similar summary statistics.) By fixing the stem migration rate to zero in the other models, we removed a source of uncertainty and thus reduced variability in inferred Ne. When migration between stems is allowed, the model is less constrained, and higher SEs for those somewhat confounded parameters is expected.

4. Table 1, Model B, lists the migration between Stems 1 & 2 as occurring for a period of 963Ka – this corresponds to the period between Tstems (1,442ka) to the fracture of stem 1 into 1E and 1S (479ka) – does this imply that stem1 and stem 2 were still in continuous genetic contact at the time of the Neanderthal split (550ka) and subsequent drastic bottleneck of remaining African population? I find it hard to conceive demographically and geographically.

Yes, this model does imply that Stem 1 and 2 were in contact via gene flow throughout this period, including the approximately 550ka divergence of Neanderthals from Stem 1. We acknowledge that the bottleneck following the divergence with Neanderthals is an unusual feature of the merger model. Essentially, when we allowed for a reticulation in Stem 1, the model finds that there must have been extreme drift just preceding the split of Stem 1E and Stem 1S. This suggests that this period in human evolution warrants greater exploration, but we caution against a literal interpretation of this detailed scenario. For this reason, we presented 2 alternate models for consideration and focus on the commonalities.

5. I have an issue with the geography of stem populations:

a. in pg. 6, you say “Mende receive a large additional pulse of gene flow from Stem 2, replacing 18% of their population 25ka.... This may indicate that an ancestral Stem 2 population occupied western or Central Africa, broadly speaking. The differing proportions in the Nama and eastern Africans may also indicate geographic separation of Stem 1S in southern Africa and Stem 1E in eastern Africa”,

i. are these geographic associations really warranted? Can you exclude a model in which all this is happening East Africa, for example Ethiopia and Tanzania, and pan-Africanness of sapiens is the result of expansion in MIS5?

We agree that because of the potential for more recent population movements, there is uncertainty when using the geographic locations of present-day populations to infer the geographic locations of past populations and episodes of gene flow. Our use of the phrase “may indicate” was meant to convey this uncertainty to the reader. We have now changed the first sentence to “Speculatively, this may indicate...” to more clearly convey the uncertainty.

We do believe that such speculative interpretation is warranted. The potential for recent population movements doesn't mean that present-day geographic locations of populations

provide no information about the likely geographic locations of past populations or events. For example, the higher genetic diversity of African populations and the presence of the mtDNA and Y chromosome basal lineages in African populations has long been taken as evidence in support of the Out of Africa model. This conclusion is based on the most parsimonious interpretation of both genetic *and* geographic information from present-day populations. Consider, for example, whether *Homo sapiens* could have evolved in China and radiated back to Africa? The ancestral population in Asia would need to be extinguished with no appreciable descendants in modern Chinese genetic data (because the basal lineages all fall in Africa rather than China).

In our models, Stem 2 contributes disproportionately to West Africans after they have diverged from East Africans 60ka. We find gene flow or a pulse from Stem 2 into West Africans continues during 0-25ka, but remains low or absent in East Africans (e.g., the migration rate from Stem 2 into West Africans is 5x that into East Africans). The finding of much more gene flow from Stem 2 in West Africa would be most parsimoniously explained if Stem 2 was located geographically in West Africa, and we believe that this is a relevant, if speculative, observation.

b. again in the discussion: "During the Middle Stone Age, the multiple merger model indicates three major stem lineages in Africa, tentatively assigned to southern (Stem 1S), eastern (Stem 1E) and western/central Africa (Stem 2)."

i. Besides the fact that, according to the models these stem populations pre-date the MSA (see my comment 6 below), again, attributing geography on the basis of where populations with different genetic proportions are today is not justified

As in our reply to the previous point (point 5), while we agree that there is uncertainty when using the geographic locations of present-day populations to infer the geographic locations of past populations, this doesn't mean that present-day geographic locations of populations provide no information about the likely geographic locations of past populations.

Consider a thought experiment where hominins evolve in only in Eastern Africa. There would need to be sufficient geographic barriers to induce the level of genetic differentiation seen between Stem 1E, 1S and 2. Note that under the merger model, there is no gene flow between Stem 1E and Stem 1S for ~350ka. While not to be taken too literally, this model does indicate strong isolation between these groups during 8 glacial / interglacial cycles from MIS 12-5. Furthermore, Stem 2 would need to be sequestered from Stem 1 for nearly 800ka. While we cannot rule out exceptionally strong geographic barriers *within* East Africa, evidence of fossil hominins across the continent suggest that the species, broadly speaking, were wide-ranging across the continent. Hence, isolation by distance between regions is a more parsimonious scenario.

c. Finally, the section about the ecological riddle "They also help explain an ecological riddle posed by the archaic hominin admixture model. Neanderthal populations were separated from early Homo sapiens by thousands of kilometers and continental geographic barriers. By contrast, an archaic hominin population in Africa would need to have stayed in relative

reproductive isolation from the ancestral human lineage over hundreds of thousands of years despite closer geographic proximity and reproductive compatibility”.

i. I have a real problem with this. Let’s consider for a moment the thousands of km separating Africans from Neanderthals, picking randomly on Nairobi as a point of reference:

- 1. Nairobi – Cape Town: 5,147.5 km*
- 2. Nairobi – Tel Aviv: 5,434.3 km*
- 3. Nairobi – London: 6,817.7 km*
- 4. Nairobi – Dakar: 8,204.0 km*
- 5. Etc*

Given that the distance between East and West Africa is substantially larger than between East Africa and northern Europe, never mind the Levant, and that the ecological barriers between regions of Africa are sufficient to generate biogeographical structure in other mammals, this paragraph does not make sense. On the contrary, if the stem populations were indeed distributed all over Africa, how can hominin diversity 300-200ka be explained? While some geographic structure probably explains the long-standing genetic identity between the stem populations, I believe that defining that geography without ancient genomes can be very misleading and not consistent with other lines of evidence.

As outlined in reply to point 14 of reviewer 1, the purpose of this paragraph was to make connections between our results and ecology, not to make strong statements about geography. So, instead of framing this discussion as an “ecological riddle” and referring to “thousands of kilometers”, we now simply highlight that there are potential ecological implications of preferring weakly-structured-stem models over archaic admixture models:

“Preferring weakly-structured-stem models over archaic-admixture models potentially has ecological implications. With a weakly structured stem, there is no need to posit that an archaic hominin population in Africa stayed reproductively isolated from the ancestral human lineage for tens thousands of years before the initiation of gene flow. Instead, there would simply have been continuous or recurrent contact between two or more groups present in Africa. “

6. My last substantive issue. I found it difficult to assess how meaningful the differences in likelihood are.

- a. S2 – single origin model: LL = -189,434*
- b. S3 – continuous migr, w/o stem migr LL = -126,644*
- c. S4 – continuous migration LL = -115,500*
- d. S5 – merger w/o stem migration LL = -107,652*
- e. S6 – merger w/ stem migration LL = -102,633*

How significant are these differences? In terms of population history, the difference between S5 and S6 is huge, for example. At the end, the models have, by necessity, much uncertainty in the many parameters fixed and estimated, and this is in stark contrast to the certainty/absolute value of the likelihood measure that determines the “best history” amongst clearly many possible histories. I know this is in the nature of testing models mathematically, but perhaps you may consider making it clear that these are hypotheses and not past realities.

We agree and have been agonizing over this question. These differences are highly “significant” from a statistical perspective, in that the more complex model describes the data significantly better than the simpler models according to the usual information criteria. However, this does not tell us that the more complex models are better descriptions of the actual history of populations, since they may be misspecified. A proper accounting for the uncertainty associated with model misspecification would require evaluating the uncertainty over all possible model specifications, which we simply do not know how to do.

This is why we chose the approach of reporting not only the maximum likelihood model, but multiple models fit to the data, and to discuss them side by side. We thought that this would also help avoid people taking our best-fitting model as “the truth”, despite our caveats stated in the slightly expanded first paragraph of the discussion:

“Any attempt at building detailed models of human history is subject to model misspecification. This is true of earlier studies, which often assumed that data inconsistent with a single origin model must be explained by archaic hominin admixture. This is also true of this study. While it remains prohibitive to fully explore the space of plausible models of early human population structure, we sought to capture model uncertainty by exploring multiple parameterizations of early history. The best-fit models presented here include reticulation and migration between early human populations rather than archaic hominin admixture from long-isolated branches. We cannot rule out that more complex models involving additional stems, continuous structure, or hybrid models including both weak structure and archaic hominin admixture may better explain the data. Because parameters related to the split time, migration rates, and relative sizes of the early stems were variable across models, reflecting a degree of confounding among these parameters, we refrained from introducing additional branches associated with more parameters during that period. Rather than interpreting the two stems as representing well-defined and stable populations over hundreds of thousands of years, we interpret the weakly structured stem as consistent with a population coalescence and fragmentation model (Scerri et al. 2019).”

Suggested improvements

Besides the above more substantive issues, I have some suggestions that the authors may want to consider:

1. The second paragraph of the introduction is great, and clearly sets out the problems with previous models/studies and the point of the paper. But the 1st paragraph is really not inspiring....

a. Why are MSA sites significant?

b. The youngest MSA is ~11ka, not 40ka

c. A few (very few!) MSA sites date to 315-130ka, so not equally distributed in time-space

We have revised the 1st paragraph to more clearly delineate that the existing archaeological and paleoanthropological evidence does not necessarily support a single geographic origin model (within Africa) and why we bring up the MSA. We believe that is important to acknowledge the extensive archaeological data from the MSA. We have removed the dates

bracketing the MSA, instead simply indicating that the MSA is coeval with early anatomically modern specimens.

d. Fossils with sapiens traits are not SIMILARLY distributed across the continent – Jebel Irhoud is twice as old as Herto and there are many fossils with no sapiens-derived traits 300ka

We have now described these anatomically derived features as intermittently present in the fossil record (see Introduction).

e. Why are the Kibish fossils not included in the list?

We agree that these fossils are important, but our list wasn't meant to be a comprehensive list, and we had already included other fossils (Herto) from Ethiopia.

f. Unless there is an ancient genome to prove otherwise, all fossils are likely to be dead-ends... Clearly, this par is trying to give some palaeo-context to the study, but it definitely needs a bit of work

We tried to rephrase this sentence to make it clearer that what we mean is that only some fossils represent (sample) populations that contributed genetically to contemporary populations.

2. In pg. 3, you say “We conclude that geographic patterns of contemporary Homo sapiens population structure date back to the late Middle Stone Age in Africa, likely arising during MIS 5.” MIS 5 is not the ‘late’ MSA, but the main period of MSA expansion half-way through the history of the industry

We have revised this sentence to simply refer to the time period rather than the MSA:

“We conclude that geographic patterns of contemporary *Homo sapiens* population structure likely arose during MIS 5.”

3. In the same paragraph, you say “Although we find evidence for earlier population structure in Africa (see below), contemporary populations cannot be easily mapped onto the more ancient ‘stem’ groups as only a small proportion of drift between contemporary populations can be attributed to drift between stems (Figures 4 and S10–S13)” – I think this could do with a bit more explaining; and S10 & S11 suggest that there’s virtually no drift shared with stems?

We agree that explanations were missing! We had a supplementary section explaining this, but a link was missing in the main text. This has been corrected. As for whether Figures S10 and S11 show “only a small proportion of drift” or “virtually no drift”, I suspect that this is a matter of interpretation that depends on our prior expectations. We find that 5 to 10% of contemporary drift between some pairs of populations explained by ancient drift is small, but not entirely negligible.

4. Table 1 of main text, parameter 'c' of merger-model: I think the divergence time be ~95ka instead of ~98ka

This has been corrected, thank you.

5. You end that paragraph in pg 4/5 saying “The two models also differ in the mode of divergence during the Middle Stone Age”. Yet, I don’t see how the models’ estimates can be related to historical events during the last 300ka – Model A has continuous migration between 2 pops (one of which is ancestral to sapiens and Neanderthals) for ~900 ka, decreasing after MIS 5; Model B has continuous migration between 2 pops until ~ 550ka, then this stops, and pop 1 eventually contributes differentially to the Nama and the ancestors of other Africans after MIS 5, and pop 2 gives rise to a second human ancestral lineage and to Neanderthal, the human pop 2 nearly disappears, then splits into 2 ~479ka, 2a + 2b pops evolve independently until they meet/merge variously with descendants of pop 1 during MIS 5. In fact, both models have populations coming together to form the ancestry of living Africans during MIS 5, so do not differ in the mode of divergence during the MSA specifically.

Why not refer to the Middle Pleistocene (780-126 ka) ? The MSA is not a period, but an industry with variable spatial and temporal distribution.

To address this concern, we have modified the subtitle “A Late Middle Stone Age common ancestry for contemporary humans” → “A Late Pleistocene common ancestry for contemporary humans”, removing the use of MSA as a chronological period. The sentence “The two models also differ in the mode of divergence during the Middle Stone Age.” was meant to highlight the lengthy separation and reticulation in the multiple-merger model. We have modified this sentence to read “The two models also differ in the mode of divergence, with the multiple-merger featuring a population reticulation during the Middle Pleistocene.” The reference to MSA on page 8 was changed to Middle Pleistocene. In the discussion, “The Middle Stone Age in Africa” has been retitled to “The Formation of Population Structure in Africa” and the first sentence now reads: “By contrast, our inferred models paint a more consistent picture of the Middle to Late Pleistocene as a critical period of change, assuming that estimates from the recombination clock accurately relate to geological chronologies.”

6. Finally, I found the temporal scale of Figure 4 very misleading – it hides the actual scale of the population processes being described.

This is indeed a drawback of plotting times on logarithmic scales and there has been some debate about this among the co-authors. There is a tradeoff: because we have more power to identify recent events, there is a higher rate of events in recent years in our inferred models, relative to the more ancient past. Plotting on a linear scale leads to collapsing recent events into an unreadable mess. For this reason, the logarithmic time-scale is quite commonly used in population genetics (see, e.g., inverse coalescence rates (PSMC), etc).

Marta Mirazon Lahr

Reviewer Reports on the First Revision:

Referees' comments:

Referee #1 (Remarks to the Author):

Generally I think the authors did a great job responding to each of my and the other reviewers comments, providing useful additional analyses in certain key areas, and at least a fair justification for decisions made in others. The changes they have made to the manuscript were in line with our comments and will improve the manuscript. I just have a few remaining comments.

1. Regarding Line 118 – “Migration, prior to the Bantu expansion, between the ancestors of the Nama and other groups is an order of magnitude weaker than that observed between western and eastern Africans (Table 1).”

This statement is somewhat confusing. Is this also prior to the East African contribution to the Nama? These two migrations into southern Africa are very close to each other. This - while the Bantu expansions in west Africa starts around 4 000 years ago. So, it is not clear what the authors imply here. Is this migration rate into the Nama ancestors before the recent admixture of Nama ancestors with East, West (Bantu) and non Africans?

2. Regarding the low divergence time estimates. The fact that when the authors jointly estimate OoA and W_Afr-E_Afr splits rather than using fixed parameters, their model results in implausibly recent divergence times at clear odds with archaeological evidence (despite taking into account colonial admixture)...suggests to me that underestimation is an underlying feature of this approach. However, I don't think this ought to be a sticking point; I think it is sufficiently transparent if they just include a note of the fact that a wide disparity exists in the literature regarding divergence times, that older estimates exist, and even what they mean by "divergence times" can be interpreted differently in the context of different models (CCr, IM, clean splits, reticulation etc).

Referee #2 (Remarks to the Author):

I would thank the authors for responding to my comments and alleviating some of the major concerns, including the concerns about the ethical clearance for the use of the Nama data. Unfortunately, a couple of the major concerns persist.

There are two major components that the manuscript highlights.

Firstly, the study shows that a model with a weakly structured stem performs better than competing models of early human divergence. However, the difference in likelihood scores though statistically significant is somewhat marginal. More importantly, it is difficult to assess the relevance of these fine differences in discriminating alternative models of human history. Given that the replacement of even the European population, in some cases, impacts the quality of model fitting, it is difficult to

ascertain whether models with more complex scenarios that involve additional populations such as East Asian or alternative Khoe-San or East African proxy could have favored a different model altogether. That said, I think the additional data presented in the revision makes a strong case for this component of the study.

The second major component that this manuscript aims to address is the dating of major splits in the human lineage based on the best-fit model. The new analyses included in the revision i.e., model-based joint estimates for out of Africa migration, East and West African split, 35-40K and 45-55 K years respectively, are much younger than the currently available estimates based on genetic data and fossil records. Therefore, if the estimates for more recent events (such as OAA) provided by the best-fit model are so different, the accuracy of estimates for much older events becomes questionable. It is quite possible that the models in general provide much earlier dates for all events or are not yet fully reliable for dating population splits. One way to do this would have been to use simpler models based on fewer populations and more recent events and calibrate the scale before extending this to deep history. As this could entail a lot of work and be well beyond the scope of the current paper, the discussion on the divergence dates needs to be more nuanced.

On balance, it seems that although the models reported in this study could be valuable tools and pointers for further investigation and discussion, the data and analysis are somewhat limited, raising serious questions about the current timeline and pattern of divergence in the AMH lineage. While the authors have pointed out some of the limitations and presented the caveats, these are scattered along the manuscript and the supplementary materials and leave a lot of room for over or misinterpretation. So, my recommendation to the authors is to include a "limitation" or "considerations for interpretation" section towards the end of the discussion and clarify that the results presented here have a strict operative range in terms of models tested, parameters (including fixed dates for events) and the exact population set used. The lack of availability of additional populations from certain groups (such as Khoe-San) and Central African foragers has not allowed them to thoroughly assess the robustness. Therefore, more comprehensive investigations are required for confirming these results. Also, it would be good to highlight that there are clear contradictions between the divergence dates generated by the model and some of the well-known estimates, and in absence of a further evaluation of the predictivity of these models in independent populations, these estimates need to be interpreted with extreme caution.

Referee #4 (Remarks to the Author):

The manuscript by Ragsdale and colleagues provides novel and important insights into the early human history. They tested and validated various models of early population structure using genome-wide data and statistics leveraging information contained in the linkage disequilibrium decay. With these data and the statistical approach implemented the authors were able to show a deep population structure within Africa, "With continuous migration between stems, population structure extends back to 1.1-1.4Ma" (lines 149-150). Their results also challenge the popular archaic-admixture models in Africa. They propose alternative migration models, which are likely to be more relevant. "Preferring weakly-structured-stem models there is no need to posit that an

archaic hominin population in Africa stayed reproductively isolated from the ancestral human lineage for tens thousands of years before the initiation of gene flow” (lines 270-273).

The analyses are extensive and rigorous. Many important questions have been addressed during the first rounds of reviews regarding the estimations of the parameters. A wide range of analyses have already been performed to validate the demographic model inferences and to carefully explore many aspects of the uncertainty of the estimations. However, I think that some important points need to be further explored based on simulated data (see my major recommendations below).

I have the following major recommendations :

1 Comparing demographic models using composite likelihoods is difficult (Choin et al. 2021, Malaspinas 2016). In this study, the composite likelihood values (Tables S2-S6) seem to be correlated with the complexity of the models (number of parameters). I don't think that the complexity of models drives the main conclusions of the study given the large differences in log-likelihoods (LL) found between the single-origin model (Tables S2) and the two retained models (the continuous-migration and merger-with-stem-migration models, Figure 3, Tables S4 and S6). However, the use of data simulated under each model may be helpful to evaluate the validity of the conclusions (Choin et al. 2021, Malaspinas et al. 2016). One way to assess it would be to generate a set of simulated data under each model (for example drawing combinations of parameters corresponding to the maximum likelihood estimates or within the confidence intervals of the estimations), compute the LLs for each simulated data and check if the highest LL really correspond to the true model. The authors may compute a confusion matrix (see Csilléry et al. 2012 for details, see also Supplementary Figure 48 of Choin et al. 2021) and also provide plots showing the range of variation of the LLs computed on simulated data under each model (see the Supplementary Figure 12 in Choin et al. 2021 or the Supplementary Figure S07.9 in Malaspinas et al. 2016 for an example of such plots).

Finally, for their model comparisons, the authors can estimate the probability that the “true” model is selected, using simulated data as observed data (see the section “model selection” in the Supplementary Information of Choin et al. 2021 for an example of implementation). I think that providing such evaluations may reassure readers. Given the differences in LL indicated in Figure 3, I even think that this analysis may reinforce their conclusions in favor of the merger-with-stem-migration model. Every dataset simulated under this model or under an alternative model should contain enough information to infer the true model in every situation (highest LL systematically obtained under the true simulated model).

*Csilléry et al. *Methods in Ecology and Evolution* (2012) abc: an R package for approximate Bayesian computation (ABC).

*Malaspinas et al. *Nature* (2016) A genomic history of Aboriginal Australia.

*Choin et al. *Nature* (2021) Genomic insights into population history and biological adaptation in Oceania.

2 Model assumptions impact the accuracy of parameter estimations as acknowledged by the authors “Model assumptions have, however, a larger impact on parameter estimates (and thus, real uncertainty)” (p5). Given the complexity of the inferred models a systematic assessment of the

accuracy of the parameter estimations and also an evaluation of the validity of the confidence intervals (CI) deserved to be conducted for each model investigated, or at least for the two models highlighted in main text. This can be easily done by using the simulated data mentioned in the comment 1 above. The author can compare true and estimated values to evaluate the extent of bias and other classically used accuracy index (see the Supplementary Figure 49 in Choin 2021, see also the Supplementary Figure S07.17 in Malaspinas et al. 2016). This new analysis may be connected to the discussions about biased estimations of split times due to model assumptions (Figure S35). In addition, the CIs obtained by bootstrapping data (section 3.3 of the Supporting Information) may provide poor estimations of the uncertainty of the estimations in some circumstances (e.g., strongly biased estimations). Some additional tests should be conducted, at least for the two models highlighted in main text, to check if CIs are large enough to contain true values as expected (“For many parameters confidence intervals based on bootstrapping are relatively narrow (Tables S2-S6)”, p5).

I understand that the recommendations 1 and 2 may represent a substantial amount of work but this kind of validation tests should be done given the importance of the questions addressed by this study. The complexity of the models proposed can be efficiently simulated, see Malaspinas et al. 2016, Choin et al. 2021 for examples. Consistent amount of genetic data (~640Mb of neutral intergenic regions) can be simulated in reasonable computation times to generate the various sets of simulated data required to validate the inferred models and to estimate the accuracy of the parameter estimations.

3 As mentioned above, the authors performed various analyses to explore the uncertainty of the estimations. For example the author initially used fixed parameters and check the effect of this assumption by performing new analyses allowing those parameters to be fit along with all other model parameters (section 6.1.3 of the Supporting Information). Similarly, it seems that the author used fixed migration rates between populations. I have understood that they did not consider migration between Mende (west) or Nama (south) and non-Africans samples. This has been done in all models and should not influence their main conclusions but the likelihoods and the parameter estimations are conditioned on a matrix of recent migration arbitrarily altered by the authors. I think that a new analysis allowing those parameters to be fit along with all other model parameters should also be done.

I have the following minor recommendations :

1 Because the differences between the conceptual models 1A and 1D are unclear I think that a clarification about the rejection of the model 1D, which is mentioned in the Supporting Information (“The recent expansion and the African multi-regional models (Figure 1A and D) have the same topology, so interpretation of the model depends on the specified or inferred divergence times.”), should also appear in main text.

2 The authors performed a wide range of analyses to validate the demographic model inferences (section 6 of the Supporting Information) but, as far as I understand, they only discussed these results (section 7 of the Supporting Information). The authors should systematically show the new likelihoods and parameter estimations obtained. For example, with a table showing all the LLs

obtained for the various re-analyses, and for each re-analysis a table showing the parameter estimations for two the models highlighted in main text (a merge between Tables S4 and S6). The authors can find a more convenient way to summarize their results.

3 The authors replaced the reference population (the Mende population in the current analysis) with another population from West Africa, the Yoruba (YRI) population (section 7.1.1 of the Supporting Information). A similar analysis should be done by replacing the Mende with another population of different geographical origin (the British or from East Africa) to fully reassure readers about the sensitivity to the reference populations used to normalize the statistics.

4 The likelihood estimations are based on a multivariate Gaussian distribution (section 3.2 of the Supporting Information). I think that authors should comment in the Supporting Information the fact that the normalized Hill-Robertson statistics are approximately normally distributed and should quote the tests of normality shown in Ragsdale and Gravel PLoS Genetics (2019). In addition, did the authors perform such tests of normality in the present study? Since they are only approximately normally distributed larger deviations may impact the likelihood estimations in a given model. A short comment on it in the Supporting Information would be helpful for readers.

5 I think that the citations of supplementary figures and tables should be slightly improved in order to help readers easily find relevant information. As an example, the Tables S3, S4, S5 and S6 should be quoted in the sentence "Allowing for continuous migration between the stem populations substantially improved the fits relative to zero migration between stems..." (lines 147-149).

6 The authors should provide the CIs in the Tables S2-S6 in addition to the standard deviations as quoted in the main text "For many parameters confidence intervals based on bootstrapping are relatively narrow (Tables S2-S6)" (p5).

7 The authors should systematically indicate the range of parameters evaluated when the statistics are predicted in the Tables S2-S6.

8 The moments predicted in present day populations depend on unknown initial conditions (Hill and Robertson, Genetics 1968). I may missed it in previous articles (Ragsdale and Gravel PLoS Genetics 2019) but I did not find any information on how the authors deal with it. The authors should add a short notice on it in the Supporting Information.

9 The labels A, B, C, D and E indicated in the legend are missing on the Figure 5.

Referee #5 (Remarks to the Author):

This contribution seeks to clarify the complexities of human origins by specifically testing the hypothesis that archaic 'ghost' populations contributed to African populations in the Pleistocene. It is well accepted that an alternative scenario for this can be found within population structure, but this paper seeks to understand this possibility better by testing a range of possible scenarios. I think

this is an interesting and valid effort, that is well written and methodologically significant in my view.

The issues I had with the paper were already picked up by the previous reviewers and I am satisfied with the responses. This includes the need to discuss the fixed parameters more explicitly, and for example, state why the time divergence between eastern and western African populations fixed at 60 ka. I think the authors did a good job at addressing other reviewer concerns. In fact I think they should include some parts of their rebuttal in the paper, e.g. the interesting rebuttal to reviewer 2 on geographic origins perhaps being in East Africa and then spreading, for example. In sum, I think this paper is innovative enough and is worthy of publication, but I have a few minor comments below.

Minor comments:

Small section attempting to fit results with the fossil record needs re-writing.

This is just a model, don't overinterpret based on highest likelihood especially when a major component of African ancestry (from central African foragers) is missing from the study – aren't these the populations that have previously been linked to archaic admixture?

Introduction Page 1 line 41-42. It isn't quite true to say that derived features were 'intermittently' present. It is more correct to say that they appear in different combinations with archaic features until some time between 100 and 40 thousand years ago. The fossil record is sparse, but derived features don't come and go as the word intermittent suggests.

Page 3 line 112: what are these archaeological studies? Some examples are required as citations here.

Following on, re. the time divergence between eastern and western African populations fixed at 60 ka - it would be interesting here to cite some of the archaeological literature that sees a stable and somewhat isolated West Africa from around 62-11 ka (i.e. Niang et al, 2020; Scerri et al., 2021).

Page 10 lines 254-257: suggest you delete this. Glacial/interglacial cycles has dramatic effects at the extremes of Africa but were not the dominant drivers for most of Africa and its interior (see Kaboth-Bahr et al., 2021 PNAS). I don't think it is feasible to try to explain/interpret all the model results in the discussion.

Page 10 lines 288-295: this needs to be rewritten and requires more circumspection. The human fossil record is so sparse, that many scenarios could be said to fit with it. Omo Kibish, has a modern braincase, but in other aspects is well beyond the range of variation of contemporary humans. Jebel Irhoud has a more modern face, but a longer, flatter braincase. I am not sure why the authors then jump to Homo naledi which is a completely different hominin species. If this is done to suggest that naledi could not have been a source of ancestry, I understand, but it is written somewhat awkwardly here. In any case, other large brained hominins were in Africa at the time like heidelbergensis which seems to be much more of a candidate for possible admixture than naledi, which has not even been associated with stone tools at this point, and a small cranial capacity. I see that another reviewer raised this. While the authors respond well in the response to reviewers, I don't see enough of these changes in the main text. In particular the authors state that "Without detailed morphological analyses, which seemed out of scope, it is difficult to say if fossils from Jebel Irhoud or

Kabwe/Broken Hill could have come from populations that were part of the weakly structured stem.”. This is a little disingenuous since there is no suggestion that Kabwe and Jebel Irhoud were the same species. There is broad agreement that Jebel Irhoud as a fossil at the root of *H. sapiens* and that Kabwe represents *heidelbergensis*. Detailed morphological assessments already exist and could easily be cited.

Author Rebuttals to First Revision:

Reviewer #1:

Generally I think the authors did a great job responding to each of my and the other reviewers comments, providing useful additional analyses in certain key areas, and at least a fair justification for decisions made in others. The changes they have made to the manuscript were in line with our comments and will improve the manuscript. I just have a few remaining comments.

1. Regarding Line 118 – “Migration, prior to the Bantu expansion, between the ancestors of the Nama and other groups is an order of magnitude weaker than that observed between western and eastern Africans (Table 1).”

This statement is somewhat confusing. Is this also prior to the East African contribution to the Nama? These two migrations into southern Africa are very close to each other. This - while the Bantu expansions in west Africa starts around 4 000 years ago. So, it is not clear what the authors imply here. Is this migration rate into the Nama ancestors before the recent admixture of Nama ancestors with East, West (Bantu) and non Africans?

The migrations are referring to the period ~60ka to 5ka, prior to more recent expansions and admixture events in the past 5000 years (e.g., Bantu expansion and East African contribution to the Nama). We have clarified this section of the text.

2. Regarding the low divergence time estimates. The fact that when the authors jointly estimate OoA and W_Afr-E_Afr splits rather than using fixed parameters, their model results in implausibly recent divergence times at clear odds with archaeological evidence (despite taking into account colonial admixture)...suggests to me that underestimation is an underlying feature of this approach. However, I don't think this ought to be a sticking point; I think it is sufficiently transparent if they just include a note of the fact that a wide disparity exists in the literature regarding divergence times, that older estimates exist, and even what they mean by "divergence times" can be interpreted differently in the context of different models (CCr, IM, clean splits, reticulation etc).

Thank you for this suggestion. To summarize our answer, we do not believe that underestimation is a systematic feature of the approach. We have clarified Section S7.2 regarding the description of fixed and free parameters. We have included a note now in the *Discussion* regarding “divergence time” and the differing assumptions of various methods (please see below).

With regard to underestimation of divergence dates, when we allowed the out-of-Africa expansion and Eastern/Western African divergence times to be jointly fit in model optimization to real data, we found that the best-fit inferred E/W divergence time was 45 – 55ka and the OOA was 38ka. We agree that these divergence times are likely too recent. When OOA is fixed at 50ka, the estimated E/W divergence is found to be 61ka with moments.LD. This result was replicated using both LD and SFS based inference with a different set of populations, and is similar to other estimates from the published literature (S3.1.1). This suggests that model uncertainty around the OOA is driving the under-estimation of our earlier E/W divergence result. Importantly, whether these two events (E/W divergence and OOA) were fixed or allowed to be fit during optimization had only a modest impact on other inferred parameters in our models. There are a multitude of reasons why the OOA event may be poorly parameterized here (and in other genetic papers). We utilize only a single non-African population (GBR) for inference rather than 2-3, and gene flow across Eurasia may impact estimates. Prior papers tend to focus on the divergence between West Africans vs. Eurasians as a proxy for the OOA, while we include 3 Eastern African populations (Gumuz / Amhara / Oromo), which provide a better source for the ancestral OOA event but lack a population from the Near East to account for accurate back-to-Africa gene flow. In summary, we believe model mis-specification related to the OOA is particularly problematic and beyond the scope of this paper. However, this single parameter has limited impact on other features of the model (Section S3.1.1).

Discussion of the requested clarification is included at the end of “The Formation of Population Structure in Africa” section of the main text (where we have now added your point regarding interpretation of different analyses):

“Population divergence” in population genetics is a complex concept due to the co-estimation of divergence time and subsequent migration; methods assuming clean vs reticulated splits can infer different split dates (Figure S28 and Figure S36). Therefore, wide variation exists in estimates of divergence time in the literature^{30,17}.

Reviewer #2:

I would thank the authors for responding to my comments and alleviating some of the major concerns, including the concerns about the ethical clearance for the use of the Nama data. Unfortunately, a couple of the major concerns persist.

There are two major components that the manuscript highlights.

Firstly, the study shows that a model with a weakly structured stem performs better than competing models of early human divergence. However, the difference in likelihood scores though statistically significant is somewhat marginal. More importantly, it is difficult to assess the relevance of these fine differences in discriminating alternative models of human history. Given that the replacement of even the European population, in some cases, impacts the quality of model fitting, it is difficult to ascertain whether models with more complex scenarios that involve additional populations such as East Asian or alternative Khoe-San or East African proxy could have favored a different model altogether. That said, I think the additional data presented in the revision makes a strong case for this component of the study.

We hope that we can assuage the Reviewer 2's lingering concerns here. We would not describe the likelihood differences as 'somewhat marginal', and agree with Reviewer 4 who states: "I don't think that the complexity of models drives the main conclusions of the study given the large differences in log-likelihoods (LL)" (please see below). To make this discussion more quantitative, we have added new simulations where we simulated hundreds of datasets under three different models and quantified our ability to recover the correct model. We found that the correct models were recovered every time, and that the scale of likelihood differences between proposed models corresponded roughly to what we have observed in the data. This confirms that the likelihood differences are at least large enough to strongly justify model selection.

As with any likelihood-based analysis, we are still sensitive to model misspecification. While swapping out populations (such as the TSI for the GBR) does change the parameterization of divergence dates among contemporary groups, this does not result in a change in the favored model type. We have not exhaustively searched all of model space, as we try to stress in the discussion – the complexity of such an effort is quite difficult, but we can confidently say that among the 4 major model types we explored, both a single origin and archaic admixture were poor fits to the data. Swapping out additional groups is unlikely to change the two robust features we newly describe: 1) a long-term Pleistocene weakly structured stem and 2) reticulation among the ancestral groups leading up to the population divergences (either via continuous gene flow or punctuated merger events.)

The second major component that this manuscript aims to address is the dating of major splits in the human lineage based on the best-fit model. The new analyses included in the revision i.e., model-based joint estimates for out of Africa migration, East and West African split, 35-40K and 45-55 K years respectively, are much younger than the currently available estimates based on genetic data and fossil records. Therefore, if the estimates for more recent events (such as OAA) provided by the best-fit model are so different, the accuracy of estimates for much older events becomes questionable. It is quite possible that the models in general provide much earlier dates for all events or are not yet fully reliable for dating population splits. One way to do this would have been to use

simpler models based on fewer populations and more recent events and calibrate the scale before extending this to deep history. As this could entail a lot of work and be well beyond the scope of the current paper, the discussion on the divergence dates needs to be more nuanced.

The recent estimated split time for the OOA event does warrant a bit more discussion, and we have included more analyses (see also reply to reviewer 1). To summarize these results and discussions:

- Ragsdale and Gravel (2019) showed that the LD statistics were unbiased in simulated models.
- As part of the validation for the moments software, we performed extensive validations including testing that parameter estimates were unbiased in simple simulated models (e.g., IM). However, we did not publish these results. To address this concern in the context of this paper, we performed additional analyses as follows.
- Using simulated data: When inferring parameters using the same model that was used to simulate data, we find unbiased parameter estimates and well-calibrated confidence intervals (Section 7.5).

Therefore, we are highly confident that the LD method is not systematically biased unless there is model misspecification. To assess whether the LD based method is particularly sensitive to model misspecification, we performed inferences based on identical misspecified models using both SFS and LD inference on real data.

- We used the Gutenkunst et al (2009) parameterization of the three-population out-of-Africa model. We separately fit it using the SFS and using the LD statistics from West African (MSL), European (GBR), and East Asian (CHB) populations to estimate the West African-Eurasian divergence time. These were inferred to be 65ka (SFS) and 55ka (LD) (Supplement section S3.1.1), consistent with our previous estimate of 60ka using LD statistics from a different set of populations (YRI, CEU, and CHB, in Ragsdale & Gravel, 2019). Because this is real data, the “correct” split time is of course unknown.

To summarize, we agree that the apparent lack of concordance with expectations from the archaeological record remains an important unresolved issue, and we have added discussion as suggested (S3.1.1). However, we do not believe that the recent OOA dates inferred by our model support a systematic bias that is specific to our method. We believe that our simulation analyses included in the supplement (Section S7.4) provide a compelling argument for why most split dates in reticulated models are more recent than split dates in simpler isolation-with-migration models.

On balance, it seems that although the models reported in this study could be valuable tools and pointers for further investigation and discussion, the data and analysis are somewhat limited, raising

serious questions about the current timeline and pattern of divergence in the AMH lineage. While the authors have pointed out some of the limitations and presented the caveats, these are scattered along the manuscript and the supplementary materials and leave a lot of room for over or misinterpretation. So, my recommendation to the authors is to include a “limitation” or “considerations for interpretation” section towards the end of the discussion and clarify that the results presented here have a strict operative range in terms of models tested, parameters (including fixed dates for events) and the exact population set used. The lack of availability of additional populations from certain groups (such as Khoe-San) and Central African foragers has not allowed them to thoroughly assess the robustness. Therefore, more comprehensive investigations are required for confirming these results. Also, it would be good to highlight that there are clear contradictions between the divergence dates generated by the model and some of the well-known estimates, and in absence of a further evaluation of the predictivity of these models in independent populations, these estimates need to be interpreted with extreme caution.

The first paragraph of the *Discussion* contains our major cautionary notes, though with a focus on adjudicating between archaic admixture and alternative models. As the reviewer points out, our discussion is well-caveated. But instead of a dedicated section containing a single list of caveats covering all discussion points, we think it is most helpful for readers to be presented with limitations and caveats near the specific interpretation.

We have also revised the following text as the last sentence of the *Discussion* “Additional African populations such as those from Central Africa, other Khoe-San groups, or pre-Holocene ancient DNA samples, could further test our proposed models.”

With regard to uncertainty related to the fixed versus free OOA event, we have further added the following sentence to the Discussion and references to Supplemental Sections 6 and 7 which discuss our validation attempts:

We performed a variety of validation tests to explore sensitivity of our assumptions, including relaxing fixed parameters (SI Section 6). Most validation tests resulted in parameters similar to the models discussed above. However, one exception was the inferred Out of Africa and eastern/western African divergences which were 10-15ka younger than our fixed parameters. The complexity of the OOA expansion warrants the inclusion of additional Eurasian populations and more detailed modeling. However, other features of our inferred models are weakly affected by this analysis choice.

Finally, with regards to the discordant divergence estimates, we have added a paragraphs which reads:

However, interpreting population divergence times in population genetics is always challenging due to the co-estimation of divergence time and subsequent migration; methods assuming clean vs reticulated splits can infer different split dates (Figures S28 and S36). Therefore, wide variation exists in estimates of divergence time in the literature[30,17].

We believe that the manuscript now provides adequate caveats.

Reviewer #4:

The manuscript by Ragsdale and colleagues provides novel and important insights into the early human history. They tested and validated various models of early population structure using genome-wide data and statistics leveraging information contained in the linkage disequilibrium decay. With these data and the statistical approach implemented the authors were able to show a deep population structure within Africa, “With continuous migration between stems, population structure extends back to 1.1-1.4Ma” (lines 149-150). Their results also challenge the popular archaic-admixture models in Africa. They propose alternative migration models, which are likely to be more relevant. “Preferring weakly-structured-stem models there is no need to posit that an archaic hominin population in Africa stayed reproductively isolated from the ancestral human lineage for tens thousands of years before the initiation of gene flow” (lines 270-273).

The analyses are extensive and rigorous. Many important questions have been addressed during the first rounds of reviews regarding the estimations of the parameters. A wide range of analyses have already been performed to validate the demographic model inferences and to carefully explore many aspects of the uncertainty of the estimations. However, I think that some important points need to be further explored based on simulated data (see my major recommendations below).

We thank the reviewer for recognizing the rigor and scope of our analyses!

I have the following major recommendations :

1 Comparing demographic models using composite likelihoods is difficult (Choin et al. 2021, Malaspina 2016). In this study, the composite likelihood values (Tables S2-S6) seem to be correlated with the complexity of the models (number of parameters). I don't think that the complexity of models drives the main conclusions of the study given the large differences in log-likelihoods (LL)

found between the single-origin model (Tables S2) and the two retained models (the continuous-migration and merger-with-stem-migration models, Figure 3, Tables S4 and S6). However, the use of data simulated under each model may be helpful to evaluate the validity of the conclusions (Choin et al. 2021, Malaspinas et al. 2016). One way to assess it would be to generate a set of simulated data under each model (for example drawing combinations of parameters corresponding to the maximum likelihood estimates or within the confidence intervals of the estimations), compute the LLs for each simulated

data and check if the highest LL really correspond to the true model. The authors may compute a confusion matrix (see Csilléry et al. 2012 for details, see also Supplementary Figure 48 of Choin et al. 2021) and also provide plots showing the range of variation of the LLs computed on simulated data under each model (see the Supplementary Figure 12 in Choin et al. 2021 or the Supplementary Figure S07.9 in Malaspinas et al. 2016 for an example of such plots).

Finally, for their model comparisons, the authors can estimate the probability that the “true” model is selected, using simulated data as observed data (see the section “model selection” in the Supplementary Information of Choin et al. 2021 for an example of implementation). I think that providing such evaluations may reassure readers. Given the differences in LL indicated in Figure 3, I even think that this analysis may reinforce their conclusions in favor of the merger-with-stem-migration model. Every dataset simulated under this model or under an alternative model should contain enough information to infer the true model in every situation (highest LL systematically obtained under the true simulated model).

*Csilléry et al. *Methods in Ecology and Evolution* (2012) abc: an R package for approximate Bayesian computation (ABC).

*Malaspinas et al. *Nature* (2016) A genomic history of Aboriginal Australia.

*Choin et al. *Nature* (2021) Genomic insights into population history and biological adaptation in Oceania.

Thank you for the suggestion. As you have suggested, we simulated 500 datasets for 3 primary models (the single origin, continuous migration, and merger with stem migration models). Each dataset is composed of 500 x 1Mb regions. We have added a confusion matrix with the computed LLs to Supplement (Table S1). We do not find that more complex (more parameters) models have a better LL for all datasets. The single origin model is nested within both of the more complex models, by setting migration from the additional stem to zero. In this case, the more complex models can always do at least as well as the simpler single origin model in terms of likelihood. We explored whether the more complex models in practice do better than the single origin model by allowing it to search parameter space. However, these optimizations resulted in the more complex model either converging to the nested single origin model (in which case it had equivalent LL) or they found a local maximum with worse LL than the single origin model. Thus when simulating under the simpler models, the simpler model was always selected. From this, we conclude that the more

complex models are not overfitting statistical noise in the data and thus, we are not getting better likelihoods with more complex models because of the increase in the number of free parameters.

The simulated confusion matrix was perfect for the 1500 simulations we performed, with the correct model selected every time:

Table S1: **Model choice from fits to simulated data.** We simulated three models, based on the best fit parameters from fits to data. Using these simulated datasets, we reinferrd four models that we tested in our analyses in the main text, including simple (single origin) and complex models. Among 500 fits, we exclusively recover the simulated model, even when fitting more complex (i.e., more heavily parameterized) models to simpler simulated models (such as the single origin model).

Simulated model↓	Single origin	Cont. migration	Merger w/out stem mig.	Merger w/stem mig.
Single origin	500	0	0	0
Cont. migration	0	500	0	0
Merger w/stem mig.	0	0	0	500

We provide additional figures as suggested by the reviewer, which indeed shows that the differences in simulated log-likelihoods when simulating under the merger with stem migrations are consistent with what we find in the real data.

Figure S5: **Log-likelihoods of fits of four models to simulated data.** In three scenarios, a simple single origin model and our preferred continuous migration and merger with stem migration models, we simulated 500 replicates of 500 Mb with the same populations and sample sizes from our reported best fit models. We then fit the four models reported in the main text to each of these simulated datasets and compared log-likelihoods. In each case, the simulated model is correctly chosen, even under the single origin scenario. We conclude our more complex models (with more parameters) are not simply fitting statistical noise.

Indeed, as described above, we find that the highest LL for each of the 500 simulated datasets is provided by the reinferrd simulated model, instead of another model (including a more complex model). Section S7.7 details these analyses and findings.

We agree that these analyses strengthen our conclusions and thank the reviewer for the suggestion.

2. Model assumptions impact the accuracy of parameter estimations as acknowledged by the authors “Model assumptions have, however, a larger impact on parameter estimates (and thus, real uncertainty)” (p5). Given the complexity of the inferred models a systematic assessment of the accuracy of the parameter estimations and also an evaluation of the validity of the confidence intervals (CI) deserved to be conducted for each model investigated, or at least for the two models highlighted in main text. This can be easily done by using the simulated data mentioned in the comment 1 above. The author can compare true and estimated values to evaluate the extent of bias and other classically used accuracy index (see the Supplementary Figure 49 in Choin 2021, see also the Supplementary Figure S07.17 in Malaspinas et al. 2016). This new analysis may be connected to the discussions about biased estimations of split times due to model assumptions (Figure S35). In addition, the CIs obtained by bootstrapping data (section 3.3 of the Supporting Information) may provide poor estimations of the uncertainty of the estimations in some circumstances (e.g., strongly biased estimations). Some additional tests should be conducted, at least for the two models highlighted in main text, to check if CIs are large enough to contain true values as expected (“For many parameters confidence intervals based on bootstrapping are relatively narrow (Tables S2-S6)”, p5).

We generated 200 bootstrapped datasets for our two highlighted (complex) models, and performed parameter fit for each of these bootstrapped datasets. We first used this to confirm that our approach almost always selected the best model in this setting (see Reviewer 4, point 1). We also used these simulations to confirm that the true simulated values are included in the 95% confidence interval 94.7% (126/133) of the time, as expected. In fact, we found that the confidence intervals we initially reported, using the Godambe Information + bootstrap approach, were overly conservative (i.e., a bit too wide). This is especially noticeable when a parameter is near the bound of the allowed parameter search space. In such cases, the Godambe Information matrix, because it is based on the local curvature of the log-likelihood function, does not know about the bound, and therefore allows the parameter to take a wider range of values relative to a standard block bootstrap where the bound is enforced in each replicate fit. For consistency, we now report the block bootstrap results throughout.

With this confirmation that our confidence intervals are well calibrated and the bias modest under the correct model specification, we were able to go beyond and provide some additional insight about the sensitivity of recent parameters to model misspecification. While the deeper history among our tested models differed in parameterization, the recent history is similarly parameterized (such as population sizes and migration rates over the past 100 thousand years), and we could evaluate the consistency of equivalent parameters across the different inferred models. As shown in Figure S39 and discussed in Section S7.6, these parameters largely overlap in their bootstrap CIs, with the exception of the single origin model for some of those parameters, such as the population size of the Mende (this is the model that also provides a worst fit to the data, and is likely highly misspecified). This shows again that model mis-specification poses a greater challenge in interpretation than estimation uncertainty, which confidence intervals are meant to describe. We are careful to alert the reader to this throughout.

I understand that the recommendations 1 and 2 may represent a substantial amount of work but this kind of validation tests should be done given the importance of the questions addressed by this study. The complexity of the models proposed can be efficiently simulated, see Malaspinas et al. 2016, Choin et al. 2021 for examples. Consistent amount of genetic data (~640Mb of neutral intergenic regions) can be simulated in reasonable computation times to generate the various sets of simulated data required to validate the inferred models and to estimate the accuracy of the parameter estimations.

We agree and find that this suggestion helped strengthen our findings. Thanks!

3 As mentioned above, the authors performed various analyses to explore the uncertainty of the estimations. For example the author initially used fixed parameters and check the effect of this assumption by performing new analyses allowing those parameters to be fit along with all other model parameters (section 6.1.3 of the Supporting Information). Similarly, it seems that the author used fixed migration rates between populations. I have understood that they did not consider migration between Mende (west) or Nama (south) and non-Africans samples. This has been done in all models and should not influence their main conclusions but the likelihoods and the parameter estimations are conditioned on a matrix of recent migration arbitrarily altered by the authors. I think that a new analysis allowing those parameters to be fit along with all other model parameters should also be done.

In preliminary fits, we included all possible pairwise migrations between populations in our model. We found that some migration rates were very small (orders of magnitude smaller than others, with scaled rates $2*Ne*m$ much less than 1). We chose to fix these migration rates to zero, to reduce the total number of parameters needing to be fit. The Nama-Mende and Nama-British migrations were two such low migration rates.

Following the reviewer suggestions, we have re-included them in model fits, to verify that they are indeed much smaller than other rates, and do not strongly impact expected statistics. As detailed in Supp. Materials section S7.8, these migration rates were very small ($2E-6$ - $5E-6$), with $2Nm$ approximately 0.05 (much less than 1). These fits are now included in the supplementary materials, and we confirmed that their inclusion does not impact other inferred parameters in the reported models.

I have the following minor recommendations :

1 Because the differences between the conceptual models 1A and 1D are unclear I think that a clarification about the rejection of the model 1D, which is mentioned in the Supporting Information (“The recent expansion and the African multi-regional models (Figure 1A and D) have the same topology, so interpretation of the model depends on the specified or inferred divergence times.”), should also appear in main text.

We have now added this clarification when introducing the models in the main text.

2 The authors performed a wide range of analyses to validate the demographic model inferences (section 6 of the Supporting Information) but, as far as I understand, they only discussed these results (section 7 of the Supporting Information). The authors should systematically show the new likelihoods and parameter estimations obtained. For example, with a table showing all the LLs obtained for the various re-analyses, and for each re-analysis a table showing the parameter estimations for two the models highlighted in main text (a merge between Tables S4 and S6). The authors can find a more convenient way to summarize their results.

We have chosen to report all these values using the standardized Demes format (Gower et al, 2022). Because the demes format has clear specifications about how parameters should be stored and defined, this avoids a common source of error and confusion. These are shared as a supplementary zipped directory. As an added convenience, this format is directly readable by many simulation tools (including moments and msprime, used in this study). We describe each analysis and data that each model was fit to (and reference the associated section in the Supporting Information) in the Readme, and log-likelihoods are given in the “description” field for each model in the Demes format.

3 The authors replaced the reference population (the Mende population in the current analysis) with another population from West Africa, the Yoruba (YRI) population (section 7.1.1 of the Supporting Information). A similar analysis should be done by replacing the Mende with another population of different geographical origin (the British or from East Africa) to fully reassure readers about the sensitivity to the reference populations used to normalize the statistics.

Thank you for the suggestion. We have swapped the Mende with the eastern African Gumuz as the reference population used to normalize the statistics. We see no changes in the inferred parameters when we do so. The results of these fits (along with every other supplementary analysis) are included as Demes-formatted demographic models in a supplementary zipped directory (see also point 2)

4 The likelihood estimations are based on a multivariate Gaussian distribution (section 3.2 of the Supporting Information). I think that authors should comment in the Supporting Information the fact that the normalized Hill-Robertson statistics are approximately normally distributed and should quote the tests of normality shown in Ragsdale and Gravel PLoS Genetics (2019). In addition, did the authors perform such tests of normality in the present study? Since they are only approximately normally distributed larger deviations may impact the likelihood estimations in a given model. A short comment on it in the Supporting Information would be helpful for readers.

We did not separately test for normality in this manuscript, but have added text as suggested to point the reader to the 2019 publication:

moments-LD uses a composite multivariate Gaussian likelihood approach to simultaneously fit relative pairwise diversity and LD statistics over a range of recombination distances. This composite likelihood and Gaussian assumption are described in more detail and validated in Ragsdale and Gravel (2019), but we outline them here. To compute likelihoods, we need an estimate of the joint distribution of summary statistics. We estimate uncertainty due to the finite amount of genetic material used in inference using bootstrap over 500 segments along the genome with roughly equal lengths of retained sequences within each segment. First, for each distance bin, we use these bootstrap samples to obtain a variance-covariances matrix across all statistics. This variance-covariance matrix is used to obtain a model likelihood for each recombination distance bin and single-locus nucleotide diversity, as a multivariate Gaussian likelihood. The full model likelihood is taken as the product of likelihoods over each bin. In other words, we followed the approach of Ragsdale and Gravel (2019) in optimizing a composite likelihood where observations in different bins were taken to be independent.

5 I think that the citations of supplementary figures and tables should be slightly improved in order to help readers easily find relevant information. As an example, the Tables S3, S4, S5 and S6 should be quoted in the sentence “Allowing for continuous migration between the stem populations substantially improved the fits relative to zero migration between stems...” (lines 147-149).

Thank you for the suggestion. We now more frequently point readers to the relevant supplementary sections, tables, and figures (including the example here).

6 The authors should provide the CIs in the Tables S2-S6 in addition to the standard deviations as quoted in the main text “For many parameters confidence intervals based on bootstrapping are relatively narrow (Tables S2-S6)” (p5).

We previously reported standard errors, estimated using the Godambe Information Matrix approach (which we have replaced with direct estimates from bootstrap replicate datasets). The 95% confidence intervals are now included in Tables S2-S6.

7 The authors should systematically indicate the range of parameters evaluated when the statistics are predicted in the Tables S2-S6.

In Section S3 (Optimization using moments), we now indicate the imposed bounds on all parameters:

We imposed bounds on many of the parameters in the fits. This is done primarily to avoid optimisation exploring parts of the parameter space that are unrealistic (e.g., negative migration rates), where the model calculation can be costly (e.g., population splits billions of years ago), and where model assumptions can break down (e.g., very small populations). Population sizes had a lower bound of $N_e = 100$, migration rates were bounded between 0 and 10^{-3} , and admixture proportions between 0 and 1. Event times were only constrained to occur in the proper order (so that populations were required to have positive existence times, for example). Most best-fit parameters were away from these boundaries.

8 The moments predicted in present day populations depend on unknown initial conditions (Hill and Robertson, Genetics 1968). I may missed it in previous articles (Ragsdale and Gravel PLoS Genetics 2019) but I did not find any information on how the authors deal with it. The authors should add a short notice on it in the Supporting Information.

We assume that the ancestral population is at mutation-recombination-drift equilibrium given its population size. This is now made more explicit in the supplement:

As is commonly done in demographic inference, all models we considered are initialized from an ancestral panmictic population at mutation-recombination-drift equilibrium. This allows for rapid initialization of the summary statistics to reasonable values. The size N_e of this ancestral population is a model-specific adjustable parameter that also serves as a reference size to define subsequent population size changes.

9 The labels A, B, C, D and E indicated in the legend are missing on the Figure 5.

Thank you for pointing this out. This has been fixed.

Referee #5 (Remarks to the Author):

This contribution seeks to clarify the complexities of human origins by specifically testing the hypothesis that archaic 'ghost' populations contributed to African populations in the Pleistocene. It is well accepted that an alternative scenario for this can be found within population structure, but this paper seeks to understand this possibility better by testing a range of possible scenarios. I think this is an interesting and valid effort, that is well written and methodologically significant in my view.

We are glad that the reviewer found the work interesting and significant! We agree that it is well understood, in principle, that population structure and archaic admixture can be difficult to distinguish. We do find, however, that previous genetics discussion has almost exclusively focused on the archaic admixture scenario. Previous papers (including our own!) tend to make rather strong claims in favour of archaic admixture:

Plagnol et al (2006)

Abstract: "Here we present a new method for addressing whether archaic human groups contributed to the modern gene pool (called ancient admixture), we find strong evidence for ancient admixture in both a European and a West African population"

Hammer et al (2011): Title: "Genetic evidence for archaic admixture in Africa"

Abstract: "allow us to infer that contemporary African populations contain a small proportion of genetic material (2%) that introgressed 35 kya from an archaic population that split from the ancestors of anatomically modern humans 700 kya."

Ragsdale et al (2019) Abstract: "We estimate that an unidentified, deeply diverged population admixed with modern humans within Africa both before and after the split of African and Eurasian populations"

Lorente Galdos et al (2019): Title: "[A pan-African panel] reveals archaic gene flow"

Durvasula et al (2020) Title: "Recovering signals of ghost archaic introgression"

Abstract: "Our analyses of site frequency spectra indicate that these populations derive 2 to 19% of their genetic ancestry from an archaic population that diverged before the split of Neanderthals and modern humans."

Thus the current literature overwhelmingly interprets the available evidence as supporting archaic admixture. We can speculate about why such an emphasis is put on this particular tantalizing scenario, but this suggests to us that the alternative scenario of weak population structure is not widely recognized as a viable model, let alone one that better describes the data. We also do agree that testing a range of scenarios was helpful in refining our understanding of this period.

The issues I had with the paper were already picked up by the previous reviewers and I am satisfied with the responses. This includes the need to discuss the fixed parameters more explicitly, and for example, state why the time divergence between eastern and western African populations fixed at 60 ka. I think the authors did a good job at addressing other reviewer concerns. In fact I think they should include some parts of their rebuttal in the paper, e.g. the interesting rebuttal to reviewer 2 on geographic origins perhaps being in East Africa and then spreading, for example. In sum, I think this paper is innovative enough and is worthy of publication, but I have a few minor comments below.

Thank you for your close reading of our previous replies and your additional suggestions. We have added context to "The Formation of Population Structure in Africa" section of the Discussion to clarify the assignment of geographic regions to ancestral events.

Minor comments:

Small section attempting to fit results with the fossil record needs re-writing.

This is just a model, don't overinterpret based on highest likelihood especially when a major component of African ancestry (from central African foragers) is missing from the study – aren't these the populations that have previously been linked to archaic admixture?

Several publications that argued for substantial archaic admixture in Africa (e.g., Hammer 2012, Ragsdale 2019, Durvasula 2020) considered West African populations (usually a single population). We have demonstrated that this signal is due to model misspecification, and the observed patterns do not strongly support archaic admixture. In papers that analyzed both West African populations and Central African populations (e.g., Hey et al, 2018), signal for ghost admixture was always strongest in the West African populations. While central African foragers are missing from our dataset, there is no reason *a priori* to believe that the signature of archaic admixture would be unique to these groups.

Regarding overinterpretation, prior papers supporting archaic admixture do so via choice of model likelihood. The absence of other classes of models from these tests, such as reticulation or a weakly structured stem, simply reinforces our cautionary discussion that demographic inference is sensitive to model misspecification. For example, our discussion starts with the caveat:

Any attempt at building detailed models of human history is subject to model misspecification. This is true of earlier studies, which often assumed that data inconsistent with a single-origin model must be explained by archaic hominin admixture. This is also true of this study. While it remains prohibitive to fully explore the space of plausible models of early human population structure, we sought to capture model uncertainty by exploring multiple parameterizations of early history...

In the section referred by the reviewer, we also accounted for this uncertainty by making the discussion conditional on our model being correct:

“If, as our model predicts, the genetic differences between the stems were comparable to those among contemporary human populations, the most morphologically divergent fossils are unlikely to represent branches that contributed appreciably to contemporary human ancestries.”

What we tried to do is simply to emphasize a difference in interpretation between previously proposed models (with archaic admixture) and the present model (with weaker structure). While all models come with caveats, we find that researchers are already speculating about who the ghost contributors suggested by previous genetics models might be, with all publications we have seen so far referring to these as archaic. We point out that a weakly structured stem, which has better support from the data, makes different predictions about who these contributors might be compared with what has been previously discussed.

Introduction Page 1 line 41-42. It isn't quite true to say that derived features were 'intermittently' present. It is more correct to say that they appear in different combinations with archaic features until some time between 100 and 40 thousand years ago. The fossil record is sparse, but derived features don't come and go as the word intermittent suggests.

We removed “intermittently” from this sentence and revised it accordingly.

Page 3 line 112: what are these archaeological studies? Some examples are required as citations here.

Following on, re. the time divergence between eastern and western African populations fixed at 60 ka - it would be interesting here to cite some of the archaeological literature that sees a stable and somewhat isolated West Africa from around 62-11 ka (i.e. Niang et al, 2020; Scerri et al., 2021).

We have added a reference to SI Section 3.1, where the details of these citations are listed by fixed parameter.

Page 10 lines 254-257: suggest you delete this. Glacial/interglacial cycles has dramatic effects at the extremes of Africa but were not the dominant drivers for most of Africa and its interior (see Kaboth-Bahr et al., 2021 PNAS). I don't think it is feasible to try to explain/interpret all the model results in the discussion.

Thank you for the clarification request, we have rewritten this sentence and provided additional context to the interglacial cycle 5e remark. We do find it useful to provide this context to the reader, and to previous literature suggesting large-scale migrations during this period.

Shifts in wet/dry conditions across the African continent 140ka to 100ka may have promoted these merger events between divergent stems. Precipitation does not neatly track interglacial cycles in Africa, and heterogeneity across regions may mean that the beginning of an arid period in eastern Africa is conversely the start of a wet period in southern Africa³³. The rapid rise in sea levels during the MIS 5e interglacial might have triggered migration inland away from the coasts, as has been suggested, e.g., for the Paleo-Agulhas plain³⁴.

Page 10 lines 288-295: this needs to be rewritten and requires more circumspection. The human fossil record is so sparse, that many scenarios could be said to fit with it. Omo Kibish, has a modern braincase, but in other aspects is well beyond the range of variation of contemporary humans. Jebel Irhoud has a more modern face, but a longer, flatter braincase. I am not sure why the authors then jump to Homo naledi which is a completely different hominin species. If this is done to suggest that naledi could not have been a source of ancestry, I understand, but it is written somewhat awkwardly here. In any case, other large brained hominins were in Africa at the time like heidelbergensis which seems to be much more of a candidate for possible admixture than naledi, which has not even been associated with stone tools at this point, and a small cranial capacity. I see that another reviewer raised this. While the authors respond well in the response to reviewers, I don't see enough of these changes in the main text. In particular the authors state that "Without detailed morphological analyses, which seemed out of scope, it is difficult to say if fossils from Jebel Irhoud or Kabwe/Broken Hill could have come from populations that were part of the weakly structured stem.". This is a little disingenuous since there is no suggestion that Kabwe and Jebel Irhoud were the same species. There is broad agreement that Jebel Irhoud as a fossil at the root of H. sapiens and

that Kabwe represents heidelbergensis. Detailed morphological assessments already exist and could easily be cited.

We have revised this section to hopefully address these concerns. As mentioned above, what we tried to do in this section is to emphasize differences in interpretation between previously proposed models (with archaic admixture) and the present model (with weak structure). The purpose of this paragraph is to discuss the range of morphologies of the fossils that date to the time period of the stems and how accepting the weakly structured stem model would change the genetics support for each as possible representatives for the ghost branches.

Reviewer Reports on the Second Revision:

Referees' comments:

Referee #1 (Remarks to the Author):

The authors addressed all of my comments sufficiently

Referee #2 (Remarks to the Author):

The authors have addressed all my concerns.

Referee #4 (Remarks to the Author):

I would like to thank the authors for responding to all my previous comments. I am pleased to see that the new analysis based on new simulations helped to strengthen their conclusions. I have no other comments. The manuscript is well written, the methods are well described and the novel and important results of this study are well supported by the data and by all the supplemental analysis performed.

Referee #5 (Remarks to the Author):

I think the authors did a really good job at addressing referee concerns, and I would like to see it published. However, there are still a few sticking points that mainly concern with being more explicit about possible shortcomings - thus giving a broad audience to possibility to better evaluate the results.

1) The out of Africa estimate remains far too young - something that several reviewers pick up and have some issues with. The authors have really tried to address this, and built arguments as to why the young estimates do not represent systematic bias in their model. Other reviewers more qualified than me in these methods can judge the robustness of these quantitative arguments. However, at the end of the day it is desirable for this issue to be made much more explicit in the main text. Nature papers are read by a very wide variety of scholars with different expertise and currently I can see many reading this paper without really grasping this issue transparently. Currently the authors added a section at the end of the 'Formation of Population Structure in Africa' part of the paper that says "However, one exception was the inferred Out-of-Africa and eastern/western African divergences which were 10–15ka younger than our fixed parameters. The complexity of the OOA expansion warrants the inclusion of additional Eurasian populations and more detailed modeling." This is just not going to be clear enough to many readers. I'd like to see a simple sentence making it very clear that the OOA estimate is too young based on what we know from archaeological, palaeontological and other genetic studies', or something to this effect. This should then be followed by a second sentence referring to the possibility of systematic bias that is addressed

in the SI, with a simple statement of why the authors do not believe this affects the key results of the paper. I particularly agree with reviewer 2 who states that "While the authors have pointed out some of the limitations and presented the caveats, these are scattered along the manuscript and the supplementary materials and leave a lot of room for over or misinterpretation." I note that the authors rejected this suggestion. Although they do attempt to include more caveats in the text I do not find that they are accessible or clear enough yet for a wide audience and may still lead to misinterpretation. I think this is a really interesting paper, but it is really important to be crystal clear as to possible limitations and issues and spell these out to a broad audience in simple terms.

2) Page 3 line 114..."supported by numerous archaeological and genetic studies" really needs references or at least a reference to a relevant SI section. SI section 3.1 is rather sparse on these details too, so a simple reference to this section is inadequate. I don't expect an exhaustive archaeological discussion but there are some pretty key papers here that are missing. I'd suggest looking at a mixture of key reviews and primary data papers - below are a few examples.

Groucutt et al., 2015. Rethinking the dispersal of Homo sapiens out of Africa. *Evolutionary Anthropology* 24, 149-164. doi:10.1002/evan.21455.

Groucutt et al., 2018. Homo sapiens in Arabia 85 thousand years ago. *Nature Ecology and Evolution* 2, 800–809.

Hershkovitz et al., 2018. The earliest modern humans outside Africa. *Science* 359, 456-459.

Author Rebuttals to Second Revision:

Referees' comments:

Referee #1 (Remarks to the Author):

The authors addressed all of my comments sufficiently

Referee #2 (Remarks to the Author):

The authors have addressed all my concerns.

Referee #4 (Remarks to the Author):

I would like to thank the authors for responding to all my previous comments. I am pleased to see that the new analysis based on new simulations helped to strengthen their conclusions. I have no other comments. The manuscript is well written, the methods are well described and the novel and important results of this study are well supported by the data and by all the supplemental analysis performed.

Referee #5 (Remarks to the Author):

I think the authors did a really good job at addressing referee concerns, and I would like to see it published. However, there are still a few sticking points that mainly concern with being more explicit about possible shortcomings - thus giving a broad audience to possibility to better evaluate the results.

1) The out of Africa estimate remains far too young - something that several reviewers pick up and have some issues with. The authors have really tried to address this, and built arguments as to why

the young estimates do not represent systematic bias in their model. Other reviewers more qualified than me in these methods can judge the robustness of these quantitative arguments. However, at the end of the day it is desirable for this issue to be made much more explicit in the main text. Nature papers are read by a very wide variety of scholars with different expertise and currently I can see many reading this paper without really grasping this issue transparently. Currently the authors added a section at the end of the 'Formation of Population Structure in Africa' part of the paper that says "However, one exception was the inferred Out-of-Africa and eastern/western African divergences which were 10–15ka younger than our fixed parameters. The complexity of the OOA expansion warrants the inclusion of additional Eurasian populations and more detailed modeling." This is just not going to be clear enough to many readers. I'd like to see a simple sentence making it very clear that the OOA estimate is too young based on what we know from archaeological, palaeontological and other genetic studies', or something to this effect.

This should then be followed by a second sentence referring to the possibility of systematic bias that is addressed in the SI, with a simple statement of why the authors do not believe this affects the key results of the paper.

I particularly agree with reviewer 2 who states that "While the authors have pointed out some of the limitations and presented the caveats, these are scattered along the manuscript and the supplementary materials and leave a lot of room for over or misinterpretation." I note that the authors rejected this suggestion. Although they do attempt to include more caveats in the text I do not find that they are accessible or clear enough yet for a wide audience and may still lead to misinterpretation. I think this is a really interesting paper, but it is really important to be crystal clear as to possible limitations and issues and spell these out to a broad audience in simple terms.

We thank the reviewer for their clear guidelines as to modifying this section in the manuscript. We have added three sentences clarifying these issues:

However, one exception was the inferred Out-of-Africa and eastern/western African divergences which were 10–15ka younger than our fixed parameters. *These younger dates are at odds with the accepted timing of the Out-of-Africa expansion that contributed to later human populations at approximately 50ka, as based on archaeological, climatic and fossil information^{35,36,37,38}. Because the inference approach is unbiased in simulations, we interpret the free estimate for eastern African vs. European divergence as reflecting our inclusion of only a single Out-of-Africa population in the model, the lack of a nearby source for back-to-Africa gene flow, and other regionally complex parameters, rather than to a systematic bias that may affect all parameters in the model. Older pan-African features of our inferred models are minimally affected by the choice of these fixed parameters (SI Section 7.2).*

2) Page 3 line 114..."supported by numerous archaeological and genetic studies" really needs references or at least a reference to a relevant SI section. SI section 3.1 is rather sparse on these details too, so a simple reference to this section is inadequate. I don't expect an exhaustive archaeological discussion but there are some pretty key papers here that are missing. I'd suggest looking at a mixture of key reviews and primary data papers - below are a few examples.

Groucutt et al., 2015. Rethinking the dispersal of Homo sapiens out of Africa. *Evolutionary Anthropology* 24, 149-164. doi:10.1002/evan.21455.

Groucutt et al., 2018. Homo sapiens in Arabia 85 thousand years ago. *Nature Ecology and Evolution* 2, 800–809.

Hershkovitz et al., 2018. The earliest modern humans outside Africa. *Science* 359, 456-459.

Thank you for the suggestions. Section SI 3.1 contains a list of fixed parameters, with citations for each choice of date. As these are wide-ranging (from Holocene expansions to Out-of-Africa), it is difficult to choose just a few for the main text. The reviewer appears most concerned with the evidence for an Out of Africa expansion at or prior to 50ka. We have now added the Groucutt et al (2015) citation to the “Formation of population structure in Africa” section. We have added a citation to Bergstrom et al. in the sentence below, which reviews paleoanthropological and genetic evidence for major events in human prehistory. We absolutely agree with the reviewer that fleshing out some of the complexity of adjudicating between paleo/archaeological evidence and genetic data around the Out of Africa event is helpful, and have expanded Section SI 3.1 considerably, including a careful evaluation of the timing of the Out of Africa expansion and admixture with Neanderthals in Section SI 3.1.1 (“Evidence for the Out of Africa timing”).

“Given this consistency in inferred recent history and the numerical challenge of optimizing a large number of parameters, we fixed several parameters related to recent population history so as to focus on more ancient events (SI Section 3.1). These parameters were primarily ones supported by multiple genetic and archaeological studies [Bergstrom et al. 2021, Nature].”